



# Bending of the concentration discharge relationship can inform about in-stream nitrate removal

Joni Dehaspe[1], Fanny Sarrazin[2], Rohini Kumar[2], Jan H. Fleckenstein[1,3] and Andreas Musolff[1]

[1]Department of Hydrogeology, UFZ- Helmholtz-Centre for Environmental Research, 04318 Leipzig, Germany
[2]Department Computational Hydrosystems, UFZ - Helmholtz-Centre for Environmental Research, 04318 Leipzig, Germany
[3]Bayreuth Center of Ecology and Environmental Research, University of Bayreuth, 95440 Bayreuth, Germany

*Correspondence to*: Joni Dehaspe (joni.dehaspe@ufz.de)

**Abstract.** Nitrate ($NO_3^-$) excess in rivers harms aquatic ecosystems and can induce detrimental algae growths in coastal areas. Riverine $NO_3^-$ uptake is a crucial element of the catchment scale nitrogen balance and can be measured at small spatiotemporal

scales while at the scale of entire river networks, uptake measurements are rarely available. Concurrent, low frequency $NO_3^-$ concentration and stream flow (Q) observations at a basin outlet, however, are commonly monitored and can be analyzed in terms of concentration discharge (C-Q) relationships. Previous studies suggest that more positive log(C)-log(Q) slopes under low flow conditions (than under high flows) are linked to biological $NO_3^-$ uptake, creating a bent rather than linear log(C)-log(Q) relationship. Here we explore if network scale $NO_3^-$ uptake creates bent log(C)-log(Q) relationships and when in turn

uptake can be quantified from observed low frequency C-Q data. To this end we apply a parsimonious mass balance based river network uptake model in 13 mesoscale German catchments (21-1450 km²) and explore the linkages between log(C)-log(Q) bending and different model-parameter combinations. The modelling results show that uptake and transport in the river network can create bent log(C)-log(Q) relationships at the basin outlet from log-log linear C-Q relationships describing the $NO_3^-$ land to stream transfer. We find that the bending is mainly shaped by geomorphological parameters that control the

channel reactive surface area rather than by the biological uptake velocity itself. Further we show that in this exploratory modelling environment, bending is positively correlated to percentage $NO_3^-$ load removed in the network ($L_{r.perc}$) but that network wide flow velocities should be taken into account when interpreting log(C)-log(Q) bending. Classification trees, finally, can successfully predict classes of low (~4 %), intermediate (~32 %) and high (~68 %) $L_{r.perc}$ using information on water velocity and log(C)-log(Q) bending. These results can help to identify stream networks that efficiently attenuate $NO_3^-$

loads based on low frequency $NO_3^-$ and Q observations and generally show the importance of the channel geomorphology on the emerging log(C)-log(Q) bending at network scales.

## 1 Introduction

Transport and transformation of nitrate ($NO_3^-$) in river networks are major controls of downstream exports to receiving lakes, reservoirs and coastal systems (Alexander et al., 2000; Billen et al., 1991; Peterson et al., 2001; Seitzinger et al., 2002; Seybold



and McGlynn, 2018). Increased $NO_3^-$ concentrations in surface waters can induce detrimental algae growths (Beusen et al., 2016; Canfield et al., 2010; Galloway et al., 1995), compromise river ecosystem health and jeopardize drinking water supplies. Since the beginning of the 20$^{th}$ century, human activities such as agricultural expansion and fossil fuel burning have mobilized additional reactive nitrogen (N), initiating and later exacerbating this problem (Seitzinger et al., 2002). In arable landscapes, which include large parts of Europe, the efficient management of aquatic $NO_3^-$ at network scales is complicated by the

spatiotemporal variability of loading patterns and hydrologic regimes as well as the lack of understanding of nutrient pathways, connected transit times and removal processes from input to export. Nevertheless, nitrate concentration and load variability can be predicted at catchment scales when relying on detailed process understanding regarding transport and biogeochemical processing (Alexander et al., 2009; Schlesinger et al., 2006; Wollheim et al., 2008). Moving beyond small scale variability and characterizing nitrate processing at the catchment scale however remains a challenge (McDonnell et al., 2007; Li et al., 2020).

Within river reaches and streams, reactive solutes like $NO_3^-$ are affected by complex interactions of physical, biological and chemical processes. Physical transport is driven by local discharge and channel geomorphology and dictates the $NO_3^-$ residence time in a reach, thus influencing the timescales at which biogeochemical processing can take place (Kirchner et al., 2000; Runkel and Bencala, 1995; Zarnetske et al., 2011). $NO_3^-$ is removed and transformed by denitrifying bacteria in the anoxic

river sediment (Birgand et al., 2007; Peterson et al., 2001), ammonified or retained through assimilation processes in the oxic or anoxic river compartments by bacteria, fungi and primary producers such as algae and macrophytes, potentially entering higher trophic levels. In the latter case, N in the form of DON (dissolved organic nitrogen) and more commonly DIN (dissolved inorganic nitrogen), together with phosphorus (P) may be released to the water column later on (Durand et al., 2011; Vanni, 2002; Vanni and McIntyre, 2016). The nutrient spiraling model (Newbold et al., 1981; Stream Solute Workshop, 1990) that

formally describes these processes has been used to quantify and compare $NO_3^-$ transport and uptake (the net result of all removal and release processes) in river reaches (Peterson et al., 2001; Mulholland et al., 2008; Hall et al., 2009) and stream networks (Ensign et al., 2006; Doyle, 2005; Marce and Armengol, 2009). Quantifying in situ $NO_3^-$ uptake is labor intensive and may involve local nutrient additions, potentially altering the ambient uptake rate (Hensley et al., 2014; Mulholland and Tank, 2002). Other methods require high frequency measurements (Jarvie et al., 2018; Kunz et al., 2017) that are mostly

limited to small spatial scales (i.e. reach scale) and can vary considerably between measuring points (Boyer et al., 2006). At the scale of entire river networks contrarily, uptake measurements are rarely available (but see Wollheim et al., 2017; Hansen et al 2018) and models are applied instead to predict spatiotemporal uptake patterns (Boyer et al., 2006; Yang et al., 2018). These models account for the spatial configuration of the stream network, an important aspect for stream biogeochemistry that is often ignored in small scale experimental approaches (Fisher et al., 2004). Spatially distributed models however, require

calibration of uncertain spatiotemporal parameters and may not reflect the essential features of the system despite fitting observed data well (Klemes, 1986).





River networks link terrestrial source zones to coastal areas and integrate biogeochemical and hydrological catchment functions across scales (Bouwman et al., 2012; Helton et al., 2018). Small streams (usually headwaters) are known to influence

the export signal disproportionally because of their overall (high) contribution to total stream length and effective $NO_3^-$ removal capacity (Alexander et al., 2000; Horton, 1945), explained by high sediment surface to water volume ratios. Generally, high removal efficiencies have been reported for river network areas with lower specific discharges (Hall et al., 2009; Hensley et al., 2014), under favorable circumstances such as high light availability, heavy in-stream vegetation (Hensley et al., 2014; Rode et al., 2016) and for streams with a high capacity for lateral and hyporheic exchange (Gomez-Velez et al., 2015; Kiel

and Cardenas, 2014). The scaling of in-stream uptake processes beyond the river reach has been approached by combined experimental-modelling studies with defined explicit scaling relationships (e.g. Basu et al., 2011; Aguilera et al., 2013; Bertuzzo et al., 2017; Lindgren and Destouni, 2004) and theoretical frameworks explaining how the river network capacity regulates solute export (e.g. Wollheim et al., 2018). Abbott et al. (2018) shows how spatially heterogeneous patterns of water chemistry stabilize while the temporal variability of nutrient concentrations persists when moving downstream, facilitating the

temporal scaling of headwater measurements. Nevertheless, insights into linking the interplay of nitrate removal processes at the network scale to downstream export patterns in space and time are largely missing.

Concentrations (C) for in-stream solutes such as carbon, major ions, particulates and nutrients are commonly monitored concurrently with discharge (Q) at the basin outlet. C-Q relationships integrate the effect of biogeochemical and hydrological

processes within the catchment and have mainly been discussed in terms of land-stream transfer and source configuration in catchments as well as subsurface retention processes (Godsey et al., 2009; Musolff et al., 2017, Bieroza et al., 2018). The shape of long term (multiple years) C-Q relationships in the log-log space is typically described by the slope of a linear regression model (Godsey et al., 2009). Here, three archetypes have been distinguished; (i) a positive log(C)-log(Q) slope, indicating enrichment, occurs when an increasing discharge additionally mobilizes solutes, (ii) a negative C-Q slope or dilution pattern

is commonly linked with source limitations and (iii) a neutral or chemostatic slope implies low variability in in-stream concentrations across a high range of discharges, a pattern observed for example for solutes derived from weathering bedrock (Ameli et al., 2017; Godsey et al., 2009). The potential information loss associated with linear and monotonic $NO_3^-$ log(C)-log(Q) analysis was addressed by Moatar et al. (2017) and Minaudo et al. (2019) for more than 200 French catchments and by Diamond and Cohen (2018) for 44 rivers in Florida, USA. These studies identified distinct linear low-flow and high-flow $NO_3^-$

log(C)-log(Q) regression slopes for a majority of the cases. Moatar et al. (2017) found that stronger positive slopes under low flow conditions correlate positively with chlorophyll-a concentrations (associated with biological processes) and attributed this condition to biological $NO_3^-$ concentration mediation in the stream. This is consistent with the findings of Hall et al. (2009) and Hensley et al. (2014) among others that in-stream uptake is more efficient under low-flow than under high-flow. Furthermore, Wollheim et al. (2017) illustrates non-linear $NO_3^-$ C-Q relationships conceptually for storm flow dynamics in a

river network, showing high retention capacities in the headwater catchments that decrease under increasing flows, changing the slope of C-Q relationships from dilution to enrichment. Based on these studies we hypothesize that the magnitude (or





efficiency) of in-stream $NO_3^-$ uptake is encoded within observed C-Q relationships, and their analysis therefore can improve our understanding of in-stream uptake processes through providing an alternative to elaborate field and modelling work aimed at quantifying $NO_3^-$ removal in stream networks. Low frequency $NO_3^-$ observations are widely available (e.g. biweekly to

monthly grab sampling, Ebeling et al., 2020 rev; Minaudo et al., 2019; Moatar et al., 2017) but if and how this data can be utilized to characterize catchment scale in-stream processing has yet to be investigated.

In this paper, it is postulated that network-scale uptake effects can be inferred from the non-linearity or bending of low-frequency, multi-annual concentration (C) and discharge (Q) observations. To test this hypothesis, we apply a parsimonious

river network model (similar to Bertuzzo et al., 2017; Helton et al., 2018; Helton et al., 2010; Mulholland et al., 2008) in 13 German catchments to explore the catchment scale transport and uptake processes that influence downstream log(C)-log(Q) patterns. The specific objectives are to (i) introduce *Curvature* as a robust metric to quantify bending of low frequency C-Q time series in the log-log space; (ii) explore the sensitivity of C-Q bending to hydrological and in-stream biogeochemical parameters (e.g. channel shape, water velocity and biological $NO_3^-$ uptake velocity); (iii) explore how C-Q bending is linked

to network scale in-stream uptake; (iv) provide guidelines if and on under what circumstances the C-Q bending can offer conclusive information on effective in-stream uptake. In this proof of concept exploratory study, we demonstrate how (existing) low-frequency monitoring data can be effectively utilized to quantify nitrate uptake in river networks and show how small scale uptake processes shape emerging patterns at catchment scales.

## 2 Methods

### 2.1 *Curvature*

The shape of C-Q relationships are often described as linear (Bieroza et al., 2018; Godsey et al., 2009; Musolff et al., 2017) or segmented linear (Meybeck and Moatar, 2012; Moatar et al., 2017; Marinos et al., 2020), implying a limit on the possible C-Q shapes and setting assumptions such as 'fixed breaking points'. Here, we introduce the concept of *Curvature* to quantify rather than describe the shape of "broken-stick" C-Q relationships, without the assumption of a fixed form. In a strict

geometrical sense the curvature (-∞; +∞) is the instantaneous rate of change of direction of a point that moves on a curve. A straight line for example has a curvature of zero and a large circle has a smaller curvature than a small circle (Pressley, 2010) (Fig. B1). Here, *Curvature* identifies the magnitude and direction of the log(C)-log(Q) section with the largest instantaneous change. To calculate *Curvature* for an observed (noisy) C-Q relationship, a smoothed spline, $f$, is iteratively fitted with increasing degrees of freedom (df) to capture the general log(C)-log(Q) shape accurately but avoid overfitting. Initially, df =

3 and the log(Q) region of the largest instantaneous change is identified as $Q_m \pm 0.05$ with $Q_m = \underset{\log Q}{arg\,max}|f''|$. Then, df is increased until, at df=i, the log(Q) corresponding to the largest instantaneous change is not within the initial $Q_m$ region



anymore. Consequently, *Curvature* is calculated for a smoothed spline fit, $f$, with df = i-1 as $\begin{cases} \max\limits_{\log Q} f'' \ if \ \left|\max\limits_{\log Q} f''\right| \geq \left|\min\limits_{\log Q} f''\right| \\ \min\limits_{\log Q} f'' \ if \ \left|\max\limits_{\log Q} f''\right| < \left|\min\limits_{\log Q} f''\right| \end{cases}$.

*Curvature* of a log(C)-log(Q) relationship could be considered as a complementary metric to the slope of the linear regression model (Godsey et al., 2009) and could serve as an alternative for segmented linear regression fits (Meybeck and Moatar, 2012; Moatar et al., 2017; Marinos et al., 2020) (Fig. B2) as it quantifies bending.

We assume here that a multi-annual (6 to 15 years) low frequency (biweekly to monthly) C-Q relationship without temporal (significant) trends in a given station has one *Curvature*. To verify this assumption in a realistic setting, *Curvature* was computed for observed nitrate (C) and Q data (1995-2010) of French water quality stations with biweekly to monthly sampling frequencies (Dupas et al., 2019). Following the removal of C outliers (falling outside of μ±3.5σ in the log space, with μ and σ representing the sample mean and standard deviation, respectively) 444 stations were selected that satisfy the following four criteria: (i) the station should have at least 70 coupled C and Q observations (total number of samples, n∈[70;402]); (ii) a minimum of 6 years of data are represented; (iii) there is no bias in the intra annual distribution of the data (i.e. never less than 10% of the C-Q observations in one season); and (iv) the station C observations had no significant temporal trends (Mann Kendall test, p-value > 0.05) (Ebeling et al., 2020 rev). We then assessed the robustness of *Curvature* to the low frequency C-Q observations in a time series by selecting different subsamples of C-Q data from the entire available time series for a given station. More specifically, 100 random time series subsamples (each with a minimum length of 70) without replacement but with overlap were taken for each station, with the subsamples passing the four criteria above, and *Curvature* was calculated for each subsampled time series. On average, the subsamples represented nearly 80% of the complete time series for a station.

The estimated *Curvature* for the observed $NO_3^-$ log(C)-log(Q) data ranges between -5.25 and 3.88 (median is -0.23, Fig. B3) and 77% of the stations are characterized by *Curvature* ≤ 0 or a linear or concave shape (similar to Moatar et al. 2017). The time series subsamples for each station generally had a small *Curvature* variability (Interquartile Range, IQR for a given station below 1) for 93% of the stations with some exceptions demonstrating a larger IQR up to 8. This indicates *Curvature* quantification for most low frequency C-Q time series is robust. The Spearman rank correlation ($\rho = 0.53$, p-value < 2.2e-16) between the absolute observed *Curvature* and IQR for each station is significant and positive, implying that C-Q relationships with a higher absolute *Curvature* have a higher uncertainty when quantifying the C-Q bending. However, *Curvature* variability (IQR) in the subsamples for each station has no significant correlation with the number of data points available for one station. This implies that *Curvature* tends to be temporally robust when the C-Q data obeys the four above criteria so that the length of the low frequency time series length does not impact the estimated *Curvature*. Overall, the proposed *Curvature* metric – quantifying the C-Q bending- is suitable to describe bending in multiannual, temporally stable log(C)-log(Q) relationships.



## 2.2 Network Model

In this work an explorative grid based (100 m x 100 m) mass balance network model (comparable to Bertuzzo et al., 2017; Helton et al., 2018; Helton et al., 2010; Mulholland et al., 2008 and conceptually shown in Fig. B4) was used to simulate in-
stream nitrate transport and biological removal on a daily basis. The model was developed in R (R Core Team, 2013).

### 2.2.1 Stream network and hydrological properties

Following Bertuzzo et al. (2017) and Helton et al. (2018), each river network node (i.e. grid cell) $i$ ($1 \leq i \leq N$) has a drainage area $A_i$ [m²] that is calculated as the sum of the total upstream drainage area $\sum_j W_{ji} A_j$ [m²] and the direct drainage area $a_i$ [m²] (e.g. laterally contributing drainage area) to grid cell $i$ (Eq. (1)):

$$A_i = \underbrace{\sum_j W_{ji} A_j}_{Upstream\ Drainage\ Areas} + \underbrace{a_i}_{Direct\ Drainage\ Area} \qquad (1)$$

where $W_{ji}$ [-] is an element in the connectivity matrix $W$ (N x N) such that $W_{ji} = 1$ if $j$ is directly neighboring and flowing into $i$ and $W_{ji} = 0$ otherwise. $A_j$ [m²] is the total drainage area to node $j$.

The total local discharge $Q_i$ at a given grid cell $i$ is proportional to the total drainage area at that grid cell, $A_i$, [m³ s⁻¹] (following
Bergstrom et al., 2016 and Bertuzzo et al., 2017) (Eq. (2)), which in turn dictates the downstream hydraulic geometry relationships of river geomorphic parameters channel width, $w_i$ [m], and average channel depth, $d_i$ [m] (Leopold and Maddock, 1953) (Eq. (2.1) and Eq. (2.2)). The local velocity in a grid cell $v_i$ [m s⁻¹] is calculated according to Eq. (2.3) and the corresponding travel time, $T_i$ [days] is computed in Eq. (2.4):

$$Q_i = Q_{t.sp} * A_i \qquad (2)$$

$$w_i = K_w * Q_i^{a_w} \qquad (2.1)$$

$$d_i = K_d * Q_i^{a_d} \qquad (2.2)$$

$$v_i = \frac{Q_i}{w_i * d_i} \qquad (2.3)$$

$$T_i = \frac{l_i}{v_i} \qquad (2.4)$$

where $Q_{t.sp}$ [m³ s⁻¹] is the specific discharge that is calculated as the ratio of the daily discharge at the outlet and the total
number of catchment grid cells, consequently varying in time. Parameters $a_w$ [-] and $K_w$ [-] are the respective exponent and coefficient parameters in the river width-discharge relationship (Eq. (2.1)); while $a_d$ [-] and $K_d$ [-] compose the exponent and coefficient parameters of the depth-discharge relationship (Eq. (2.2)), respectively. The flow length through a grid cell $i$, $l_i$ [m], equals 100 or $100\sqrt{2}$ m for horizontal/vertical or diagonal flow directions, respectively. The ratio of $a_d$ to $a_w$ corresponds to a parameter $r$ [-]$\in \mathbb{R}^+$ which prescribes the cross section geometry relation such that a triangular channel cross
section is represented by $r = 1$, a parabolic channel cross section by $r = 2$ and channel cross sections with progressively flatter





bottoms and steeper banks by increasing values of $r$ (Dingman, 2007). The width-discharge relation in Eq. (2.1) is conceptually illustrated in Fig. B5 for two sets of $a_w$ and $K_w$.

### 2.2.2 Nitrate uptake

Similar to Eq. (1) the incoming load, $L_{in,i}$ [mg s$^{-1}$], to a river network grid cell $i$ is the sum of upstream load contributions $L_{in.up,i}$ [mg s$^{-1}$] and direct land to stream loading $L_{in.ls,i}$ [mg s$^{-1}$], given that L = CQ (Eq. (3)). The contribution of direct land to stream loading concentration can be expressed as a power law (Musolff et al. 2017) with the exponent $b$ [-], the slope in the log(C)-log(Q) relationship that is an indicator of the C-Q archetype (Godsey et al., 2009) and coefficient $c$ [-]. Following Jawitz and Mitchell (2011), the coefficient $c$ is calculated to yield the long-term mean in-stream input concentration $C_{mean}$ [mg L$^{-1}$] (Eq. (A1)). Additional NO$_3^-$ sources such as the load resulting from NO$_3^-$ release within the stream network as point sources are not considered here (similar to Bertuzzo et al., 2017; Wollheim et al., 2006), so that only concave or linear log(C)-log(Q) patterns (*Curvature* $\leq$ 0) can be simulated. Also, we do not consider other loading processes that may create bending at the catchment outlet (e.g., shifts in transport pathways and solute sources, Marinos et al. 2020).

$$L_{in,i} = \underbrace{L_{in.up,i}}_{Upstream\ Loads} + \underbrace{L_{in.ls,i}}_{Direct\ Land\ to\ stream\ Loading} = \sum_j W_{ji}L_j + c * (Q_{t.sp} * a_i)^{b+1} \tag{3}$$

The modelled in-stream NO$_3^-$ uptake follows first order removal kinetics (Alexander et al., 2000; Boyer et al., 2006; Ensign and Doyle, 2006), such that the outgoing load from grid cell $i$, $L_i$ [mg s$^{-1}$] is a fraction of the incoming load $L_{in,i}$ (Eq. (4)) and the absolute removed load $L_{r,i}$ [mg s$^{-1}$] can be described as (Eq. (5)) with the in-stream processing chiefly occurring at the sediments and biofilm at the benthic-pelagic interface (Wollheim et al., 2006). Here, $L_{r,i}$ is influenced by separate hydrological ($\frac{P_i * l_i}{Q_i}$) and biological ($v_f$) components (similar to Bertuzzo et al., 2017).

$$L_i = L_{in,i} * e^{-\frac{v_f * P_i * l_i}{Q_i}} \tag{4}$$

$$L_{r,i} = L_{in,i} - L_i = L_{in,i} * (1 - e^{-\frac{v_f * P_i * l_i}{Q_i}}) \tag{5}$$

where $P_i$ is the cross section wetted perimeter calculated from the Manning equation (using the bed slope $S_i$ and assuming a fixed roughness coefficient = 0.03 [m m$^{-1}$]) in open channels (Eq. (A2)). The uptake velocity parameter $v_f$ [m day$^{-1}$] indicates the rivers total biological nutrient demand (areal uptake, $U$ [mg m$^{-2}$ day$^{-1}$] relative to in-stream concentration $C_{mean}$ [mg L$^{-1}$]) with $v_f = k_i d_i$ and $k_i$ the first order removal constant (Ensign and Doyle, 2006; Wollheim et al., 2006). The parameter $v_f$ accounts for the processes altering the rate and form of downstream NO$_3^-$ delivery (Doyle, 2005) and is therefore not limited to denitrification only. We assume that $v_f$ is independent of the in-stream NO$_3^-$ concentration $C_{mean}$ (Pennino et al., 2014; O'Brien et al., 2007) such that the areal uptake rate $U = v_f * C_{mean}$ is tightly linked with $C_{mean}$ in a first order relationship. Others (e.g., Hensley et al., 2014; Mulholland et al., 2008; O'Brien et al., 2007) contrarily found explicit scaling relationships where $v_f$ decreases non-linearly for increasing $C_{mean}$ (10$^{-4}$ – 10$^1$ mg L$^{-1}$) when considering distinct catchments. However, in



Germany, the $NO_3^-$ concentration range across a range of catchments is small ($10^{-1}$ – $10^1$ mg L$^{-1}$ according to Ebeling et al.,
2020 rev) and rivers generally have minor longitudinal concentration variability (Hensley et al., 2014; Ensign and Doyle, 2006)
which suggest independent definitions of $v_f$ and $C_{mean}$.

The Damköhler number $Da$ [-] is calculated as the ratio between transport ($\tau_T$) and reaction ($\tau_R$) timescales and is often used
to characterize the relative importance of hydrological and biogeochemical processes in hydrological connected systems
(Oldham et al., 2013; Kumar et al., 2020):

$$Da = \frac{\tau_T}{\tau_R} = \frac{TT}{k^{-1}} \qquad (6)$$

where, $\tau_T$ represents as the effective travel time, $TT$ [days] or the exposure time scale under advective conditions. We
estimated the catchment wide $TT$ as the spatiotemporal median of the sum of all downstream $T_i$ (Eq. (2.4)) for a grid cell in
the network ($\sum_i^{0ut} T_i$) (similar to Bergstrom et al., 2016). Whereas $\tau_R$ represents the reactive time scale of biological processes.
It is approximated as $k^{-1}$ [days$^{-1}$] with the effective catchment wide $k$ estimated as the spatiotemporal median of the grid-
scale first order reaction constant $k_i = d_i/v_f$.

### 2.3 Exploring *Curvature* with Monte Carlo Simulations

Monte Carlo simulations are performed to explore how *Curvature* evolves from a range of model input parameter combinations
in a variety of catchments (Sect 2.3.1 below). These simulations utilize the same set of 11107 unique parameter combinations
in each of study catchments that, during one model run, are each kept constant in time and uniform in space for simplicity. The
unique parameter combinations are generated by Latin Hypercube sampling from uniform parameter ranges that are set
according to literature values (Table 1). Some physical constraints were also imposed such that the channel geometry
parameters $a_w$ and $a_d$ must obey continuity principles ($a_w + a_d < 1$ and $a_w > a_d$, following Leopold and Maddock, 1953).
The main simulation results are i) *Curvature* [-], deduced from simulated log(C)-log(Q) relationships when minimum 80 % of
the C data is above the 'detection limit' of 0.002 mg L$^{-1}$ $NO_3^-$; and ii) the network wide percentage load removed $L_{r.perc}$ [%].
The latter is calculated as the median of the ratio between the daily absolute removed load and the daily absolute incoming
load in the river network. While all outputs can be spatially and temporally explicit on a daily time step, *Curvature* is examined
at the catchment outlet, integrating both spatial and temporal aspects. The Monte Carlo results are subsequently subjected to a
global sensitivity analysis with the PAWN method (Pianosi and Wagener; 2015) to elucidate influential model parameters.
Furthermore a correlation analysis is conducted to explore how these influential parameters impact simulated *Curvature*.
Finally, a Classification and Regression Tree algorithm (CART, Breiman et al., 1984) allowed us to visualize parameter
interactions as detailed in Sect. 2.3.2 below.

**Table 1: Network model parameter ranges for the Monte Carlo simulations.**

| Parameter | Unit | Description | Range | References |
|-----------|------|-------------|-------|------------|





| $v_f$ | [m day$^{-1}$] | Uptake velocity | $10^{-4}$; 0.25 | Marce and Armengol, 2009 |
|---|---|---|---|---|
| $b$ | [-] | Slope b lin. regress. log(C)-log(Q) | -1.5; 1.5 | Musolff et al., 2017, Ebeling, 2020b |
| $C_{mean}$ | [mg L$^{-1}$] | Land to stream concentration | $10^{-4}$; 20 | Ebeling, 2020b |
| $K_w$ | [-] | Coefficient width-Q function | 2.6; 20.2 | Andreadis et al., 2013 |
| $a_w$ | [-] | Exponent width-Q function | 0.01; 0.54 | Andreadis et al., 2013; Dingman, 2007 |
| $K_d$ | [-] | Coefficient depth-Q function | 0.12; 0.63 | Andreadis et al., 2013 |
| $a_d$ | [-] | Exponent depth-Q functiom | 0.28; 0.667 | Andreadis et al., 2013; Dingman, 2007 |

**2.3.1 Catchment selection**

We applied the network model in 13 mesoscale catchments across Germany with varying sizes (21-1450 km²) and distinct geophysical settings as stream order, median discharge and catchment shape (quantified with the Horton form factor; Horton,
1945) (Table 2). The catchments were selected based on a database of water quality and catchment characteristics within Germany (Ebeling, 2020a). Three nested sub-catchments for the Selke as well as the Holtemme river system, both part of the Bode, a well-studied river system near the Harz Mountains in central Germany were included additionally (Fig. 1) (Ehrhardt et al., 2019; Rode et al., 2016; Winter et al., 2020; Mueller et al., 2018). All catchments had ~10 years of uninterrupted daily Q data available between 1995 and 2010 (Musolff, 2020). The selected catchments were delineated in ArcMap (ESRI, 2011)
from a 100 m x 100 m DEM (EEA, 2013; Ebeling et al., 2020 rev). A flow direction, flow accumulation and valley slope grid in the same resolution were established. The channel threshold drainage area for the network delineation was set to 150 grid cells (1.5 km²), which agreed well with the observed river network, resulting in a tree shaped river network with N grid cells or nodes.

**Table 2: Catchment properties summary: Catchment Area, median Elevation, Slope and Topographical Wetness Index (TWI), maximum Strahler Stream Order, Horton form factor, Drainage Density, median discharge (Q) and the coefficient of variation of the discharge (CV Q).**

| ID | River | Area | Med. elevation | Med. slope | Med. TWI | Stream order | Network length | Horton form factor | Drainage density | Med. Q | CV Q |
|---|---|---|---|---|---|---|---|---|---|---|---|
| | | [km²] | [m] | [°] | [-] | [-] | [km] | [-] | [km km$^{-2}$] | [m³ s$^{-1}$] | [-] |
| 1 | *Dahme* | 20.9 | 105 | 1.50 | 10.08 | 2 | 11 | 0.67 | 0.52 | 0.02 | 1.13 |
| 2 | *Kraichbach* | 422.5 | 164 | 2.84 | 9.45 | 4 | 228 | 0.23 | 0.54 | 0.85 | 0.47 |
| 3 | *Wertach* | 658.1 | 833 | 4.30 | 9.17 | 4 | 391 | 0.14 | 0.59 | 10.60 | 0.96 |
| 4 | *Ammer* | 713.7 | 858 | 8.34 | 8.80 | 4 | 416 | 0.29 | 0.58 | 14.98 | 0.84 |
| 5 | *Modau* | 88.6 | 272 | 5.61 | 8.47 | 3 | 47 | 0.42 | 0.53 | 0.52 | 0.80 |
| 6 | *Leine* | 993.2 | 276 | 4.40 | 8.95 | 4 | 525 | 0.45 | 0.53 | 6.22 | 0.85 |
| 7 | *Speyerbach* | 142.0 | 187 | 3.58 | 9.84 | 3 | 104 | 0.17 | 0.73 | 0.66 | 0.64 |
| 8 | *Stör* | 1452.2 | 25 | 0.90 | 10.63 | 5 | 905 | 0.46 | 0.62 | 14.10 | 0.76 |
| 9 | *Holtemme* | 272.5 | 258 | 3.58 | 9.49 | 4 | 145 | 0.17 | 0.53 | 1.04 | 1.01 |
| 10 | *Selke Silberhütte* | 94.5 | 456 | 4.02 | 8.72 | 3 | 49 | 0.27 | 0.51 | 0.56 | 1.34 |




| 11 | Selke Meisdorf | 282.1 | 342 | 3.94 | 9.03 | 3 | 160 | 0.35 | 0.57 | 0.70 | 1.34 |
| 12 | Selke Hausneindorf | 460.1 | 263 | 2.90 | 9.60 | 4 | 256 | 0.37 | 0.56 | 0.65 | 1.50 |
| 13 | Schleuse | 263.2 | 597 | 9.12 | 7.92 | 4 | 139 | 0.79 | 0.53 | 2.88 | 1.07 |

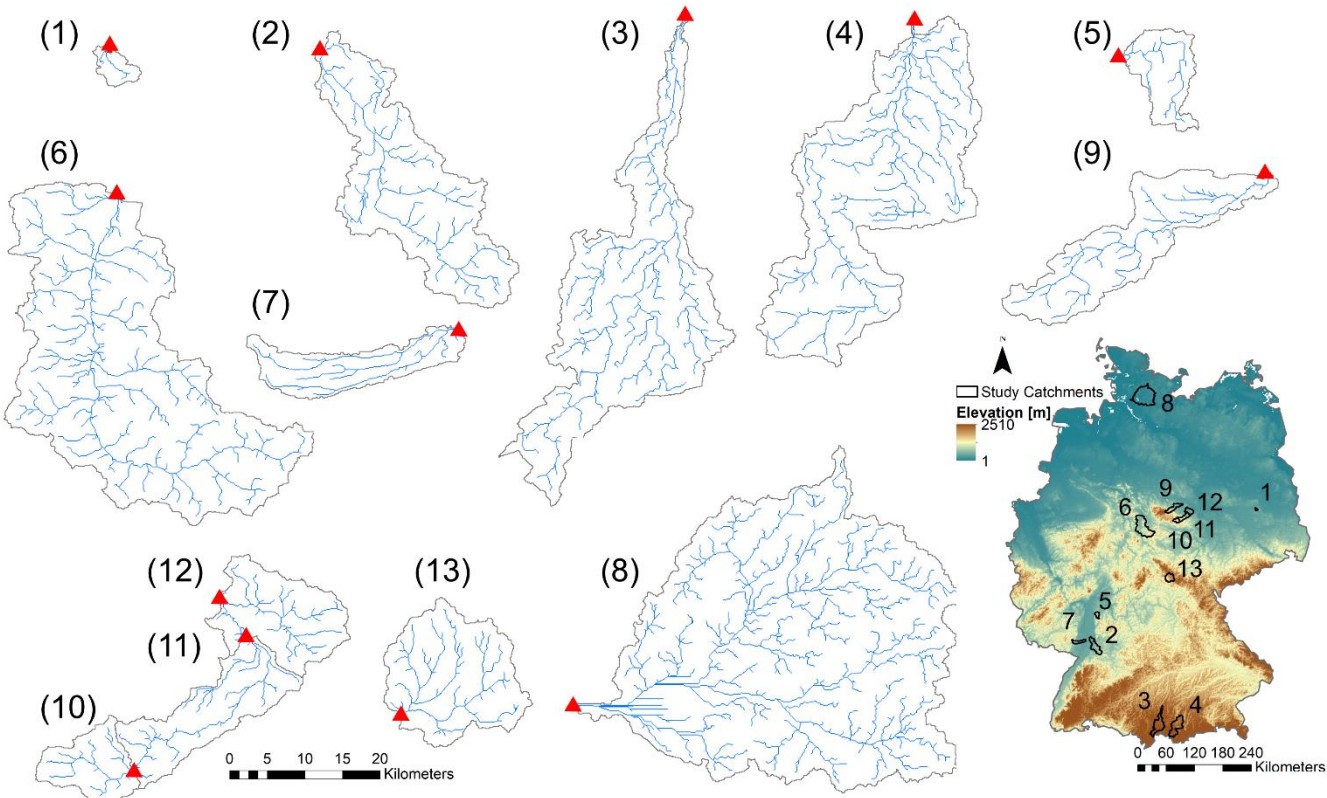

**Figure 1: Germany DEM with the location and outline (shape) of selected catchments, along with their drainage networks (in blue) and outlet location (red triangle). See Table 2 for catchment ID's and properties.**

To verify the model's ability to reproduce realistic concentration time series and *Curvature*, modeled and simulated data were compared in the Selke catchment (at Meisdorf gauging station; 282 km², Table 2) where extensive field campaigns and modelling studies have been conducted related to in-stream processes (Rode et al., 2016; Dupas et al., 2017; Yang et al., 2019; Yang et al., 2018). This relatively homogeneous upstream part of the Selke consists of forest and cropland and is characterized by consistent export regimes (Winter et al., 2020). For an input parameter combination (Table C1) set to reasonable values for this catchment (Rode et al., 2016), the land to stream $NO_3^-$ inputs averaged 1.2 kg N day$^{-1}$ km$^{-2}$ which is similar to the 1.9 kg N day$^{-1}$ km$^{-2}$ reported by Winter et al. (2020) for the Selke River (Meisdorf ); and it is well within the general 0.001 to 100 kg N day$^{-1}$ km$^{-2}$ range established by Mulholland et al. (2008). Flow velocity had a spatiotemporal median value of 0.47 m s$^{-1}$, which is also comparable with measured flow velocities (Risse-Buhl et al., 2017). Furthermore, daily discharge data, monthly





nitrate concentrations (Winter et al., 2020) and integrated uptake measurements (Rode et al., 2016; Yang et al., 2019) were available for the Selke River (Meisdorf) between 2000 and 2012. Also, the spatially explicit nature of the network model was exploited here for the set input parameter combination (Table C1) to gain an insight into how the interplay of transport and uptake processes at every network grid cell can result in a curved C-Q pattern at the catchment outlet.


### 2.3.2 PAWN sensitivity analysis and correlation analysis

We performed a global sensitivity analysis (GSA) using the moment independent PAWN method (Pianosi and Wagener (2015). The method allowed for estimating the effect of the parameter inputs on the entire model output distribution and can be applied to rank the inputs and identify the uninfluential ones. The resulting PAWN sensitivity indices were estimated from

generic input-output samples created with the numerical approximation strategy proposed by Pianosi and Wagener (2018). With this strategy, the range of variation of each input $x_i$ is partitioned into a number $n_i$ of equally probable 'conditioning' intervals ($I_{i,k}, k = 1, ..., n_i$), i.e. each interval contains the same number of data points. Given a scalar model output $y$ (here *Curvature*), the PAWN method compares the output conditional Cumulative Distribution Function (CDF) ($F_y(y)$), computed by concurrently varying all the inputs, and the $n_i$ conditional CDFs for that input ($F_{y|x_i}(y|x_i \in I_{i,k})$). Each conditional CDF

is obtained by varying all inputs within their entire range except for $x_i$, whose values are contained within one of the $n_i$ conditioning intervals. The Kolmogorov-Smirnov statistic (KS) is then calculated as the maximum vertical distance between the conditional and unconditional CDFs, while the PAWN sensitivity index ($S_i$) for input $x_i$ aggregates the results over all conditional CDFs through a summary statistic as presented in Eq. (7):

$$S_i = \underset{k=1...n_i}{\text{stat}} KS(I_{i,k}) \tag{7}$$

where $KS(I_{i,k}) = \underset{y}{\max}|F_y(y) - F_{y|x_i}(y|x_i \in I_{k,i})|$

In this study, we applied Eq. (7) using $n_i = 10$ conditioning intervals for each input parameter and the maximum KS value, $KS_{max}$, as summary statistics, which is an appropriate metric for screening non-influential input parameters. We estimated confidence intervals of the sensitivity indices using 15000 bootstrap resamples and checked the robustness of the results. The PAWN analysis was carried out using the Python version of the SAFE toolbox for global sensitivity analysis (Pianosi et al.,

300 2015).

To explore the direction of change in the C-Q bending at the catchment outlet resulting from variations in the model parameters and the catchment in-stream uptake, a Spearman rank correlation analysis was performed including all the simulated catchment responses and parameter combinations. These correlations were visualized in a correlation matrix using the 'corrplot' package

in R (Wei and Simko, 2020).

### 2.3.3 Identify parameter and model output interactions with classification tree





Finally, we aim to determine if C-Q bending at the catchment outlet (specifically *Curvature*) informs about the network wide in-stream uptake. Thereto, a recursive modelling approach is proposed, using the Classification and Regression Trees algorithm

(CART, Breiman et al., 1984) which allows for the identification of non-linear synergistic interactions among model parameters and output variables. This non-parametric method segregates classes for a response variable by progressively splitting selected predictor variables in a binary way. The resulting decision tree is simple and intuitive to interpret and can facilitate the fast characterization of river networks. The response variables include the effective catchment wide removal efficiency $L_r$, the Damköhler number $Da$ and the uptake velocity $v_f$, while the predictors are *Curvature*, the median network

velocity $v$ and all of the model input parameters except for $v_f$ (Table 1). For each response variable, three classes are defined representing low, intermediate and high ranges found in the literature (Table 3) that each contain 5 % of the simulation outputs (obtained by distributing the non-missing model simulations over 20 percentiles). The overall CART accuracy for each response variable is assessed by attributing 80 % of the simulation outputs in the low, intermediate and high classes to a training sample and assigning the remaining 20 % to a test sample. The training sample is then used to construct the classification tree

while the test sample is needed to assess the prediction accuracy and calculating the performance statistics for each class. The CART analysis was performed using the 'caret' package in R with the Gini impurity measure as splitting criterion (Kuhn, 2020).

**Table 3: Classes containing low, medium and high values for response variables $v_f$ (uptake velocity), $L_{r.perc}$ (percentage load**
**removed) and $Da$ (Damköhler number) are used for the CART training and testing samples. Similar classes are obtained for model output *Curvature*. These classes stem from distributing the non-missing simulation data over 20 percentiles and selecting the percentiles corresponding to low, medium and high literature values with the respective percentile number (1-20) indicated in brackets. The training sample for constructing the CART model was then allocated 80% of this data and the test sample 20 %.**

| Variable | Units | Low | Medium | High | References |
|---|---|---|---|---|---|
| $v_f$ | [m day$^{-1}$] | $10^{-4}$-0.01 (1) | 0.10-0.11 (10) | 0.23-0.24 (20) | Birgand et al., 2007, Marce and Armengol, 2009 |
| $L_{r.perc}$ | [%] | 3.8-5.2 (7) | 28.7-35.1 (15) | 63.0-75.3 (19) | Birgand et al., 2007 |
| $Da$ | [-] | 0.17-0.25 (3) | 0.88-1.02 (10) | 3.25-4.19 (18) | Oldham et al., 2013 |
| *Curvature* | [-] | -0.70;-0.51 (3) | -0.25;-0.22 (9) | -0.03;-0.01 (18) | Dupas et al., 2019 |

**3 Results and Discussion**

**3.1 Model validation in the Selke River (Meisdorf)**

To evaluate the network model performance in a realistic setting, we implemented the model with a fixed parameter combination (Table C1) in the Selke catchment and aimed to capture C-Q dynamics at the basin outlet. The simulated $NO_3^-$ concentration time series for the Meisdorf station in Fig. 2a shows a seasonal pattern that follows the observation concentration

data reasonably well (Nash-Sutcliffe Efficiency; NSE = 0.50, percent bias; pbias = -0.4 %). This seasonality is also reflected in simulated daily percentage of load removed (the ratio between the daily total removed load and the daily total incoming



load in the river network); and ranges from almost 0 % to 3.4 % in this case, with the median $L_{r.perc}$ value equal to 0.41 %.
The highest removal efficiencies are simulated in fall and summer and coincide with low simulated $NO_3^-$ concentrations at the
catchment outlet. The observed nitrate concentrations generally show an enrichment export pattern in the log(C)-log(Q) space
(b = 0.40, R² = 0.56) and a *Curvature* of -0.35 which agrees well with the simulated *Curvature* of -0.28 (Fig. 2b). The observed
low nitrate concentrations coincide with low discharges in fall and summer, while high concentrations occur mainly in winter
when discharges are higher.

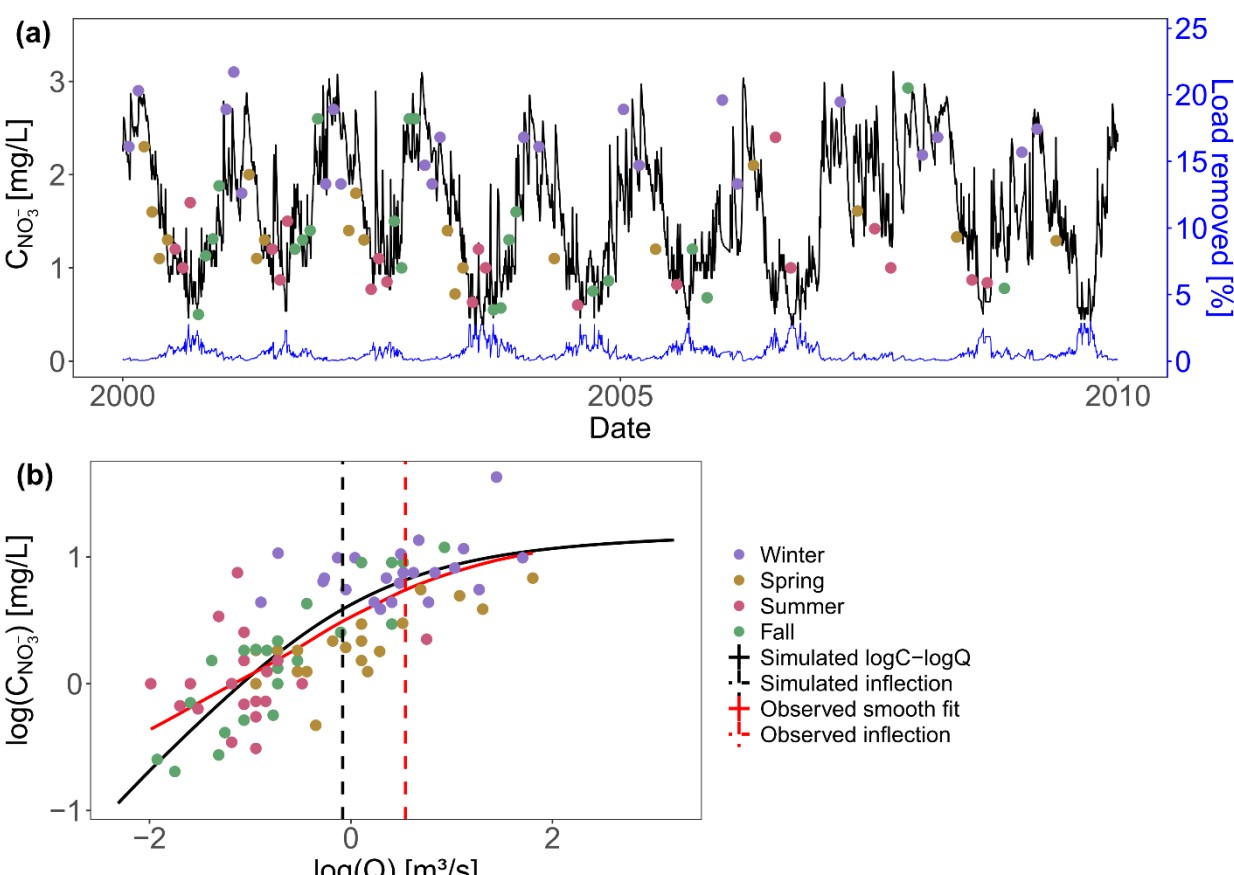

**Figure 2: (a) Simulated and observed $NO_3^-$ concentrations at the Selke Meisdorf gauging station for a 10 year simulation period**
**(2000-2010; NSE=0.50). One data point (C~5 mg L⁻¹) is not shown here. The simulated median percentage of load removed in the**
**stream network (blue line) is given during the same time period. (b) The observed $NO_3^-$ concentrations and Q are log transformed**
**and plotted together with the simulated C-Q data for 2000-2010. A smoothed spline is fitted to the observed and simulated C-Q data;**
**and *Curvatures* of -0.35 and -0.28 are calculated at the respective discharges of 1.72 m³ s⁻¹ and 0.92 m³ s⁻¹, indicating the smoothed**
**spline inflection points.**


Within the Selke Meisdorf river network the simulated *Curvature* is largely contained within -1.12 to -0.29 (10$^{th}$ and 90$^{th}$
quantiles respectively) for the given parameter combination (Table C1, Fig. 3). High *Curvatures* (< -1.12) are found





exclusively at grid cells with a low total drainage area ($A_i$ < 9 km²) and *Curvature* becomes stable with increasing drainage
areas (inset Fig. 3, Fig. B5). The incoming ($L_{in.ls}$ and $L_{in.up}$; Eq. (3)), removed ($L_r$; Eq. (5)) and outgoing absolute load ($L_i$;
Eq. (4) with L = CQ) as function of Q in the log-log space are shown in Fig. 3 for three selected grid cells on the main river
stem with low (C), intermediate (B) and high (A) drainage areas. The corresponding log(C)-log(Q) relationship for the outgoing
load ($L_i$) at the outlet (A) is presented in Fig. 2b. Note that *Curvature* is calculated from log(C)-log(Q) relationships rather
than log(L)-log(Q). The loads in grid cell A, B and C generally increase with discharge while the load removal efficiency
decreases with discharge. The highest removal efficiencies are found in the headwater grid cell C (39 % for low discharge),
followed by mid-stream grid cell B (3 % for low discharge) and the outlet A (0.5 % for low discharge). However, the total
absolute load removed ($L_r$, sum per year) is largest for the mid-stream grid cell B (66.2 kg N year$^{-1}$), followed by the headwater
cell C (25.6 kg N year$^{-1}$) and the outlet A (21.0 kg N km$^{-2}$ year$^{-1}$). Finally, the total yearly incoming load ($L_{in.ls} + L_{in.up}$)
increases in the downstream direction from 658 kg N year$^{-1}$ in the headwaters to 72716 kg N year$^{-1}$ at the basin outlet.

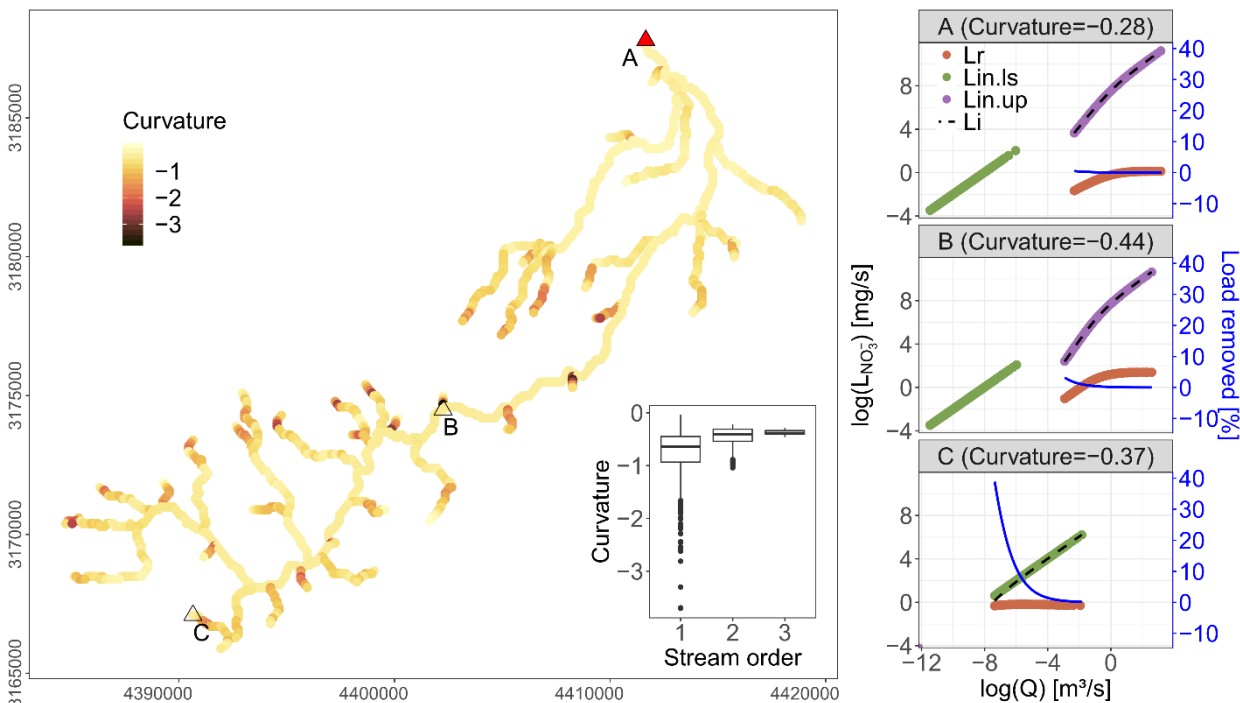

**Figure 3: Spatial distribution of simulated *Curvature* in the Selke river network (Meisdorf) for a selected parameter set (see Table
C1). Three representative grid cells covering low (A), intermediate (B) and high (C) total drainage areas show the incoming land to
stream load as L$_{in.ls}$ (Eq. (3)), the incoming load from upstream as L$_{in.up}$ (Eq. (3)), the removed load as L$_r$ (Eq. (5)) and the outgoing
load as L$_i$ (Eq. (4)) in the log(L)-log(Q) space. The load removed as a percentage of the incoming load is presented on the secondary
axis. The corresponding *Curvature*s for these grid cells are calculated from the log(C)-log(Q) relationships rather than log(L)-log(Q).**




With uniform, constant parameters the network model does not account for a spatiotemporal parameter variability. Nevertheless, it successfully (see NSE and pbias) reproduces the seasonality of the observed concentrations over the 2000-2010 period for the Selke Meisdorf catchment (Fig. 2a). For comparison, Yang et al. (2018) found a similar performance (NSE = 0.47, pbias = -3.35 %) when applying a fully distributed model with 16 calibrated parameters in this catchment between 1997 and 2009. The uptake velocity $v_f$ for our simulation was set to 0.098 m day$^{-1}$ to closely match the observed (assimilatory) uptake range of 0.009 to 0.103 m day$^{-1}$ for the Selke Meisdorf river network (Rode et al., 2016), the annual percentage load removed equals 4.7 % which is within a comparable range reported in prior studies (Rode et al., (2016) and Yang et al., (2018) found annual means of 4.8 and 7.6 % respectively). Yang et al. (2018) reported very high uptake efficiencies (up to 75 %) for summer seasons that were caused by low $NO_3^-$ concentrations (0.21 mg N L$^{-1}$) and load (L = CQ), which are not represented in our model simulation (the lowest simulated $NO_3^-$ concentration equaled 0.4 mg N L$^{-1}$). Additionally, due to the parsimonious structure of the proposed model, we did not account for the temporally changing effects of environmental factors like temperature and light availability that might (seasonally) influence uptake efficiencies in the river network. Nevertheless, these reported high low flow uptake efficiencies in summer are not a main contributor to the annual percentage load removed that is dominated by high flows, generally recorded during winter. Thus for the Monte Carlo simulations (Sect. 3.2 below) we calculated $L_{r.perc}$ as the median of the daily percentage load removed rather than the total removal efficiency for the entire simulated time period to better represent an effective long term network wide removal capacity.

The interplay of incoming, removed and outgoing load at each network grid cell shapes the log(L) and log(C)-log(Q) relationships; and thus the estimated *Curvature* at the catchment outlet (Fig. 3). Land to stream loading ($L_{in.ls}$) that varies linearly with direct incoming discharge at a given grid cell (Eq. (3) with L = CQ) in the log space (*Curvature* = 0) can lead to a bent outgoing log(C)-log(Q) relationship where concentration or load ($L_i$) varies non-linearly with discharge (*Curvature* ≠ 0). The onset of a bent log(C)-log(Q) pattern (*Curvature* = -0.37) is illustrated in the headwater grid cell C in Fig. 3 where $L_{in.ls}$ is the only incoming load (upstream incoming load, $L_{in.up}$ equals 0 in this case). The absolute removed load is higher under increasing Q while the percentage load removed is lower, which explains observed C-Q patterns with higher log(C)-log(Q) slopes for low flows than for high flows (Moatar et al., 2017; Wollheim et al., 2008; Doyle, 2005; Wollheim et al., 2017; Basu et al., 2011). This decreased $NO_3^-$ load removal efficiency at the basin outlet (spatial scale) or during events (temporal scale) can arise because stream morphology characteristics such as depth and water velocity, that correlate with varying discharge, constitute higher surface-to-volume ratios at the headwaters (generally low flows) than at the outlet (higher flows) (Peterson et al 2001; Hensley et al 2014). In the Selke Meisdorf case, uptake and land to stream loading at the downstream grid cells (B and A in Fig. 3) have a decreasing local impact on the outgoing load due to the large upstream contributions that increase in the downstream direction (see explicit scaling relationship for input flux in Bertuzzo et al., 2017). This is also explained by Wollheim et al. (2018) who suggests that the river network saturates as supply exceeds biological 'demand'. Dupas et al. (2017) on the other handshows how $NO_3^-$ uptake effects are decreasingly visible in C-Q observations



downstream and concentrations largely matched those estimated by a conservative mixing model. The saturation effect with
the accumulation of large load is reflected in the *Curvature* converging to a constant value when moving from upstream to
downstream or from a lower order to a higher order river reach (Fig. 3, B6). This also corroborates the recent findings of
Abbott et al. (2018) who found that the temporal variability (here reflected in the C-Q relationship) of nutrients is preserved
moving downstream in a river network. Overall the Selke example shows that the network model can realistically reproduce
the bending of observed $NO_3^-$ C-Q relationships that evolve from the decreasing removal efficiency at higher discharges.

### 3.2 Monte Carlo simulation results

To elucidate how *Curvature* at the catchment outlet is shaped by in-stream transport and removal, a Monte Carlo simulation
was run for the same 11107 model input parameter combinations (Table 1) in each of the 13 German catchments (Fig. 1). The
overview of the model outputs for each study catchment in Table 4 shows that catchments 1, 5 and 11 display the smallest $10^{th}$
quantile *Curvature* values of -1.61, -1.40 and -1.24 (highest bending) while the catchments 4 and 6 registered higher (lowest
bending) and less variable *Curvature* ($10^{th}$ quantiles at -0.31 and -0.35) (Fig. B7). Catchments 3, 4 and 8 are characterized by
high discharges (Table 2) at the catchment outlet and demonstrate low percentages of load removed, $L_{r.perc}$ ($90^{th}$ quantile at
29.8, 32.1 and 19.3 % respectively). The highest $L_{r.perc}$ are found in catchments 1 and 10 (98.4 and 95.1 % for the respective
$90^{th}$ quantiles). The regression slope of the log(C)-log(Q) relationship at the basin outlet, $b_{out}$, is positively skewed for all
catchments compared to the slope $b$ of the land-to stream loading function that had no positive or negative preference (Table
1, Eq. (3)) with the most positive slopes found in catchment 5. The distribution of the concentrations at the catchment outlet,
$C_{out}$, are generally similar across all catchments ($10^{th}$ and $90^{th}$ percentiles within 0 to 6.2 mg $L^{-1}$) and are significantly less
variable than the land-to-stream incoming concentration (parameter $C_{mean}$) that varied from $10^{-4}$ to 20 mg $L^{-1}$ across all the
simulations (Table 1). The highest $C_{out}$ are found in the largest catchment 8. The median water velocity $v$ (Eq. (2.3)) is
between 0.01 and 0.5 m s$^{-1}$ for the $10^{th}$ and $90^{th}$ quantiles of all the study catchments. With the largest $v$ simulated for
catchments 3 and 4 that also have the highest discharge. The median river network travel time, $TT$, for all simulations and
catchments ranges from 0.1 to 4 days between their respective $10^{th}$ and $90^{th}$ quantiles and remarkably have no clear relationship
with catchment properties as the total river network length (Table 2). Finally, the Damköhler number, $Da$ (Eq. (6)), is variable
around 1 with the highest values, indicating reaction driven conditions, found for catchments 2 and 12 that respectively range
from 0.6 to 10.3 and 0.7 to 10.8 for the $10^{th}$ and $90^{th}$ quantiles. The lowest $Da$ values are found for catchments 4 and 10 ($90^{th}$
quantile < 2) implying more transport driven conditions.

**Table 4: $10^{th}$ and $90^{th}$ quantiles of model outputs *Curvature*, percentage load removed, $L_{r.perc}$, Damköhler number $Da$, regression**
**slope of the log(C)-log(Q) relationship at the basin outlet, $b_{out}$, the median concentration at the basin outlet, $C_{out}$, the median water**
**velocity, $v$, that is calculated with the channel shape parameters $a_w$, $a_d$, $K_w$ and $K_d$ and discharge (Eq. (2.3)) and median river**
**network travel times, $TT$, for each of the 13 German catchments.**





| Catch. ID | Curvature [-] | | $L_{r.perc}$ [%] | | $Da$ [-] | | $b_{out}$ [-] | | $C_{out}$ [mg L$^{-1}$] | | $v$ [m s$^{-1}$] | | $TT$ [days] | |
|---|---|---|---|---|---|---|---|---|---|---|---|---|---|---|
| | 10th | 90th | 10th | 90th | 10th | 90th | 10th | 90th | 10th | 90th | 10th | 90th | 10th | 90th |
| 1 | -1.61 | -0.01 | 2.6 | 98.4 | 0.3 | 6.4 | -0.65 | 2.22 | <10$^{-4}$ | 4.68 | 0.01 | 0.25 | 0.1 | 1.8 |
| 2 | -1.04 | -0.01 | 0.9 | 78.5 | 0.6 | 10.3 | -0.42 | 2.31 | <10$^{-4}$ | 2.36 | 0.02 | 0.29 | 0.6 | 3.9 |
| 3 | -0.43 | -0.02 | 0.2 | 29.8 | 0.2 | 2.8 | -0.54 | 1.96 | 0.01 | 5.27 | 0.07 | 0.48 | 0.5 | 3.3 |
| 4 | -0.33 | -0.01 | 0.2 | 32.1 | 0.1 | 1.5 | -0.60 | 1.85 | 0.03 | 5.56 | 0.07 | 0.50 | 0.3 | 1.9 |
| 5 | -1.40 | -0.01 | 1.3 | 85.1 | 0.1 | 2.0 | -0.49 | 2.43 | 0.01 | 3.76 | 0.04 | 0.38 | 0.2 | 1.5 |
| 6 | -0.35 | -0.02 | 0.5 | 54.6 | 0.3 | 4.3 | -0.58 | 1.84 | 0.02 | 3.75 | 0.04 | 0.36 | 0.4 | 2.7 |
| 7 | -0.44 | -0.01 | 0.8 | 72.6 | 0.2 | 3.6 | -0.52 | 2.09 | 0.01 | 4.4 | 0.04 | 0.38 | 0.4 | 2.8 |
| 8 | -0.63 | -0.01 | 0.1 | 19.3 | 0.2 | 4.1 | -0.71 | 1.70 | 0.07 | 6.24 | 0.05 | 0.39 | 0.5 | 3.0 |
| 9 | -0.68 | -0.01 | 0.8 | 70.4 | 0.3 | 5.3 | -0.53 | 1.99 | 0.01 | 3.43 | 0.04 | 0.35 | 0.5 | 3.1 |
| 10 | -0.79 | -0.01 | 1.9 | 95.1 | 0.1 | 1.9 | -0.45 | 2.33 | 0.01 | 3.88 | 0.04 | 0.36 | 0.2 | 1.2 |
| 11 | -1.21 | -0.01 | 1.6 | 85.6 | 0.5 | 7.4 | -0.48 | 2.35 | <10$^{-4}$ | 2.41 | 0.03 | 0.32 | 0.5 | 3.5 |
| 12 | -0.97 | -0.01 | 1.5 | 83.2 | 0.7 | 10.8 | -0.49 | 2.07 | <10$^{-4}$ | 2.43 | 0.02 | 0.29 | 0.5 | 4.1 |
| 13 | -0.46 | -0.01 | 1.3 | 72.9 | 0.1 | 1.5 | -0.72 | 1.69 | 0.05 | 4.35 | 0.05 | 0.42 | 0.2 | 1.4 |

The Monte Carlo output in Table 4 shows reasonable results, as simulated *Curvatures* for all catchments and all parameter combinations (80 % of the values between -0.70 and -0.012, Table 4 and Fig. B7) are comparable with the *Curvature*s from

NO$_3^-$ log(C)-log(Q) relationships in the French catchments (80% of the values between -0.41 and -0.067, Fig. B3) (Dupas et al., 2019). Note that simulated *Curvature* can only be smaller than and equal to zero as the model takes into consideration in-stream uptake (no release) and a uniform land to stream loading function. For the model output $L_{r.perc}$, a wide range of uptake efficiencies were captured from almost 0 to near to 100 % (Mulholland et al., 2008) for some simulations and a median value of 14.4 % across simulations. This simulated range however exceeds the proposed range by Birgand et al. (2007) of 10 to 70%

of N removal for agricultural drainage networks at annual time scales. High removal percentages (median over the simulated time period of daily percentage load removed in the network exceeding 95 %) are registered for 3.4 % of all simulations while very limited load removal ($L_{r.perc}$ < 5 %) occurred for 32.1 % of all the simulations. Other simulation outputs such as the effective velocity $v$ surprisingly rendered similar distributions across the catchments (Table 4) given that the median Q varied for almost three orders of magnitude at the basin outlet (Table 2). Their specific discharges (Sect. 2.2.1) were similar and by

taking the spatiotemporal median $v$ as an effective catchment value for each simulation the (more numerous) headwater grid cells were better represented than the grid cells close to the basin outlet. A similar effect is found for the range of the effective travel time $TT$. Generally these similar $v$ and $TT$ distributions from model simulations between catchments align with the notion of Langbein and Leopold (1964) that drainage networks evolve naturally to transport water (and sediment) most efficiently such that an equilibrium between channel form and water and sediment load is imposed (Leopold and Maddock,

1953). Finally, also Damköhler numbers $Da$ exhibited realistic ranges, mostly distributed around 1 (Oldham et al., 2013;



Ocampo et al., 2006), with 36.5 % of the simulations < 0.8 and 50.8 % > 1.2 indicating that more simulations reaction driven than transport driven. Note that here $Da$ only takes into account in-stream transport and uptake.

As for the simulations, the same parameter input set is applied in each catchment, differences in model outputs between the catchments result from the combination of distinct transport and uptake processes in each river network. From Table 4 and Table 2 it is clear that differences in these three model outputs (i.e. $Curvature$, $L_{r.perc}$ and $Da$) between the catchments cannot be attributed to a single catchment property such as total network length or basin area. This could be due to a number of factors, for example $Curvature$ has the highest variability between simulations in the smallest catchment 1, compared to the other catchments, which could be due to variability in local loading and uptake patterns in the network driven by Q that are still 465 visible at the catchment outlet. Following the simulated Selke Meisdorf example in Sect. 3.1 (Fig. 3, Fig. B6), it is shown that $Curvature$ tends to converge to a constant value with increasing drainage areas (similar to Abbott et al., 2018 for nutrient concentrations, Dupas et al., 2017 for nutrient uptake and Bertuzzo et al., 2017 for DOC removal). Drainage area is however not the only catchment property influencing $Curvature$ at the outlet. For example, catchment 6 is the second largest catchment (Table 2) and has the least bent (and least variable) log(C)-log(Q) relationships. The network structure could possibly play a 470 role here as the largest catchment 8 has some large tributaries near the basin outlet (Fig. 1), which could bypass removal and transport high load during events, introducing a more variable $Curvature$ (Mineau et al., 2015; Helton et al., 2018). The percentage load removed, $L_{r.perc}$, is notably lower catchments with high Q – like 3, 4 and 8 (Table 4) which follows the narrative in Sect. 3.1 that uptake efficiency decreases with increasing Q because of increasing loads to the system (Wollheim et al., 2018; Mulholland et al., 2008) that also result in less efficient uptake within the reactive surface area (Peterson et al., 475 2001; Hensley et al., 2014). The high $L_{r.perc}$ in small catchments 1 and 10 could then be attributed to their low Q, however why the small catchment 5 does not have similar uptake performance is less clear. Similarly, differences for the third output variable, $Da$, between the catchments are hard to pin down based on certain properties. Generally the model output variability between the catchments (as a result of different catchment properties) is minor compared to the output variability within the catchments (due to the effect of the chosen input parameter set). Nevertheless catchment properties as the drainage area and 480 the network structure might influence observed $Curvature$ in a way that cannot be disentangled with our approach and should be taken into consideration when interpreting log(C)-log(Q) relationships for a given catchment.

### 3.3 $Curvature$ sensitivity analysis and model parameter correlation
The PAWN sensitivity analysis clarifies the influence of each of the independent input parameters (Table 1), a variable derived 485 from these input parameters, the median water velocity $v$ (Eq. (2.3)), and an output variable, the Damköhler number, $Da$ (Eq. (6)) on $Curvature$. The sensitivity index $KS_{max}$ in Fig. 4 and Table C2 shows that across all catchments $Curvature$ is most sensitive to the exponents in the width-Q relation $a_w$ ($KS_{max}$ = 0.62) and depth-Q relation $a_d$ ($KS_{max}$ = 0.51) with little variability between the catchments ($KS_{max}$ has a low Coefficient of Variation, CV, of 0.06 and 0.22 respectively). The slope

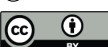



of the linear loading function, $b$, is least important in shaping *Curvature* ($KS_{max} = 0.14$) however a high variability of $KS_{max}$

is observed (CV = 0.76) that is caused by larger sensitivities for catchments 1 and 12 ($KS_{max}$ near 0.45). *Curvature* is equally

sensitive to $v_f$ and $C_{mean}$ ($KS_{max}$ 0.18 and 0.19) but $v_f$ exhibits higher variability in $KS_{max}$ than $C_{mean}$ (CV 0.59 and 0.47),

caused by catchments 12 and 1. Furthermore, over all the catchments *Curvature* is sensitive to the median velocity $v$ and the

Damköhler number $Da$ ($KS_{max}$ equals 0.64 and 0.31 respectively, CV 0.26 and 0.38). When considering the catchments

individually, basin 1 with smallest discharge has the highest median $KS_{max}$ (0.59) across all input parameters, while catchment

4 that has the highest discharge exhibits the lowest median $KS_{max}$ (0.13). Additionally, *Curvature* is very sensitive to the

velocity $v$ in catchment 1 ($KS_{max} = 0.95$), while it is least sensitive to $v$ in catchment 4. Overall, the results indicate that

*Curvature* more sensitive to the model parameters in the low Q catchment 1 compared to the large Q catchment 4, while the

other catchments show no clear order in *Curvature* sensitivity according to their catchment properties. For example in nested

catchments 10, 11 and 12 (Fig. 1), the largest catchment 12 has the highest $KS_{max}$ (0.50) and lowest CV (0.26) over all the

input parameters, indicating that here *Curvature* is more sensitive to the input parameters here than in the smaller sub-

catchments 10 and 11.

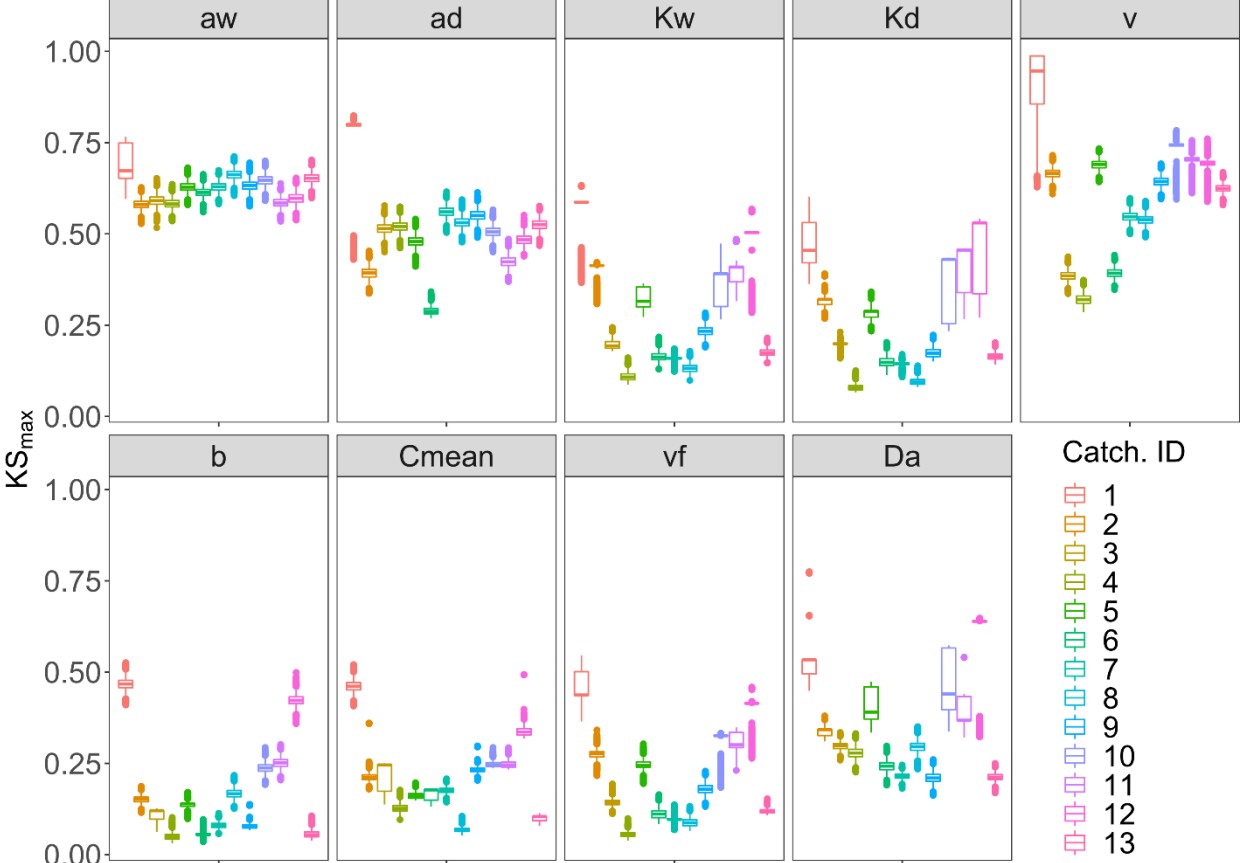





**Figure 4: The $KS_{max}$ sensitivity index for each of the model input parameters and each of the 13 simulated catchments. The input**
**parameters related to the channel geometry ($a_d$, $a_w$, $K_d$ and $K_w$), land-to-stream loading ($b$ and $C_{mean}$) and biogeochemistry ($v_f$)**
**are shown together with two variables derived from some of the input parameters: the median velocity $v$ and the Damköhler number**
**$Da$. Each boxplot displays 15000 bootstrapped estimates of $KS_{max}$ for each of the 13 simulated catchments.**

In a next step the estimated *Curvature* across all simulations is correlated to the model input parameters as well as to output
variables like the percentage load removed, $L_{r.perc}$, the log(C)-log(Q) slope at the catchment outlet, $b_{out}$, the median
concentration at the basin outlet $C_{out}$ and the uptake constant $k$ to identify the strength and direction of their relationship. The
resulting Spearman correlation matrix (Fig. 5) reflects the PAWN sensitivity findings, with the highest *Curvature* correlation
found with parameters $a_w$ ($\rho = 0.68$) and $a_d$ ($\rho = 0.56$) and input variable $v$ ($\rho = 0.57$). *Curvature* is independent of $v_f$ ($\rho = -
0.04$) but shows a negative correlation with $L_{r.perc}$ ($\rho = -0.36$), suggesting that lower *Curvature* (higher bending) is related to
a higher $L_{r.perc}$. Furthermore, *Curvature* is negatively correlated to the log(C)-log(Q) regression slope at the catchment outlet
$b_{out}$ ($\rho = -0.28$) such that higher bending coincides with more positive $b_{out}$. The variable $v$ is additionally strongly negatively
correlated with $L_{r.perc}$ ($\rho = -0.87$) so high percentage load removed occurs at low velocities. $Da$ on the other hand is positively
correlated to $L_{r.perc}$ ($\rho = 0.58$) which indicates that higher $Da$ are occurring together with higher load removed. $Da$, thereby
seems to be controlled more tightly by variation in $k^{-1}$ ($\rho = -0.71$) than by $TT$ ($\rho = 0.48$). Finally $C_{out}$ is negatively correlated
with $L_{r.perc}$ ($\rho = -0.82$) and $Da$ ($\rho = -0.61$).



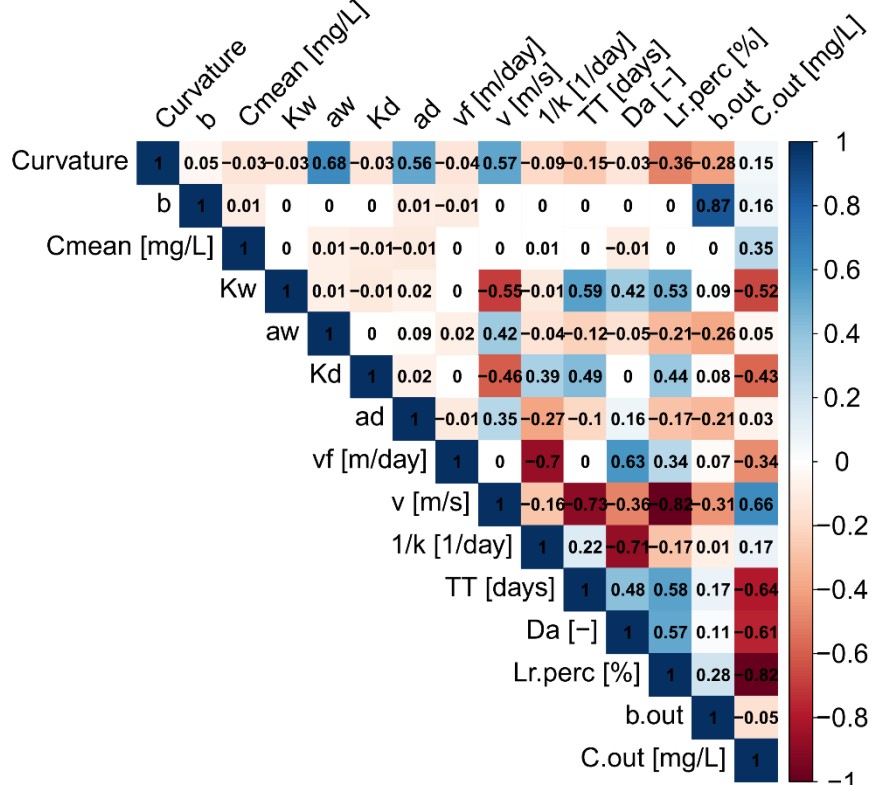


**Figure 5: Correlation matrix for the model parameter inputs: channel depth and width exponents $a_d$, $a_w$ and coefficients $K_d$, $K_w$, slope of the land-to-stream loading, $b$, concentration of the land-to-stream load $C_{mean}$, uptake velocity $v_f$ and outputs: The bending of the log(C)-log(Q) relationship at the catchment outlet, *Curvature*, effective stream velocity $v$, first order uptake constant $k$, travel time $TT$, Damköhler number $Da$, daily percentage load removed $L_{r.perc}$ and slope of the log(C)-log(Q) relationship and median**

**concentration at the outlet $b_{out}$ and $C_{out}$. The Spearman rank correlation coefficients ($\rho$) are given for each combination.**

The PAWN and correlation analysis results suggest the input parameters dictating the channel morphology $a_w$ and $a_d$ (Sect. 2.3), are controlling factors for the magnitude of the bending in log(C)-log(Q) curves at the catchment outlet. Parameters $a_w$ and $a_d$ are the exponents in the respective width-Q and depth-Q relationships (Eq. (2.1) and (2.2)) and influence the response

of the wetted perimeter ($P_i$, Eq. (A2)) in a given reach in the network and thus the reactive surface area ($P_i * l_i$) to changes in discharge. This is conceptually illustrated in Fig. B5. The correlation analysis outcomes imply that low *Curvature* (high bending) and low $a_w$ and $a_d$ occur together (Fig. 5). This is evident from the underlying model parameterizations, wherein the absolute load removed $L_{r,i}$ (Eq. (5)) is related with the width and depth exponents explicitly (Eq. (A3)) where $a_w$ and $a_d$ constitute the exponent of $Q(1 - a_w - a_d)$. When the latter term is large (small $a_w$ and $a_d$) there is a larger difference between





the effect of low and high Q's on the local absolute removed load and which can lead to a higher *Curvature* (Sect. 3.1, Fig. B5). Network based modelling studies often set the width exponent $a_w$ to a value of 0.5 that was found to be representative for rivers globally (Bertuzzo et al., 2017; Rode et al., 2016; Wollheim et al., 2018). This a-priori fixed $a_w$ may, however, strongly affect the simulated C-Q dynamics at the basin outlet as is demonstrated here. *Curvature* finally shows the lowest sensitivity to the loading parameters $b$ and $C_{mean}$ that influence the incoming load to a grid cell (Eq. (3)) and thus impact the local absolute load removed $L_{r,i}$ (Eq. (5)) rather than the percentage removed load $L_{r.perc}$. This indicates that the contribution of local incoming load in the downstream direction has a limited impact on the log(C)-log(Q) bending at the catchment outlet. Much like in the example shown for the Selke Meisdorf catchment in Sect. 3.1 where the locally contributing Q's are generally smaller (or equal for the headwaters) than the total Q in a given network reach so that the influence of the loading parameters $b$ and $C_{mean}$ on the total load decreases in more downstream reaches (Sect. 3.1, Fig. 3). Generally, for the entire river network, whether parameter $b$ has an enriching ($b > 0$), chemostatic ($b \sim 0$) or diluting ($b < 0$) character or whether the land to stream loading concentration $C_{mean}$ is small or large has little influence on the observed bending at the outlet.

Although *Curvature* only has an intermediate sensitivity to the uptake velocity $v_f$ and they don't correlate well, $v_f$ is an important 'boundary condition' for log(C)-log(Q) bending at the catchment outlet. No biological demand (low $v_f$) would mean that none of the incoming load would be removed from the river network. The outlet signal would in this case be solely driven by the discharge controlled transport processes and no bending would be observed (*Curvature* = 0). Because $v_f$ is defined as a constant within one simulation that is independent of the local nutrient concentration (Sect. 2.2.2), the percentage of load removed in the network is mainly controlled by the varying hydrological conditions here represented by the effective network wide velocity $v$ ($L_{r.perc}$ and $v$, $\rho = -0.82$). This confirms that discharge and channel morphology are among the most important predictors of removal (Alexander et al., 2000; Seitzinger et al., 2002; Wollheim et al., 2006). The role of $v$ was further examined the context of restored and channelized streams (Kunz et al., 2017) and agree with our findings that decreased $v$ influences N cycling (Peterson et al., 2001). Nevertheless, by examining a range of $v_f$ in the Monte Carlo simulations, the positive correlation between the percentage load removed and the biogeochemical uptake velocity is clear ($\rho = 0.32$).

The PAWN and correlation analysis results show that *Curvature* is sensitive to the Damköhler number $Da$ ($KS_{max} = 0.31$ Fig. 4, Table C2) that has a high positive correlation with the percentage load removed $L_{r.perc}$ ($\rho = 0.58$; Fig. 5). This indicates that high $Da$ occur concurrently with more efficient removal and is in line with others (Ocampo et al., 2006) who found sometimes almost 100 % $NO_3^-$ removal in the riparian zones of an agricultural catchment with $Da$ exceeding 2. The transport timescale $TT$ that makes up $Da$ ($\rho = 0.48$; Fig. 5) together with the inverse of the first order uptake constant $k^{-1}$ ($\rho = -0.71$; Eq. (6)) are examined for classes of low, median and high $Da$ (defined in Table 3) in Fig. 6a to disentangle which values of $k^{-1}$ and $TT$ occur together and can constitute a certain $Da$ range (each class contains 5 % of all simulations). It is shown here that low $Da$ are driven by both low $TT$ and high, variable $k^{-1}$ implying a transport driven system with limited $NO_3^-$ removal





(median $L_{r.perc}$ equals 2.4 % in Fig. 6a for low $Da$). High $Da$, contrarily, have high $TT$ and low $k^{-1}$, fostering intermediate uptake percentages (median $L_{r.perc}$ = 27.1 %). Although also $v_f$ clearly differentiates for classes of low, medium and high $Da$
in Fig. 6a, the corresponding *Curvature* values are similar in their range and mean. Nevertheless, this does not mean that $Da$ is not influencing *Curvature* at the basin outlet as there could be interactions with other inputs that are not captured here (which is supported by the PAWN findings, where $Da$ appears to be influential).

From the *Curvature* perspective (Fig. 6b) we identify model output ranges of $L_{r.perc}$, $Da$ and input variable $v_f$ that constitute
low, median and high *Curvature* classes (Table 3). High *Curvature* (~ -0.02) is thereby linked to low $L_{r.perc}$ (median 4.8 %), while low *Curvature* (~ -0.60) is connected to higher and more variable $L_{r.perc}$ (median 33.6 %), generally indicating that more bent systems are more efficient in terms of removal and vice-versa. To explore some cases when this latter statement might not be true, we examine the input parameter ranges where high bending simulations (*Curvature* < -0.51, 13.1 % of all simulations) occur concurrently with low percentage removal ($L_{r.perc}$ < 5.2 %, 0.9 % of all simulations) on the one hand and
high percentage removal ($L_{r.perc}$ > 63.0 %, 4.9 % of all simulations) on the other hand in Fig. B8a. Here, it is seen that high bending, low uptake cases mainly occur for simulations with a high effective velocity $v$ (driven by lower values for the channel shape parameters $K_w$, $K_d$, $a_w$ and $a_d$). As discussed before, low $a_w$ and $a_d$ are correlated with a high bending (low *Curvature*) and *Curvature* is most sensitive to these parameters. However, we show here that these low $a_w$ and $a_d$ do not lead to a more efficient $NO_3^-$ uptake if the other channel shape parameters cause relatively high velocities (median $v$ > 0.1 m s$^{-1}$), throughout
the network. Although the latter case is shown to be true for a minor percentage of all simulations (0.9 %), it still explains why low *Curvature* (high bending) can be connected to a wider range of $L_{r.perc}$. A similar analysis in Fig. B8b for low bending (*Curvature* > -0.03) shows that concurrent high removal simulations ($L_{r.perc}$ > 63.0 %) are even rarer (0.1 % of all simulations) compared to concurrent low removal ($L_{r.perc}$ > 5.2 %; 7.4 % of all simulations). Deviations from the expected 'high *Curvature* - low $L_{r.perc}$ pattern are also here driven by (very low) $v$. In this case however, $a_w$ and $a_d$ are generally high in both cases
(leading to high *Curvature*) and the different $v$ stem from coefficients $K_w$ and $K_d$ that are higher in high removal simulations. Finally, Fig. 6d illustrates that low medium and high uptake velocities $v_f$ lead to distinct $Da$ and $L_{r.perc}$ but do not show up in the bent signal at the catchment outlet.



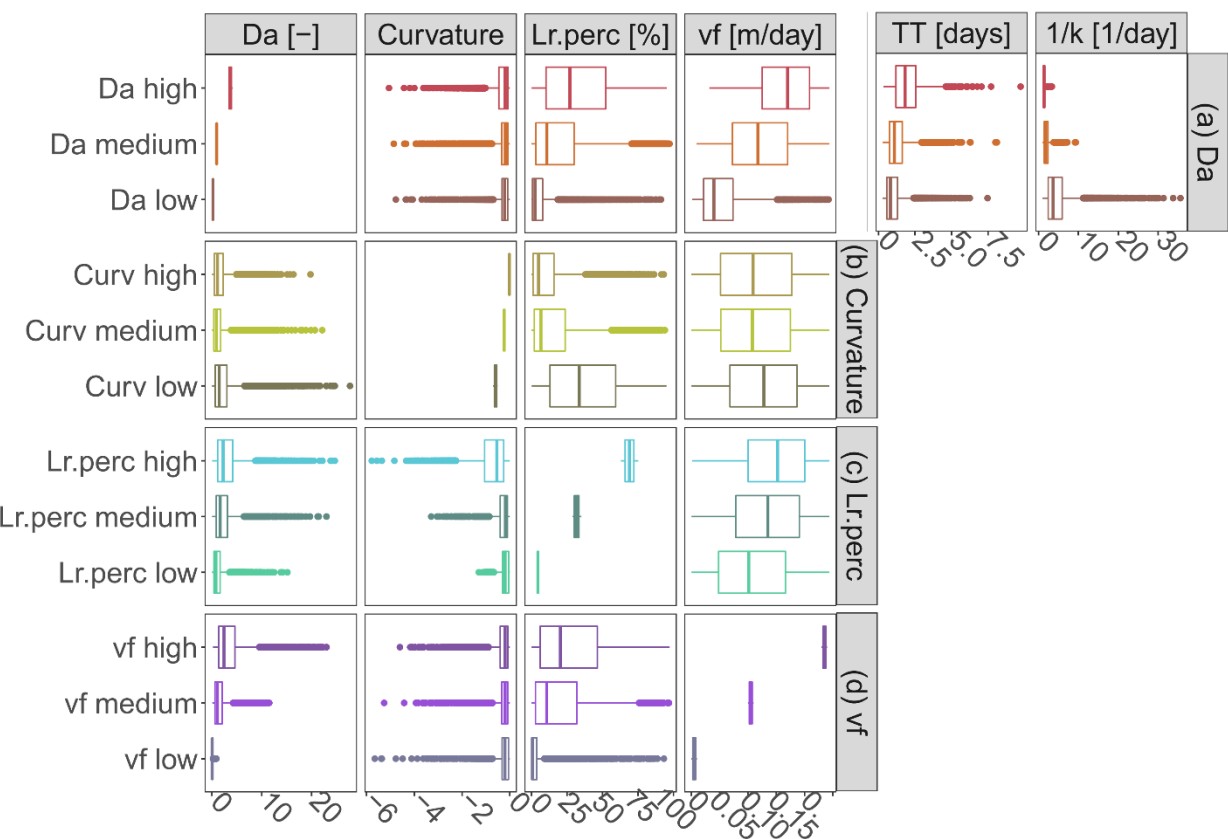

**Figure 6: The corresponding simulated ranges for high, median and low values (Table 3) of the main simulation outputs: (a) Damköhler number $Da$, (b) $Curvature$, (c) Percentage load removed $L_{r.perc}$ and (d) uptake velocity $v_f$ are shown for the same variables (in the columns). The median travel time, $TT$, and the inverse of the first order uptake constant, $k^{-1}$ are given additionally for low medium and high $Da$.**

### 3.4 Predicting in-stream processing with *Curvature*

To determine if observed C-Q bending at the catchment outlet (here *Curvature*) can be utilized to quantify in-stream uptake in the upstream river network and to visualize model parameter interactions, a classification tree was established for low, medium and high values (Table 3, Fig. 6) of the response variables $L_{r.perc}$, $Da$ and $v_f$ (Fig. 7a, b and c respectively). The prediction accuracy metrics in Table C3 and the probability histograms in Fig. 7 show that $L_{r.perc}$ can be predicted relatively well (overall accuracy of 0.66) compared to the other response variables $Da$ (accuracy 0.51) and $v_f$ (accuracy 0.40). The fitted CART models all perform significantly better than a random allocation of simulation results to each class for each response variable (Accuracy > No Information Rate, p-value < 2.2e-16). While the classes for $L_{r.perc}$ and $v_f$ are partitioned using only the network effective velocity $v$ and *Curvature*, predicting $Da$ in our case requires information on the channel geomorphology parameters the width coefficient $K_w$ and the depth exponent $a_d$. The histograms for each of the response variables in Fig. 7





indicate the probability of a test sample to be of a certain class when following the partition rules in the respective decision

tree. For example, for $L_{r.perc}$ the probability that the daily percentage load removed is small (around 8%) exceeds 0.95 when

the effective velocity $v$ in the catchment is larger than 0.22 m s$^{-1}$; while the probability that $L_{r.perc}$ is high (around 70%) in

this case is close to 0 (Fig. 7a). For $v_f$ the lowermost (1) and highest (20) classes are predicted most accurately (0.58 and 0.56

respectively, Table C3) and indicate that when the velocity is not very small and *Curvature* is smaller than -0.51 (more bent),

$v_f$ is most likely high (probability 0.59). For $Da$, the lower and higher classes can be predicted most accurately (0.69 and 0.68

respectively), for example, $Da$ is small with a probability of 0.58 when $K_w$ is relatively low ($< 6.8$). When on the other hand

$K_w$ exceeds 6.8 and $a_d$ is larger than 0.4 or when $a_d$ is smaller than 0.4 but *Curvature* is smaller than -0.45 and $v$ is very small

($< 0.04$ m s$^{-1}$) it is most likely that $Da$ is large.

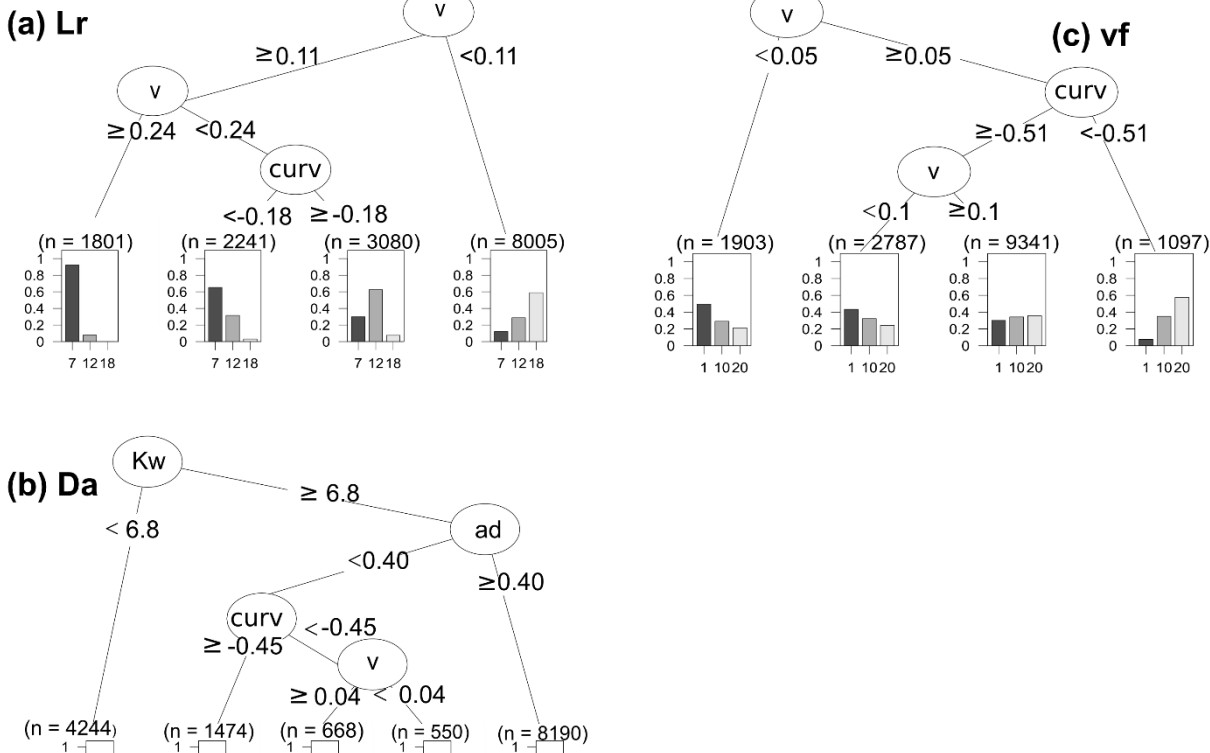

**Figure 7: CART decision tree for the response variables $L_{r.perc}$ (accuracy = 0.66), $Da$ (accuracy = 0.51) and $v_f$ (accuracy = 0.40). The classes of each response variable are defined in Table 3 and the prediction metrics for these classes are stated in Table C3. The histograms illustrate the probability of a test sample to be from a certain class when following the binary splits of the decision tree.**





These findings demonstrate that log(C)-log(Q) bending at the catchment outlet, together with the median velocity and the
response of the width and the depth of a channel to discharge (parameters $K_w$, $K_d$, $a_w$ and $a_d$) can help to classify the in-stream
daily percentage load removed $L_{r.perc}$, the Damköhler number $Da$ and to a certain extent the uptake velocity $v_f$. The velocity
may be concluded from the channel shape, discharge (Eq. (2.3)) and the topography with the channel shape parameters
sometimes available from rating curve information or detectable from high resolution satellite pictures. The CART models
could help obtain an initial probability of $NO_3^-$ removal efficiency in a river network, especially in a context where network
wide uptake measurements are scarce (Wollheim et al., 2017; Hensley et al., 2014) and physical, fully distributed models are
not always feasible to apply (Boyer et al., 2006; Klemes, 1986). Although the CART models are developed using 'only' the
13 German catchments included in the Monte Carlo analysis, in Sect. 3.2 and Table 4 we shown that the output variability
between the catchments (as a result of different catchment properties) is minor compared to the output variability within the
catchments (due to the effect of the input parameter set). Nevertheless, the prediction performance of these CART models
might be influenced in unknown ways when applied to catchments with dissimilar catchment sizes, network structures or
hydrological regimes.

## 4 Conclusions

In this study, we explore how $NO_3^-$ log(C)-log(Q) relationships, observed at a basin outlet, can display bending as a result of
network scale in-stream uptake processes. We established a parsimonious grid based river network model for 13 distinct
German catchments and investigated the influence of in-stream loading, transport and uptake parameters on the bending of
log(C)-log(Q) relationships. Based on our exploratory analysis we conclude that:

- Noisy, multi-annual and low frequency $NO_3^-$ log(C)-log(Q) relationships at a basin outlet can be described as bent
  and this bending can be robustly quantified with the new *Curvature* metric. *Curvature* tends to converge with
increasing drainages areas and is temporally stable on multi-annual time scales.

- A bent log(C)-log(Q) relationship (*Curvature* < 0) at the basin outlet can arise from log-log linear land to stream C-
  Q relationships and uptake within the river network. This supports the hypothesis that more positive slopes under low
  flow (bended log(C)-log(Q) curves) are linked to biological $NO_3^-$ concentration mediation in the stream (Moatar et
  al., 2017); and connects *Curvature* (as a quantitative measure) to observations of increased removal efficiency under
low flows (Wollheim et al., 2017).

- The bending at the catchment outlet is primarily shaped by the channel geomorphological parameters, $a_w$ and $a_d$
  (exponents in the respective stream width and depth to discharge relationships; with *Curvature* sensitivity indices
  $KS_{max}$ equal to 0.62 and 0.51; and Spearman correlation coefficient, $\rho$, equaling 0.68 and 0.56 respectively) and less
  by the uptake velocity $v_f$ ($KS_{max} = 0.18$, $\rho = -0.04$), given that $v_f$ differs from zero. In that case *Curvature* would
equal zero and the log(C)-log(Q) relationship would be solely shaped by the accumulation of upstream load. Thus,
  the change of reactive channel bed area with discharge (mediated by $a_w$ and $a_d$) has a greater influence on the bending





at the outlet than the biological removal capacity (here $v_f$). Additionally we demonstrate that an a-priori fixed $a_w$ might strongly affect the simulated C-Q dynamics at the basin outlet.

- *Curvature* at the basin outlet can be linked to the network-wide removal efficiency $L_{r.perc}$ ($\rho = -0.36$), indicating that
systems with more bending in their log(C)-logQ-relationship are more efficient in terms of removal and vice-versa. It is, however, clear that also cases with high bending (*Curvature* < -0.51) and low removal ($L_{r.perc}$ < 5.2 %, 0.9 % of all simulations) or low bending (*Curvature* > -0.03) with high removal ($L_{r.perc}$ > 63.0 %, 0.1 % of all simulations) exist that are imposed by respective higher and lower network wide median velocities. This shows how the velocity, $v$, (calculated from the channel shape parameters $a_w$, $a_d$, $K_w$ and $K_d$) may mediate the connection between $L_{r.perc}$
and *Curvature* and stresses that $v$ should be considered when interpreting log(C)-log(Q) bending.

- Simple classification trees - like CART - can be useful for predicting low, median and high classes of response variables $L_{r.perc}$, the Damköhler number $Da$ and $v_f$. They provide useful insights on how catchments with low frequency concentration and discharge time series (that are generally available) can reveal information on the upstream river network uptake performance.

To evaluate the generality of the results presented here, *Curvature* should be calculated for $NO_3^-$ concentration observations of a larger range of catchments and linked to the respective catchment properties. Properties such as light and stream ecological state can serve as proxies for uptake performance and for example topographic gradient can be a proxy for network transport velocity. Such a data-driven exploration would further elucidate the linkages between nutrient uptake efficiency and low-frequency C and Q observations.






**Appendix A**

c calculation (Jawitz and Mitchell, 2011)

$$c = e^{\mu_c - b*\mu_q}$$ (Eq. A1)

*With*

$$\mu_q = mean(\log Q_{t.sp} * a_i)$$

$$\mu_c = \log meanC - \frac{\sigma_c{}^2}{2}$$

$$\sigma_c = \sqrt{b^2 * \sigma_q{}^2}$$

$$\sigma_q = \sqrt{var(\log Q_{t.sp} * a_i)}$$


Stream channel wetted perimeter $P_i$ [L], where A is the cross-sectional area [L²], $R_H$ [L] is the hydraulic radius and $w_i$ [L], $d_i$ [L] and $v_i$ [L/T] are the local stream width, average depth and velocity respectively. $S_i$ [L/L] is the stream bed slope and n [-] is the Manning roughness coefficient that is equal to 0.03 for all simulations.

$$P_i = \frac{A}{R_H} = \frac{w_i*d_i*S^{\frac{3}{4}}{}_i}{(v_i*n)^{\frac{3}{2}}}$$ (Eq. A2)


The load removed in a grid cell (Eq. (5)) with the width and depth exponents, $a_w$ and $a_d$, stated explicitly.

$$L_{r,i} = L_{in,i} * (1 - e^{-\frac{v_f*(K_w*K_d)^{\frac{5}{2}}*S^{\frac{3}{4}}*n^{\frac{3}{2}}}{Q_i^{1-a_w-a_d}}})$$ (Eq. A3)

The velocity in a grid cell (Eq. (2.3)) with the width and depth exponents, $a_w$ and $a_d$, stated explicitly.

$$v_i = \frac{Q^{1-a_w-a_d}}{K_w*K_d}$$ (Eq. A4)





**Appendix B**

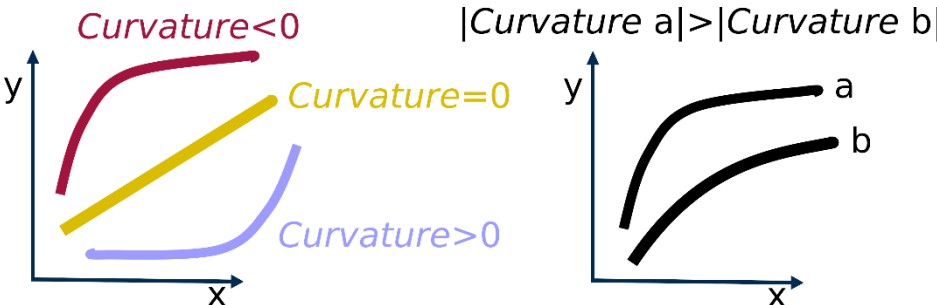

**Figure B1: Conceptual figure explaining *Curvature*. *Curvature* is calculated as the largest instantaneous rate of change of direction of a point that moves on a curve. When *Curvature* < 0 the curve is concave, for *Curvature* > 0 the curve is convex.**

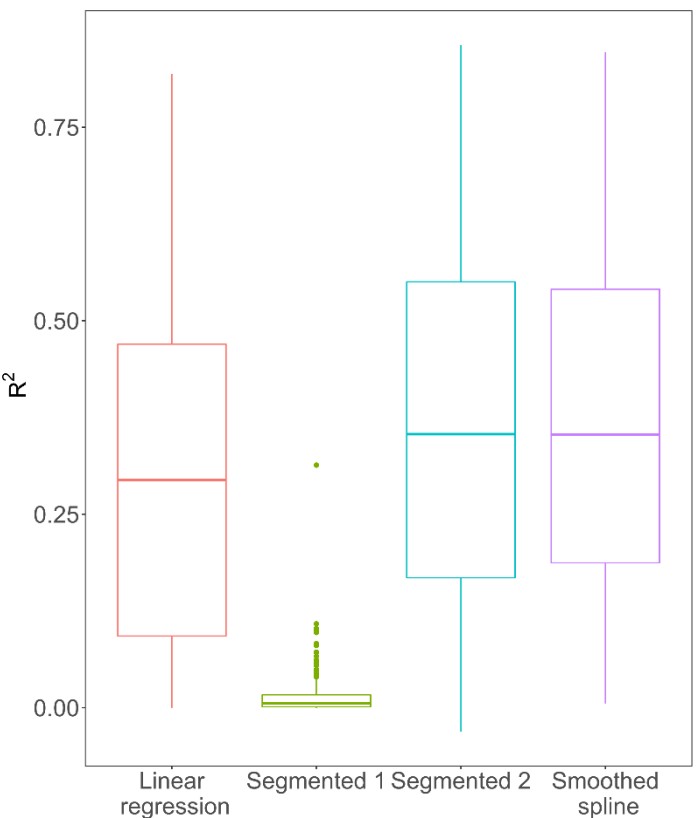

**Figure B2: $R^2$ for four models fit to log(C)-log(Q) data of 444 French stations (Dupas et al., 2019). The smoothed spline method used for calculating curvature is compared to a simple linear regression fit, a segmented linear regression (Segmented 1) with a fixed breakpoint at the median Q (Meybeck and Moatar, 2012) and a segmented regression without a fixed breakpoint as described in**
**Marinos et al. (2020).**



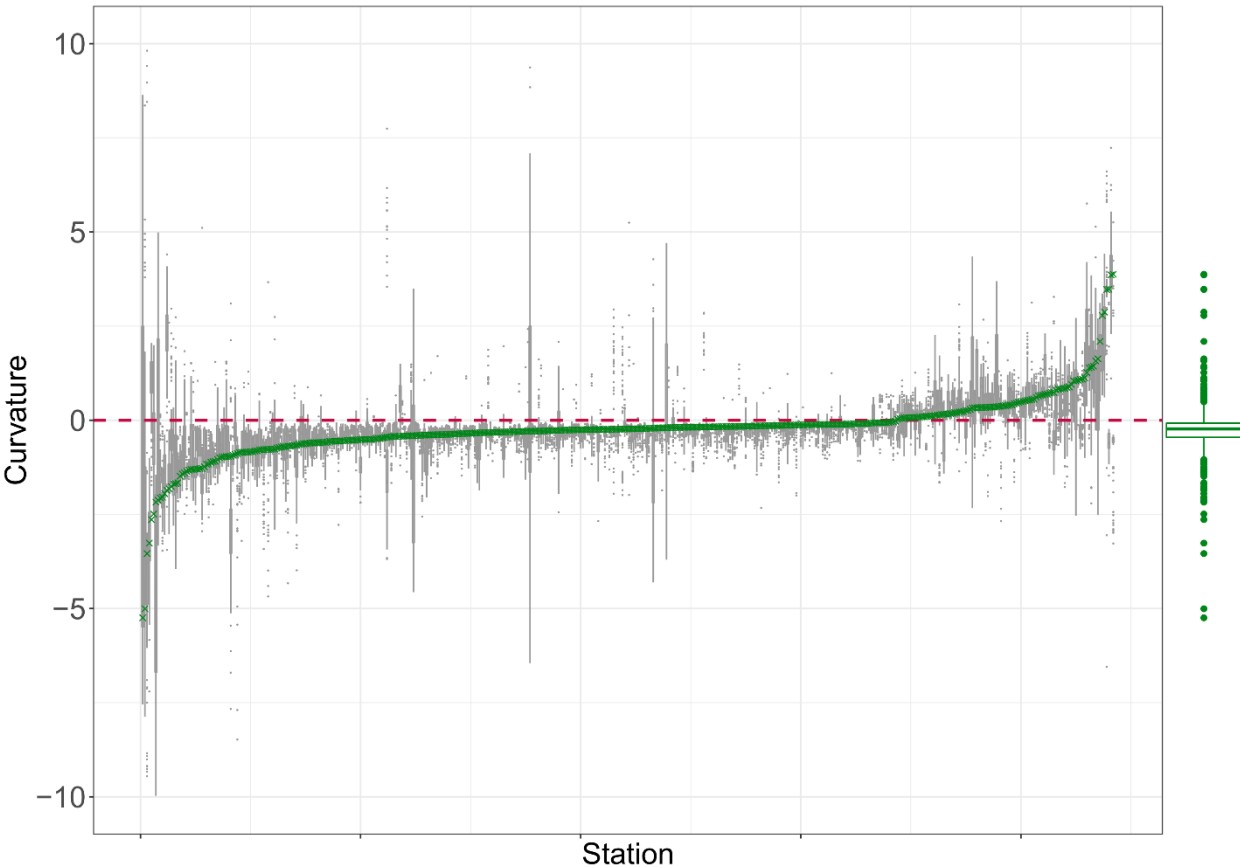

**Figure B3:** *Curvature* **of nitrate log(C)-log(Q) data for 442 French monitoring stations arranged from left to right with increasing** *Curvature* **(green crosses). For a given station, the grey boxplot represents the temporal robustness of this metric by subsampling 100 times from the original time series. The green boxplot indicates the range and distribution of all observed station** *Curvature* **values.**





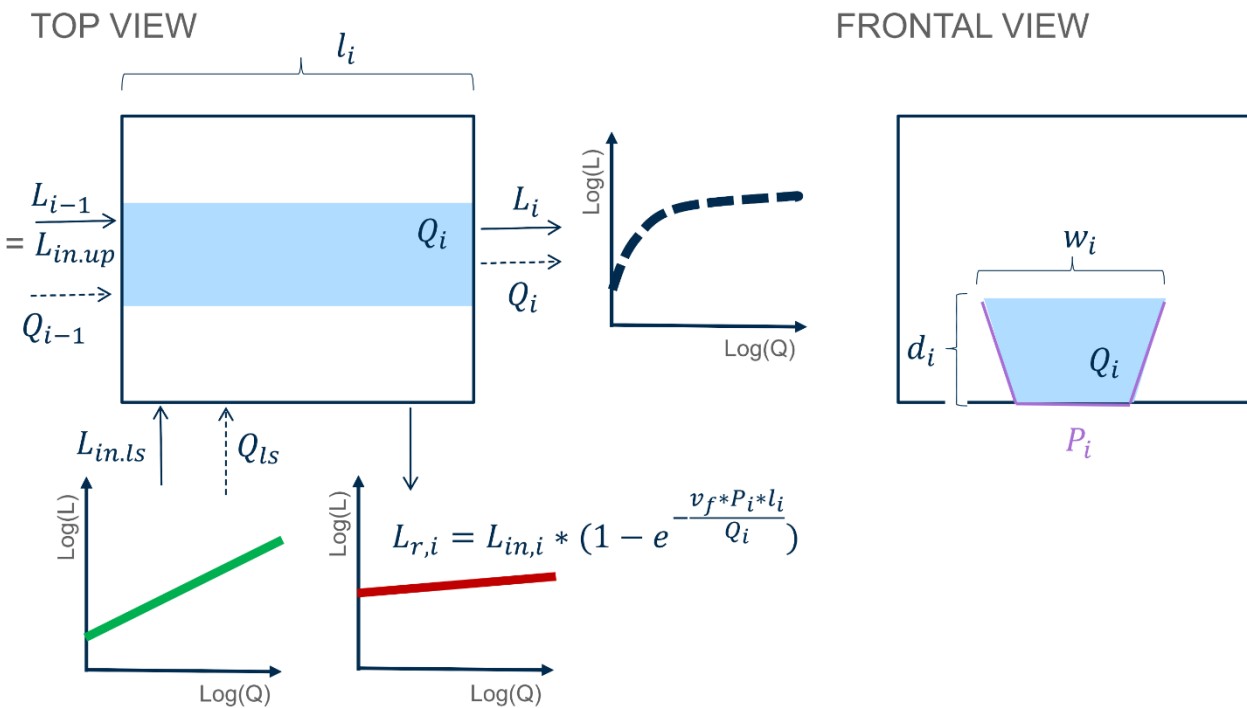

**Figure B4: Network model, illustrated for one grid cell.**





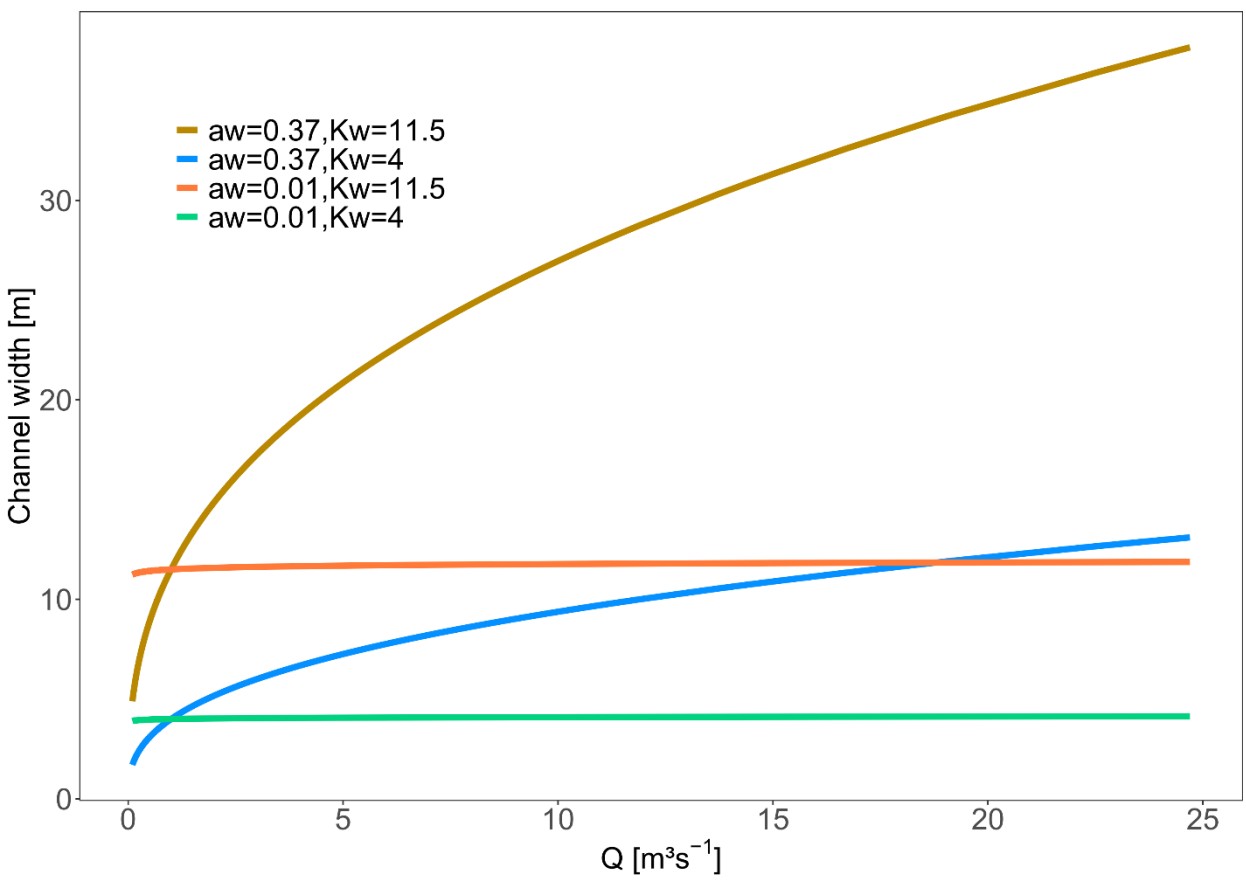

**Figure B5: The effect of parameters $a_w$ and $K_w$ on the channel width illustrated for the Q timeseries at the Selke Meisdorf station.**




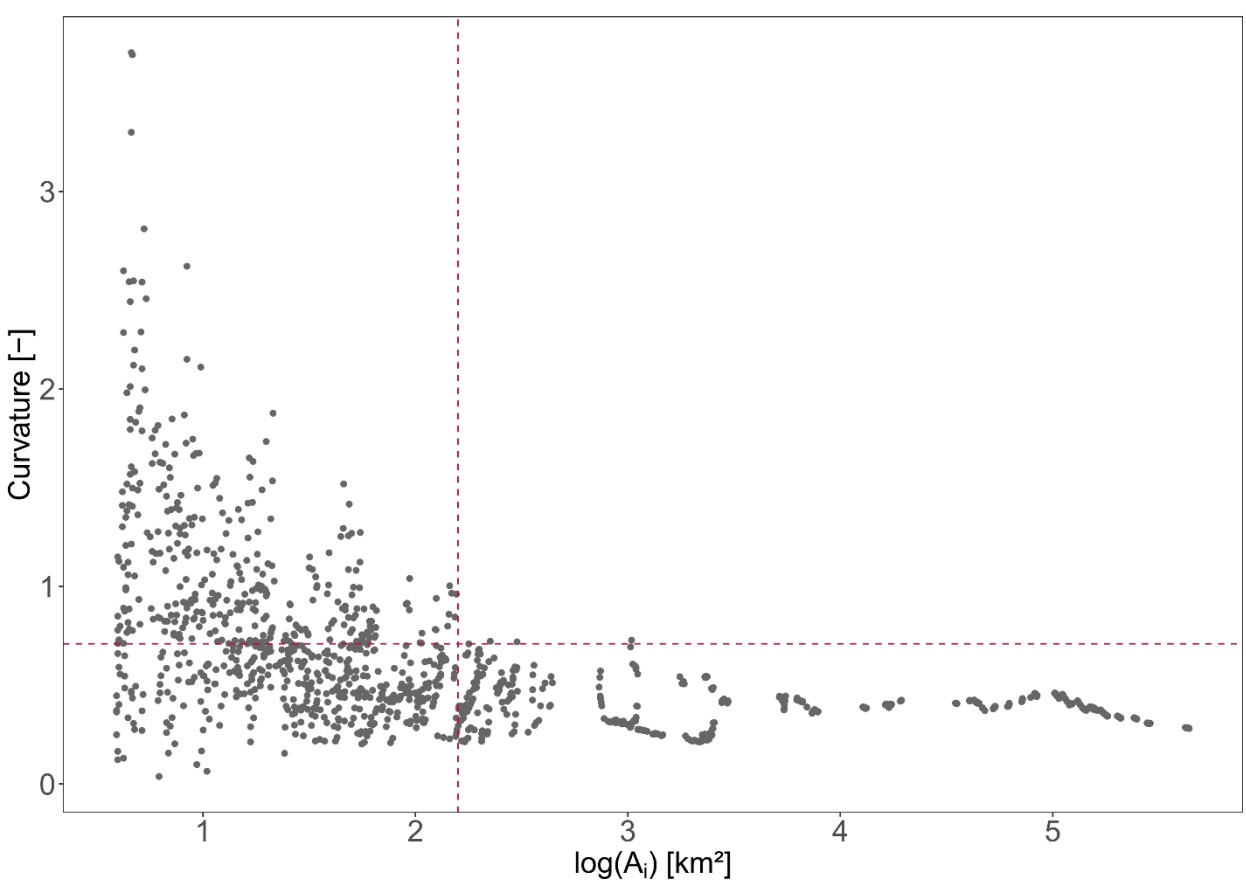

**Figure B6:** *Curvature* **as a function of total drainage area** $A_i$ **(Eq. (2)) for each gridcell of the Selke Meisdorf validation run with one uniform and constant parameter set (Table C1).**




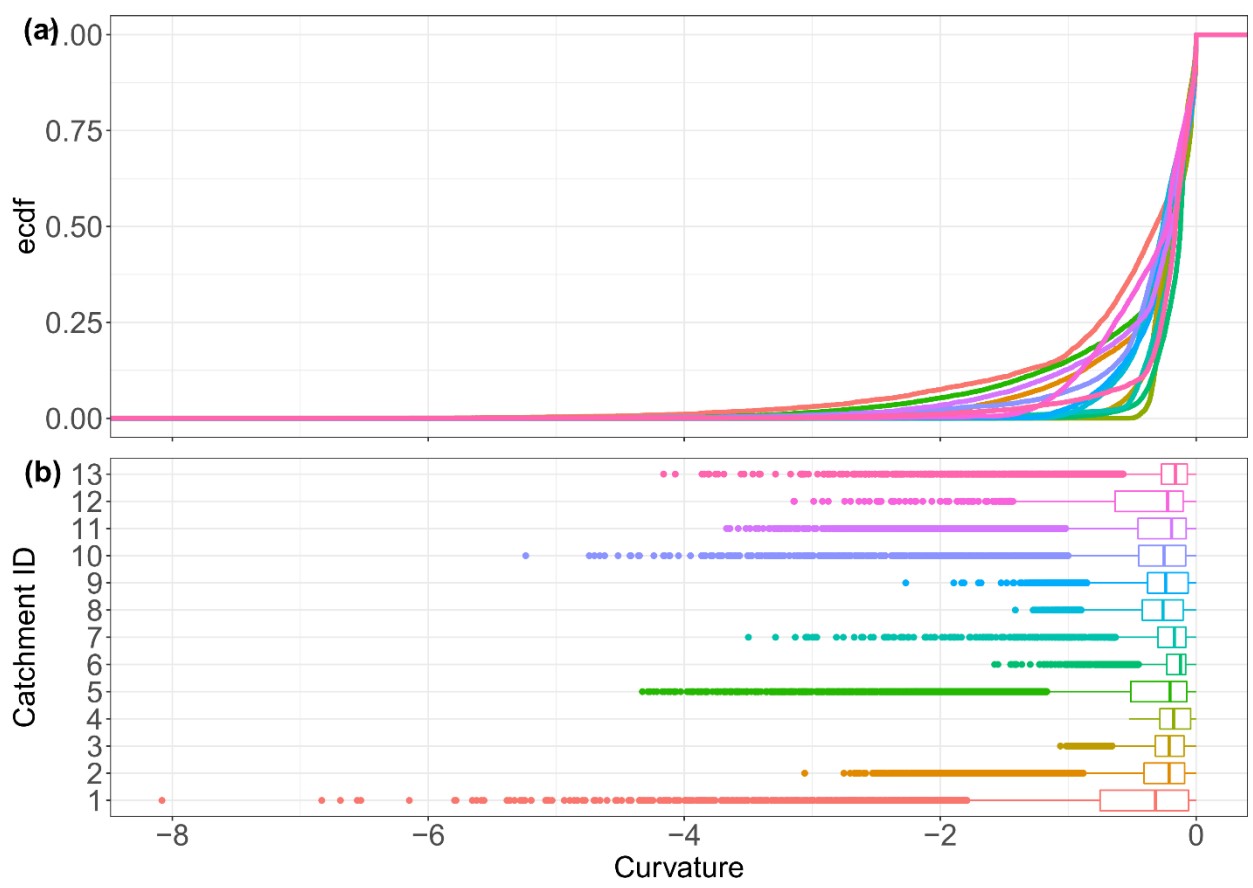

**Figure B7:** *Curvature* **distributions resulting from running the same 11107 input parameter combinations in each of the 13 catchments (a) shows the elemental cumulative distribution and (b) boxplots. None of the catchments have a normally distributed** *Curvature* **set according to the Kruskal-willis (p<0.05) test.**





Figure B8: Two specific cases of Fig. 6, (a) when *Curvature* is low (high bending) and $L_{r.perc}$ is low opposed to high and (b) when *Curvature* is high (low bending) and $L_{r.perc}$ is high opposed to low.





## Appendix C

**Table C1: Validation range for all parameters**

| Parameter | Validation Value Selke |
|-----------|------------------------|
| $v_f$ | 0.098 |
| $b$ | 0.014 |
| $C_{mean}$ | 3.014 |
| $K_w$ | 2.75 |
| $a_w$ | 0.09 |
| $K_d$ | 0.17 |
| $a_d$ | 0.49 |

**Table C2: PAWN sensitivity indices $KS_{max}$ for all the parameters and all the catchments, together with median and coefficients of variation (CV).**

| Parameter | Catchment ID | | | | | | | | | | | | | Median | CV |
|-----------|------|------|------|------|------|------|------|------|------|------|------|------|------|--------|------|
| | 1 | 2 | 3 | 4 | 5 | 6 | 7 | 8 | 9 | 10 | 11 | 12 | 13 | | |
| $v_f$ | 0.44 | 0.28 | 0.14 | 0.06 | 0.24 | 0.11 | 0.10 | 0.09 | 0.18 | 0.33 | 0.30 | 0.41 | 0.12 | 0.18 | 0.59 |
| $b$ | 0.47 | 0.15 | 0.12 | 0.05 | 0.14 | 0.05 | 0.08 | 0.17 | 0.08 | 0.24 | 0.25 | 0.42 | 0.05 | 0.14 | 0.76 |
| $C_{mean}$ | 0.46 | 0.21 | 0.25 | 0.13 | 0.16 | 0.18 | 0.18 | 0.07 | 0.23 | 0.25 | 0.25 | 0.34 | 0.10 | 0.19 | 0.47 |
| $K_w$ | 0.59 | 0.41 | 0.19 | 0.11 | 0.32 | 0.16 | 0.16 | 0.13 | 0.23 | 0.39 | 0.41 | 0.50 | 0.17 | 0.23 | 0.51 |
| $a_w$ | 0.75 | 0.58 | 0.59 | 0.58 | 0.63 | 0.61 | 0.63 | 0.66 | 0.63 | 0.65 | 0.58 | 0.60 | 0.65 | 0.62 | 0.06 |
| $K_d$ | 0.45 | 0.32 | 0.20 | 0.08 | 0.29 | 0.15 | 0.14 | 0.09 | 0.17 | 0.46 | 0.43 | 0.53 | 0.16 | 0.20 | 0.55 |
| $a_d$ | 0.80 | 0.39 | 0.52 | 0.52 | 0.48 | 0.29 | 0.56 | 0.53 | 0.55 | 0.51 | 0.42 | 0.48 | 0.53 | 0.51 | 0.22 |
| **Median** | 0.59 | 0.39 | 0.25 | 0.13 | 0.33 | 0.18 | 0.18 | 0.17 | 0.24 | 0.43 | 0.42 | 0.50 | 0.18 | | |
| **CV** | 0.31 | 0.46 | 0.53 | 0.81 | 0.53 | 0.65 | 0.67 | 0.76 | 0.58 | 0.45 | 0.41 | 0.26 | 0.74 | | |
| **Variable** | | | | | | | | | | | | | | | |
| $Da$ | 0.53 | 0.34 | 0.30 | 0.28 | 0.39 | 0.24 | 0.22 | 0.30 | 0.21 | 0.44 | 0.37 | 0.64 | 0.21 | 0.31 | 0.38 |
| $v$ | 0.95 | 0.67 | 0.39 | 0.32 | 0.69 | 0.39 | 0.55 | 0.54 | 0.64 | 0.74 | 0.70 | 0.69 | 0.62 | 0.64 | 0.26 |

**Table C3: Performance statistics for each of the classes for the variables $L_{r.perc}$, $Da$ and $v_f$ predicted by CART.**

| | $L_{r.perc}$ | | | $v_f$ | | | $Da$ | | |
|---|---------|----------|----------|--------|----------|----------|--------|----------|----------|
| | class 7 | class 12 | class 18 | class 1 | class 10 | class 20 | class 3 | class 10 | class 18 |
| **Sensitivity** | 0.63 | 0.38 | 0.94 | 0.43 | 0.00 | 0.77 | 0.63 | 0.07 | 0.82 |
| **Specificity** | 0.91 | 0.89 | 0.69 | 0.74 | 1.00 | 0.36 | 0.75 | 0.97 | 0.54 |
| **Pos Pred Value** | 0.77 | 0.62 | 0.61 | 0.45 | NA | 0.38 | 0.56 | 0.54 | 0.47 |
| **Neg Pred Value** | 0.83 | 0.75 | 0.96 | 0.72 | 0.67 | 0.76 | 0.80 | 0.68 | 0.86 |
| **Prevalence** | 0.34 | 0.32 | 0.34 | 0.34 | 0.33 | 0.33 | 0.34 | 0.33 | 0.33 |
| **Detection Rate** | 0.21 | 0.12 | 0.32 | 0.14 | 0.00 | 0.26 | 0.21 | 0.02 | 0.27 |



| Detection Prevalence | 0.28 | 0.12 | 0.32 | 0.32 | 0.00 | 0.68 | 0.38 | 0.04 | 0.58 |
|---|---|---|---|---|---|---|---|---|---|
| **Balanced Accuracy** | 0.77 | 0.63 | 0.82 | 0.58 | 0.50 | 0.56 | 0.69 | 0.52 | 0.68 |

**Acknowledgements** This research was supported by funding from the InStream cohort and discussions with the participating scientists. Support to FS was provided by the Reduced Complexity Models project co-funded by the Helmholtz Association;

and FS and RK acknowledge the Advanced Earth Modelling Capacity (ESM) project funded by the Helmholtz Association. We further thank Remi Dupas for providing French nitrate CQ data and Pia Ebeling for the German catchments characteristics data.

**Author contributions** JD and AM designed the study together with RK. JD developed the model code, carried out the
simulations and interpreted them. FS provided help with the PAWN method. JD and AM prepared the manuscript draft and all co-authors contributed to reviewing and editing the manuscript.

**Competing interests** The authors declare that they have no conflict of interest.

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
