# Peer review of "Bending of the concentration discharge relationship can inform about in-stream nitrate removal"

_Hydrology and Earth System Sciences, 2021_

## Author Comment (AC1)

**Response to Reviewer 1**

We would like to thank reviewer 1 for their comments on the paper 'Bending of the concentration discharge relationship can inform about in-stream nitrate removal'. We address all the reviewer comments (in italic) one by one below with responses in normal font.

*R1.1: The paper aims to explain evolution of concentration-discharge patterns in a stream network using a modelling approach. This is an interesting topic, following observations from many systems, where the c-q patterns become homogenised downstream, i.e. from highly variable and positive c-q slopes in first order streams to more linear responses, near chemostatic responses in downstreams.*

*The modelling approach adopted here explains one aspect of these previous observations, i.e. changes in curvature. The authors show that in 1st order streams curvature is larger than in higher order streams which can be explained by hydrological accumulation and homogenisation when moving downstream. Simply speaking, 1st order streams can have a larger variation in concentration sources compared to bigger streams. And/or activation/deactivation of these sources requires changes in flow discharge that can result in the 'bent' c-q relationship or simply speaking different slopes of the relationship for different flows.*

We thank the reviewer for this comment and we generally agree here. The main aim of the paper is to examine if network scale nitrate uptake effects can be inferred from the bending of low frequency; multi-annual concentration (C) and discharge (Q) observations. Thereto we apply a parsimonious river network model (similar to Bertuzzo et al., 2017; Helton et al., 2018; Helton et al., 2010; Mulholland et al., 2008) in 13 German catchments to explore the catchment scale transport and uptake processes that influence downstream log(C)-log(Q) patterns (l.103-107).

We show that *Curvature* converges when moving from lower order to higher order streams in Fig. 3 where the spatial distribution of simulated Curvature in the Selke river network (Meisdorf) for a selected parameter set is shown (l.365-366). In this case, uptake and land to stream loading at the downstream grid cells have a decreasing local impact on the outgoing load due to the relatively larger upstream contributions that increase in the downstream direction (l.400-410). The convergence of *Curvature* in higher order streams is also shown in the results of the Monte Carlo simulations, where >10 000 parameter sets were applied to 13 German catchments (l.460-482). Here, simulated *Curvature* in lower order catchments 1, 5 and 11 has a higher variance and an overall lower mean value (higher bending) than the simulated *Curvature* in higher order catchments 4 and 6 (Table 4, l.414-417) . It is however

clear that these differences between the catchments cannot be attributed to a single catchment property such as total network length or basin area (l.461-462)

*R1.2: The paper is heavy on modelling that can conceal the main findings of the paper, which is that 1) curvature is predominant in 1st order streams and 2) curvature is better explained by flow characteristics than nitrate uptake velocity. These findings can be explained by higher rate of biological processes in headwater streams (hypothesis investigated in this paper) but there are also other factors that can explain bending of the c-q curves, like stream morphology and flow-stage relationships, activation/deactivation of sources in relation to flow including presence of sewage pollution, drains etc. Thus, I am not convinced that the paper provides the one and only explanation for the observed patterns, rather than provides a plausible explanation for one of the possible explanations. This should be clearly communicated in the paper.*

In this paper, we indeed investigate a possible explanation for bent log(C)-log(Q) relationships, namely that log(C)-log(Q) bending can inform about in-stream $NO_3^-$ removal. We don't aim to state that in-stream uptake is the only explanation for C-Q bending but rather that it is one possible explanation that is motivated from previous observation- and model driven studies (Moatar et al., 2017; Hall et al., 2009; Hensley et al., 2014; Wollheim et al., 2017, l.90-99). Note that flow stage relationships (and stream geomorphology in general) were accounted for in our modelling setup (l.176-177).  We will make the point that we aim to investigate only one of the possible explanations for C-Q bending more explicit and clear in the revised version of the manuscript by adjusting the introduction, assumptions in the methods and conclusions. For example, we would edit line 313 to read: "Finally, we aim to determine if, **within this modelling framework**, C-Q bending at the catchment outlet (specifically *Curvature*) informs about the network wide in-stream uptake."

*R1.3: The modelling focus of the paper is, however, very dense and takes precedence over the problem – variations to c-q patterns and their controls. I would suggest 'moving' the modelling to the background of the paper and focusing more on the problem. This refocusing would make the paper easier to understand to a non-modelling reader and set it better in the previous research on the topic.*

We appreciate the reviewer's suggestion - however we would like to keep the modeling part entact, without losing the main focus of the paper. In this respect we believe that a thorough description of the Monte Carlo method and the uptake model are essential for the interpretation of the results as our study of the c-q patterns and their controls is modelling-based. However, we will revise the manuscript,

and in particular the conclusion section, to ensure that the key findings can be understood by all the readers. We refer to comment WW.11 for the specific changes suggested to improve the conclusion.

*R1.4: Finally, the original concentration and flow data disappear in the paper convoluted in different models. E.g. linking curvature to other models with inherent uncertainty like Damköhler number or uptake velocity. Showing more raw data in the paper, e.g. providing traditional quantifications of the c-q slopes would be very useful for the reader to link their knowledge of the subject with the new findings of this paper. Also when showing variation in curvature, I would like to know how frequent are concave vs. convex shapes.*

Thank you again for your detailed suggestions. Our analysis relies on 'raw data'/ observations in the following instances; i) low frequency C and Q data for 444 French catchments that is used to validate the *Curvature* metric; ii) measured $NO_3^-$ concentrations in the Selke Meisdorf station to validate the conceptual network model structure by comparing modelled observed $NO_3^-$ at the network outlet (Fig. 2a and b); iii) measured daily Q time series for 13 German catchments (l.253) that were used as a direct input for the explorative model to simulate $NO_3^-$ concentrations (and *Curvature*) for a range of parameter combinations. This latter point we will clarify in l.258 so this sentence would read: "All catchments had ~10 years of uninterrupted daily Q data available between 1995 and 2010 (Musolff, 2020) which was needed as an input for the network model." Also note that the Damköhler number (Da) and the uptake velocity (vf) are important metrics in the context of our study that we analyse to understand the relative importance of hydrological and biogeochemical processes (l.223-225 for Da) and define the uptake model (l.213-221 for vf) in the first place.

We aim to show how a time series of measured concentration (discharge is not shown here) is transferred to the C-Q space in Fig. 2 in the manuscript. In this figure the smoothed spline fits, used for calculating *Curvature*, are compared for the observed and simulated data. We already state the "classical" log-log linear slope in l. 345 but agree that it would be interesting to show the linear C-Q fit in Fig. 2b. Note that the simulation results in Table 4 report the log-log linear C-Q fit 'bout' at the catchment outlet for the different parameter combinations, next to the Curvature and the percentage load removed Lr among others.

[Figure]

In the French data that is used to validate the *Curvature* metric 77 % of the stations are characterized by Curvature ≤ 0 or a linear or concave shape (l.147). For the simulated *Curvatures* only concave shapes are generated as no point sources are considered in our approach (l.194-195 and comment R1.9).

*R1.5: General: Large number of studies show that high-freq and low-freq c-q relationships are governed by different factors. Please be clear in your paper based on which type of data you derive/base your assumptions on.*

With all due-respect to the reviewer efforts and time, we clearly mention several times in the manuscript that we consider low frequency log(C)-log(Q) relationships (l.10, 25, 99, 107, 131, 141, 151, 156, 640, 644, 669).. However, to further emphasize this point, we will add explicitly that the motivation for this study was a large-scale observational study based on low-frequency conventional monitoring (Moatar et al., 2017) so l.90 would read: "These studies identified distinct linear low-flow and high-flow NO3- log(C)-log(Q) regression slopes for a majority of the cases, **using low frequency monitoring data.**"

*R1.6: The title could be improved. I am not a big fan of bent c-q, maybe come up with a better term? For example curved c-q relationship as opposed to linear?*

We understand the reviewer's preference, but we believe that our chosen title is more informative of our work. At this junction, we would also like to recall some previous works  - well accepted in the community - that mention non-linear C-Q relationships. For example, Moatar et al, 2017 uses the term 'nonlinear' to describe 'bent' C-Q relationships, while Diamond and Cohen, 2017 talk about 'slope breaks' and Marinos et al, 2020 'piecewise power law model'. Because the referenced terminology alludes to descriptive characteristics (linear or not) rather than quantitative (amount of nonlinearity) we chose the term 'bending'. As 'bending' is our key expression for that we would prefer to keep the title.

*R1.7: Line 13 not clear what you mean by more positive slopes. Be exact.*

Line 13 says: "…that more positive log(C)-log(Q) slopes under low flow conditions (than under high flows) are linked to biological NO3- uptake, …".

*R1.8: Lines 13-15 – what about point source pollution impact on low flow concentration?*

We thank the reviewer for addressing this point. Although for the explorative modelling approach proposed in this paper, additional $NO_3^-$ sources such as incoming load resulting from point sources are not considered (similar to Bertuzzo et al., 2017; Wollheim et al., 2006) (l.194-195), we agree that point sources can have an impact on the in-stream N status. Therefore we will add the following sentences to Sect. 3.4 where we discuss the interpretation of C-Q bending at the catchment outlet: "For the parsimonious explorative modeling approach applied in this study, we mainly focused on the impacts of diffuse sources. Point source pollution, though not so significant as that of diffuse sources, can have impact during low-flow periods. Disentangling the contribution of NO3- from these two sources are challenging and remain open for further investigations."

---

## Author Comment (AC3)

**Response to Wil Wollheim**

**WW.1**: This study addresses concentration vs. discharge relationships in streams and rivers. The authors hypothesize that instream uptake will result in "bent" logC vs. logQ relationships, because net instream removal will cause lower concentrations than expected given loadings at low flows. They apply a new metric (curvature) to quantify this effect using both field data and modeling results from 13 different river networks that range in size and characteristics. They also do an extensive analysis across parameter space to understand which network characteristics have most influence on curvature and network scale removal. They find that in stream uptake can indeed lead to more bent logC-logQ relationships (because the curvature parameter is more negative), and that channel hydraulics (the width and depth vs. Q relationships) have the strongest influence. Uptake velocity (the biological parameter) seems to have less influence, which was surprising. They suggest that the curvature parameter could be used to quantify network scale removal using only the C vs. Q information, adding a potentially useful tool to understand network scale dynamics.

I think this is overall an interesting analysis and potentially a very useful approach for quantifying network scale uptake. There are a few things to consider further, emphasize or discuss, and a couple of things that would increase the understandability.

We thank the reviewer Wil Wollheim for their useful comments and their interest in our work. We address all the reviewer comments (italic) one by one below with responses in normal font.

**WW.2:** The result depends strongly on the assumption that the relationship between C and Q for loading from the landscape remains linear across seasons (i.e. the parameter b is constant). One of the difficulties getting at broad scale aquatic function is isolating landscape inputs and aquatic processes (inherent in any river network scale analysis). The constant "b" assumption is what allows inference that the bent C vs. Q relationship results from network-scale nutrient retention. Given that this analysis uses C and Q measured across seasons (as opposed to individual storm events), with seasonality correlated with flow conditions, how likely is that? That is, the loading C vs. Q relationship will differ between summer and winter, with the former tending to have lower C (e.g. due to higher riparian uptake). Would that also result in bent curves? I think this is an important consideration, worthy of some discussion.

We thank the reviewer for this comment. It is correct that the conclusions presented in this paper depend on the assumption of a linear land to stream loading vs Q with a slope that remains constant throughout the seasons. If the loading C vs. Q would differ seasonally, this could indeed result in bent loading curves which would make it hard to attribute observed bending at the catchment outlet to instream processes. There are however indications that for land to stream NO3- loading, b can be constant (Basu et al., 2011). This also connects to the idea of nitrate being mainly transport limited and not source limited especially in catchments with agriculture. There is a clear indication of this in our well studied test catchment Selke on the basis of high-frequency nitrate concentration analysis: Storm event-CQ slopes rarely changed over the seasons with similar mobilization patterns in summer and winter (Winter et. 2021). We will incorporate those new findings into the text. Nevertheless, we cannot exclude alternative loading patterns when detecting bent C-Q relationships in observed data (esp. in low-nitrate environments with potential source limitations), which we will mention and discuss explicitly in the revised manuscript. In our work we explore if in-stream removal can result in bent log(C)-log(Q) relationships at the catchment outlet and find that it can. We therefore argue that the assumption of a constant 'b' is feasible for use in the explorative modelling approach presented in this paper but surely deserves attention in future research.

**WW.3**: I agree with one of the findings that the hydraulic dimensions are among the dominant factors when considering network scale removal. However, in this analysis (if I understand right), a single hydraulic equation is used for width (and depth), i.e.  $w = Kw * Q \wedge aw$ , and a single aw is applied over both space and time. However, the hydraulics of rivers are such that the change in width with changing flow at any given site (due to storms) differs from the change in mean flow in the downstream direction (at-a-site vs. downstream hydraulic relationships). Typically the at-a-site change in w is much lower (~0.1) than in the downstream direction (~0.5) (See Knighton 1998. Fluvial forms and processes: a new perspective). It appears the best calibrated fit for one of the watersheds (Table C1) was 0.09, closer to the typical at-a-site relationship. This will greatly affect the pattern of removal within the network (small vs. large rivers) as well as with changing flow. Note that the constant (Kw) is the width (m) when Q =1m3/s. So if you have a low aw, that means large rivers stay relatively narrower and small rivers stay relatively wide (since width doesn't change much). The calibrated aw is closer to the at-a-site change (where increasing flow is accommodated mostly by changes in velocity) than the downstream change (where increasing flow is accommodated mostly by changes in width). This may explain why uptake velocity is relatively unimportant (which I was surprised by), and also why water velocity comes out as so important. It would be worth confirming whether the modeled widths match observations, and

reporting the mean width of small headwater rivers (<5km2) and larger rivers (> ~400km2) to evaluate if they are reasonable.

We thank the reviewer for these remarks. Width (w) and depth (d) have distinct parameters for their hydraulic equations, i.e.  $w = Kw * Q \land aw$  (Eq.2.1) and  $d = Kd * Q \land ad$  (Eq.2.2), respectively. Note that both channel hydraulic parameters; w and d, depend on discharge (Q) and therefore they vary depending on flow conditions. However the reviewer is right that each equation is applied over both space and time. This will be made clear in the revised manuscript when these equations are first introduced.

The effect of a low and high aw leading to constant or varying stream widths respectively the reviewer describes is also illustrated in the conceptual Fig. B5. This figure is currently briefly referenced in the manuscript, however based on the reviewer's comments we would add a sentence explaining this more in the revised manuscript.

In the 'At-a-station' panel of Fig. R2 below we evaluate the changes in velocity, depth and width with Q for a grid cell in the middle of the Selke network (point B in Fig. 3 in the manuscript). The 'Downstream' panel - that considers all the network grid cells - shows the channel characteristics width, depth and velocity for a time t, with a Q of 0.70 m3 s-1 at the outlet. The values for the parameters aw, Kw, ad and Kd are the same for both scenarios (Table C1).

Figure R2 shows a larger variability of Q in the at-a-station panel with higher Q's that are indeed accommodated mostly by the increasing velocity. As the *Curvature* metric captures the shape of an 'At-a-station' log(C) vs log(Q) relationship that is driven by the Q variability, it might explain why the channel

characteristics come out as more important for shaping this signal, compared to the uptake velocity. Therefore, we will add this consideration to the section discussing the PAWN sensitivity analysis.

In the figure below (R3) we show the median widths at the catchment outlet for each of the >11000 model parameter combinations used in the Monte Carlo simulation. These parameter combinations were chosen randomly within some set physical boundaries (I.233-237; Table 1) and will therefore cover realistic as well as unrealistic stream channel widths at the catchment outlet as can be seen in Fig. R3. The width for the selke Meisdorf was reported by Rode et al., 2016 and is indicated in the boxplot with an asterisk. This point falls well within the simulated width range for this catchment. Following this comment, we will add a sentence in the revised manuscript stating that the modelled widths are to a large degree reasonable. The figure with the width at the outlet will be added to the supporting information.

*WW.4:* It is interesting and a bit surprising that vf had a relatively small impact. The authors state that if vf = 0, there is no bending (conservative) – and of course I agree. But it seems that a low vf would then result in only slight bending, which will only increase as vf increases. Does this pattern not occur? The choice of vf in the paper is appropriate for denitrification, but it is on the low side total N uptake (assimilation) which could be 5-10x higher than for denitrification (e.g. Mulholland et al. 2008 found denitrification was ~15% of gross nitrate uptake). Net assimilation may also be important in watersheds at certain times, particularly during lower flow summers (storing N over medium time scales, or transforming to PON or DON). Might this ever be a factor in the watershed considered. Could the Monte Carlo analysis address this possibility by using a higher Vf to determine at what point vf dominates the bending?

We thank the reviewer for their comment. In Fig. R4 below we show C-Q relationships for increasing uptake velocities vf resulting from network model simulations of the Selke Meisdorf catchment. As an example, two different aw are displayed with the other parameter values as stated in Table C1. We see that although concentrations gradually decrease for increasing vf, *Curvature* remains rather constant for these example simulations apart from values close to zero for very small vf. Because this figure can help to understand the dynamics between vf, aw and *Curvature* we will add it in the revised manuscript.

---

## Author Response (AR1)

Response 2

**Response to editor**

Dear prof. Dr. Stamm,

Thank you for your comments. We address each comment (in italic) with a response in normal font.

**Ed1.** *Title and terminology. Reviewer 1 questioned the title and asked for a substitute for "Bending the concentration discharge relationship…". After reading again the manuscript, the comments, and the response I think that the open issue is broader than just the title and concerns your terminology on bending and curvature in general.*

*Let me start with some general reflections on the use of terminology. Terms should be used in a way to facilitate communication between the authors of a paper and its readers. Using a common vocabulary or terminology helps to convey the message. This implies that if a term has been established in a scientific community to describe a well-defined object, concept etc. one should use this term instead of introducing a new one.*

*However, this is exactly what you in your paper. You argue that the term bending were to be preferred over others such as nonlinear because it points to the quantitative aspect of nonlinearity. I have doubts about that argument for two reasons: first, for my understanding bending is not more quantitative than other terms. Second, you introduce another term, which by itself is not very precise. Therefore, I suggest to stick to an existing term (e.g., The degree of non-linearity … " or simply "Curvature of …"). If you stick to the term Bending, provide a clear linguistic argument by a native speaker that this verb was more adequate than the other alternatives.*

*Throughout the manuscript you use the term curvature. Curvature in itself has very clear mathematical meaning to which you also refer to in sec. 2.1. However, you use the term in two different ways. On L. 119 – 122 you describe the general meaning. But on L. 118, you state that you introduce the new concept curvature. This is misleading and distracts the reader from what you have actually done: you have introduced the maximum (absolute value of) curvature of (a fitted spline for) the log(C)-log(Q) relationship as a metric. Labelling this metric with a specific term e.g., Curvaturemax may help the reader to be always clear of what you mean.*

*I suggest that you change the title to something like "Curvature of the concentration discharge relationship can inform about in-stream nitrate removal". Pay also attention to distinguish clearly between the general meaning of curvature (as it is established) and the specific metric you have introduced. In my view, such a change in terminology would get your ideas clearer across without any loss of content. You may have also noticed that several comments by the reviewer indicate the difficulty in always understanding what you mean (see e.g., WW.7).*

*Should you disagree, you have to clear arguments why my reasoning falls short.*

We thank the editor for the elaborate comments and suggestions on the terminology that is used in this manuscript. We agree that the term curvature has a clear mathematical definition that should not be confused with the metric '*Curvature'* that is introduced here. To distinguish our metric from the

aforementioned mathematical definition, we label it with the term $Curv_{max}$ in the revised manuscript. Similarly, for the terminology in the title we would not use 'curvature' as it could be interpreted to refer to the mathematical definition of curvature rather than the introduced metric $Curv_{max}$ which in itself also does not fit in the title.

Previous research describes the shape of 'non-linear' C-Q relationships with the terms "segmented linear", "piecewise power law relationship" or "broken stick" (l.115-119), however this is to the best of our knowledge the first manuscript quantifying the (general) shape of such C-Q relationship with a single parameter. We therefore argue that introducing the intuitive terminology of 'bending' is feasible and necessary.  The suggested option of "the degree of non-linearity" would be too general in our opinion and less straightforward to interpret as "amount of bending", which we now introduced in the manuscript in l. 129: "$Curv_{max}$ of a log(C)-log(Q) relationship could be considered as a complementary metric to the slope of the linear regression model (Godsey et al., 2009) and could serve as an alternative for segmented linear regression fits (Meybeck and Moatar, 2012; Moatar et al., 2017; Marinos et al., 2020) (Fig. B2) as it quantifies the degree of non-linearity as the amount of bending." We also ran this by a native English speaker who used the wording of curvature and bending interchangeably in this context "…greater curvature or more bending…". Taking into account the previous argument on the possible confusion with the mathematical interpretation of curvature, we wish to  keep the wording "bending" in the manuscript title; which in our opinion reflects the main content of the presented work i.e., about quantifying a general shape of C-Q relationship and extracting  information about in-stream nitrate removal.

**Ed2.** *Balance between modelling and (empirical) results. Reviewer 1 criticized that the manuscript was too heavy on the modelling side hiding the actual scientific problem (to understand the non-linearity of C-Q relationships). Also Reviewer 2 asked at some points for more empirical data. You – in my view – point out correctly that the main focus of this manuscript was modelling. Therefore, this part requires sufficient details to be understood. Nevertheless, you indeed hide important empirical data in the Method section. The paragraph on L: 146 – 156 reports empirical observations and does not belong into the Method section. Moving this part into the Results provides a much stronger empirical basis for why your model approach is relevant.*

*You mention that you have used the measured data of the 13 German catchment as input for exploratory. However, you don't show these data (or did I overlook tem?) nor do you reveal the comparison of the respective model outputs to the actual observations.*

*Given that you draw important conclusions from your model results, the reader should learn about how well the models (with the ensemble of model results for each catchment based on the Monte Carlo simulations) compare to actual observations: do these ensembles include the observations or do they substantially deviate? The reader needs to be informed about such aspects.*

*In a similar way, statements such as on L. 514 – 515 can be tested with the empirical data (the French and the German data set (e.g., (Ebeling et al., 2021)).*

*Additionally, the text is often very lengthy. Consider shortening and being concise. Skip sentences of secondary importance (e.g. such as L. 498 – 501)*

We thank the editor for their supportive comments. We moved the paragraph on l.126-156 to the results section in the revised manuscript to provide a clearer empirical basis for the modelling approach as suggested (Section 3.1: Empirical curvmax).

The 13 German catchments are selected based on their distinct sizes and geophysical and hydrological settings (e.g. stream order, median discharge and catchment shape) (l.253-258, Table 2). These are then used to apply the exploratory model in a range of realistic stream networks with corresponding observed discharges. This is in contrast to other works that applied this network based uptake model in synthetic Optimal Channel networks (OCN, Bertuzzo et al., 2017; Helton et al., 2018) which delineate networks based on energy expenditure.

It is in these realistic 13 stream networks that we apply the exploratory model. Note that the discharge at the catchment outlet was the only time-varying measured data used here (this is now specified in the revised manuscript in response to R.1.5). To validate if our exploratory model is feasible and can produce realistic results, we verify the model components in the Selke catchment, where we have access to both high-quality, long time-period observations of discharge and $NO_3^-$ concentrations at the outlet of the catchment (Winter et al., 2020). From this verification exercise, we concluded the adequacy and feasibility of the uptake model; and then continue to apply a Monte Carlo based approach wherein we explore which input parameters can lead to a bended C-Q relationship in different settings of network structures. So we don't aim to reproduce the catchment specific concentrations at the outlet in the 13 catchments, but rather explore the parameter space to be able to link certain parameter combinations to the C-Q bending. Nevertheless, we now report some additional plots where we show that realistic values for the Selke validation example (for example for channel width) are within the simulated Monte Carlo range. We also clarified that specifically the river network and the daily discharge at the catchment outlet serve as network model inputs in Sect. 2.3.1.

The statement on l. 514-515 the editor refers to: "$Da$ on the other hand is positively correlated to $Lr.perc$ ($\rho$ = 0.58) which indicates that higher $Da$ are occurring together with higher load removed." is unfortunately not justifiable with the empirical study the editor suggests, as that previous study does not quantify nitrate retention in study catchments. However, there are other studies that specifically focused on the link between $Da$ and the removal of nitrate (for example, Ocampo et al., 2006) which we have discussed in l.494-497.

We agree with the Editor's remark that the sentence mentioned (l.498-501) is repetitive; and have therefore removed it in the revised manuscript. We also thoroughly checked the manuscript for repetitive statements and have removed them accordingly, like the sentence at the end of the last paragraph of Section 3.2 (also see editor comment Ed5).

**Ed3.** *Reviewer 1 asked for a more in-depth discussion of the relevance of point sources. Your responded that "Disentangling the contribution of $NO_3^-$ from these two sources are challenging and remain open for further investigations.". I have problems to follow this argument. Point sources often cause a rather constant input over time resulting in a dilution effect in C-Q relationships. Based on the fact that you have hundreds of catchments with data sets at hand (the French ones plus the German ones just published by some of the co-authors regarding C-Q relationships in Ebeling et al., (2021)), you can check the data sets in how many catchments and data sets such an dilution effect can be observed. Based on this analysis, one can also discuss – at least qualitatively – how these point sources may have affected the curvature analysis.*

We agree with the editor that the initial response to comment R1.8 was limited and aim to clarify things better. Although for the explorative modelling approach proposed in this paper, additional $NO_3^-$ sources such as incoming load resulting from point sources are not considered (similar to Bertuzzo et al., 2017; Wollheim et al., 2006) (l.194-195), we agree that point sources can have an impact on the in-stream N status. However, large scale assessments across Germany (Ebeling et al, 2021) and France (Moatar et al., 2017, see Fig. 7 therein) explicitly looked for waste water impacts in C-Q relationships. There was no impact found for waste-water derived nitrate on the C-Q shape, but rather influences on the concentration level. More specifically diffuse inputs (N surplus from agriculture and atmospheric deposition) contribute 30 to 60 times more flux into the catchments than N- from waste water. Out of the diffuse inputs about 75% is held back within the catchments, while the waste water is going directly into the river. However, this does not compensate for the high diffuse inputs in relation to point sources (Ebeling et al., 2021). In essence, 'Point sources were found not the main controlling sources for NO3- concentrations in Germany nor a dominant factor for NO3- concentration variability (C-Q relationships) (Ebeling et al., 2021). Furthermore it is worth noting that dilution patterns of NO3- are very rare across the study catchments, and are not connected to urban influences (Ebeling et al., 2021). We address this in the methods Section 2.2.2 of the revised manuscript where we have first introduced the assumption of not considering point sources. Later we revisit this assumption made on other potential processes shaping the C-Q relationship in Sect. 3.4 where we discuss the interpretation of C-Q bending at the catchment outlet.

**Ed4.** *L. 372 – 373: You mention that the model was able to properly represent the seasonality in the Selke River. However, Fig. 2 does not demonstrate the seasonal model predictions. Please show the respective data.*

We thank the editor for this comment. The seasonal model predictions are included in Fig. 2 as the black line. The legend of this figure has now been adjusted to "Simulated C" to make this point clearer to the reader. (Also see comment R1.4)

**Ed5.** *The manuscript is rather lengthy due to unnecessary repetitions or reporting on details of secondary importance. Please carefully check the entire manuscript and skip redundancies (e.g., content of L.487, 511 – 514 is repeated L. 527 – 532; L. 489 parameter b, L: 539, L. 544 – 546).*

We thank the editor for the helpful suggestions. . We agree that some sentences were redundant in the previous version of the manuscript (also see Ed. 2). We have carefully checked the manuscript for the redundancy and duplications

**Ed6.:** *Velocity is not equal to rate: velocity is generally the term for describing the physical displacement of an object in space while rate describes e.g., mass turn over in time (for biological and chemical processes). I suggest replace nitrate uptake velocity by uptake rate.*

We thank the editor for this comment. The uptake velocity (parameter $v_f$) refers to a mass transfer velocity of nutrient molecules moving (vertically) through the water column to the benthos (Ensign and Doyle, 2006) and is a well established term (Bertuzzo et al, 2017; Ensign and Doyle, 2006; Wollheim et al., 2006; Marcé et al., 2018). We adapted l.206-208 in the revised manuscript to include the physical meaning of vf: "The uptake velocity parameter $v_f$ [m day$^{-1}$] refers to the vertical movement of $NO_3^-$ molecules from the water column towards the biofilm at the pelagic-benthic interfaces and the

sediments where the in-stream processing chiefly occurs with $v_f = k_i d_i$ and $k_i$ the first order removal constant (Ensign and Doyle, 2006; Wollheim et al., 2006; Marcé et al., 2018)."

**Ed7**.: *L. 438 – 441: Why do you compare these results with the French catchments and not with the actual values form the 13 catchments?*

The French catchments serve a two-fold purpose in this manuscript. First, they are used to verify the robustness of the newly introduced *Curv$_{max}$* metric. This is done by selecting 444 low frequency C-Q relationships from the French data set that follow certain selection criteria (Section 2.1), taking subsamples of the entire time series for one station, calculating *Curv$_{max}$* and checking the robustness of the *Curv$_{max}$* calculation (Section 3.1). Second, we use the computed *Curv$_{max}$* for the entire time series in the selected French catchments as a realistic range to compare our simulated *Curv$_{max}$* to corresponding ones generated with the various input parameter sets (>10000) in the 13 river networks.

With the explorative network model, we do not aim to reproduce observed NO3- concentrations at the catchment outlet but rather uncover the link between certain input parameter combinations and the amount of log(C)-log(Q) bending at the catchment outlet. We selected 13 German river networks that vary in size, shape and stream order among others to be able to apply the network model in a range of river networks.

We have revised the text and clarified this part better in Section 2.3.1 and in newly added Section 2.3.2 where the model evaluation and Monte Carlo output are described in detail. Additionally we have added a couple of sentences in l.477-483 so that it now reads: "The Monte Carlo output in Table 4 shows reasonable values for the different variables, taking into account that the goal of this modelling exercise was not to reproduce catchment specific conditions but rather explore how uptake influences C-Q bending for a range of parameter combinations that represent a spectrum of possible catchment conditions. The simulated *Curv$_{max}$* for all 13 German study catchments and parameter combinations (80 % of the values between -0.70 and -0.012, Table 4 and Fig. B5) are comparable with the range of *Curv$_{max}$* from log(C)-log(Q) relationships in the French catchments (80% of the values between -0.41 and -0.067; Fig. B4) (Dupas et al., 2019). "

**Ed8**.: *L. 509: what is the slope of the logC-logQ relationship? Did you also calculate a linear model? You haven't explained this.*

We thank the editor for this comment. The variable $b_{out}$ is indeed the slope of the fitted linear model of the logC-logQ relationship, calculated at the catchment outlet. We previously missed to explain this in Method Section 2.3, and thus have revised the sentence as: "The **simulated variables** are i) *Curv$_{max}$* [-], deduced from simulated log(C)-log(Q) relationships when minimum 80 % of the C data is above the 'detection limit' of 0.002 mg L$^{-1}$ $NO_3^-$; ii) the network wide percentage load removed $L_{r.perc}$ [%] which is calculated as the median of the ratio between the daily absolute removed load and the daily absolute incoming load in the river network; iii) the median network travel time, TT [days]; (iv) the Damköhler number $Da$ [-]; (v) the slope of the linear regression fit of the log(C)-log(Q) relationship at the catchment outlet, $b_{out}$ [-]; (vi) the median concentration at the catchment outlet, $C_{out}$ [mg L$^{-1}$] and the median water velocity, $v$ [m s$^{-1}$]."

**Response to Reviewer 1**

We would like to thank reviewer 1 for their comments on the paper 'Bending of the concentration discharge relationship can inform about in-stream nitrate removal'. We address all the reviewer comments (in italic) one by one below with responses in normal font.

*R1.1: The paper aims to explain evolution of concentration-discharge patterns in a stream network using a modelling approach. This is an interesting topic, following observations from many systems, where the c-q patterns become homogenised downstream, i.e. from highly variable and positive c-q slopes in first order streams to more linear responses, near chemostatic responses in downstreams.*

*The modelling approach adopted here explains one aspect of these previous observations, i.e. changes in curvature. The authors show that in 1st order streams curvature is larger than in higher order streams which can be explained by hydrological accumulation and homogenisation when moving downstream. Simply speaking, 1st order streams can have a larger variation in concentration sources compared to bigger streams. And/or activation/deactivation of these sources requires changes in flow discharge that can result in the 'bent' c-q relationship or simply speaking different slopes of the relationship for different flows.*

We thank the reviewer for this comment and we generally agree here. The main aim of the paper is to examine if network scale nitrate uptake effects can be inferred from the bending of low frequency; multi-annual concentration (C) and discharge (Q) observations. Thereto we apply a parsimonious river network model (similar to Bertuzzo et al., 2017; Helton et al., 2018; Helton et al., 2010; Mulholland et al., 2008) in 13 German catchments to explore the catchment scale transport and uptake processes that influence downstream log(C)-log(Q) patterns (l.103-107).

We show that *Curvature* converges when moving from lower order to higher order streams in Fig. 3 where the spatial distribution of simulated Curvature in the Selke river network (Meisdorf) for a selected parameter set is shown (l.387). In this case, uptake and land to stream loading at the downstream grid cells have a decreasing local impact on the outgoing load due to the relatively larger upstream contributions that increase in the downstream direction (l.441). The convergence of *Curvature* in higher order streams is also shown in the results of the Monte Carlo simulations, where >10 000 parameter sets were applied to 13 German catchments (l.504-508). Here, simulated *Curvature* in lower order catchments 1, 5 and 11 has a higher variance and an overall lower mean value (higher bending) than the simulated *Curvature* in higher order catchments 4 and 6 (Table 4) . It is however clear that these

differences between the catchments cannot be attributed to a single catchment property such as total network length or basin area (l.503-504)

*R1.2: The paper is heavy on modelling that can conceal the main findings of the paper, which is that 1) curvature is predominant in 1st order streams and 2) curvature is better explained by flow characteristics than nitrate uptake velocity. These findings can be explained by higher rate of biological processes in headwater streams (hypothesis investigated in this paper) but there are also other factors that can explain bending of the c-q curves, like stream morphology and flow-stage relationships, activation/deactivation of sources in relation to flow including presence of sewage pollution, drains etc. Thus, I am not convinced that the paper provides the one and only explanation for the observed patterns, rather than provides a plausible explanation for one of the possible explanations. This should be clearly communicated in the paper.*

In this paper, we indeed investigate one possible explanation for bent log(C)-log(Q) relationships, namely that log(C)-log(Q) bending can inform about in-stream $NO_3^-$ removal. We don't aim to state that in-stream uptake is the only explanation for C-Q bending but rather that it is one possible explanation that is motivated from previous observation- and model driven studies (Moatar et al., 2017; Hall et al., 2009; Hensley et al., 2014; Wollheim et al., 2017, l.90-99). Note that flow stage relationships (and stream geomorphology in general) were accounted for in our modelling setup (l.171-172). We made the point that we aim to investigate only one of the possible explanations for C-Q bending more explicit and clear in the revised version of the manuscript by adjusting the introduction, assumptions in the methods and conclusions. For example, we edited l. 325 to read: "Finally, we aim to determine if, within this modelling framework, C-Q bending at the catchment outlet (specifically *Curv.max*) informs about the network wide in-stream uptake." We want to refer the reviewer to editor comment Ed3. for a discussion of the point source impact on the C-Q relationship and the associated changes to the manuscript.

*R1.3: The modelling focus of the paper is, however, very dense and takes precedence over the problem – variations to c-q patterns and their controls. I would suggest 'moving' the modelling to the background of the paper and focusing more on the problem. This refocusing would make the paper easier to understand to a non-modelling reader and set it better in the previous research on the topic.*

We appreciate the reviewer's suggestion - however we would like to keep the modeling part intact, without losing the main focus of the paper. In this respect we believe that a thorough description of the

Monte Carlo method and the uptake model are essential for the interpretation of the results as our study of the C-Q patterns and their controls is modelling-based. However, we revised the manuscript, and in particular the conclusion section, to ensure that the key findings can be understood by all the readers. We refer to comment WW.11 for the specific changes to improve the conclusion.

*R1.4: Finally, the original concentration and flow data disappear in the paper convoluted in different models. E.g. linking curvature to other models with inherent uncertainty like Damköhler number or uptake velocity. Showing more raw data in the paper, e.g. providing traditional quantifications of the c-q slopes would be very useful for the reader to link their knowledge of the subject with the new findings of this paper. Also when showing variation in curvature, I would like to know how frequent are concave vs. convex shapes.*

Thank you again for your detailed suggestions. Our analysis relies on 'raw data'/ observations in the following instances; i) low frequency C and Q data for 444 French catchments that is used to validate the *Curv.max* metric; ii) measured NO3- concentrations in the Selke Meisdorf station to validate the conceptual network model structure by comparing modelled observed NO3- at the network outlet (Fig. 2a and b); iii) measured daily Q time series for 13 German catchments that were used as a direct input for the explorative model to simulate NO3- concentrations (and *Curvature*) for a range of parameter combinations. This latter point we clarified in Section 2.3.1, with the specific edits mentioned in the response to comment Ed.2.

Also note that the Damköhler number (Da) and the uptake velocity (vf) are important metrics in the context of our study that we analyse to understand the relative importance of hydrological and biogeochemical processes and define the uptake model in the first place.

We aim to show how a time series of measured concentration (discharge is not shown here) is transferred to the C-Q space in Fig. 2 in the manuscript. In this figure the smoothed spline fits, used for calculating *Curvature*, are compared for the observed and simulated data. We already mention the "classical" log-log linear slope in l. 372 (b.out = 0.40, $R^2$ = 0.56) but agree that it would be interesting to show the linear C-Q fit in Fig. 2b. We therefore have added the observed linear fit to the updated Fig. 2b. Note that the simulation results in Table 4 report the log-log linear C-Q fit 'bout' at the catchment outlet for the different parameter combinations, next to the Curv.max and the percentage load removed Lr among others.

[Figure]

**Figure 2: (a) Simulated and observed concentrations at the Selke Meisdorf gauging station for a 10 year simulation period (2000-2010; NSE=0.50). One data point (C~5 mg L⁻¹) is not shown here. The simulated median percentage of load removed in the stream network (blue line) is given during the same time period as well as the simulated C with no uptake ( =0). (b) The observed concentrations and Q are log transformed and plotted together with the simulated C-Q data for 2000-2010. A smoothed spline is fitted to the observed and simulated C-Q data (described as observed smooth fit and simulated C respectively in the legend); and *Curv$_{max}$* of -0.35 and -0.28 are calculated at the respective discharges of 1.72 m³ s⁻¹ and 0.92 m³ s⁻¹, indicating the smoothed spline inflection points.**

In the French data that is used to validate the *Curvature* metric 77 % of the stations are characterized by Curvature ≤ 0 or a linear or concave shape (l.350). For the simulated *Curvatures* only concave shapes are generated as no non-linear loading patterns are considered in our approach (response to comment Ed. 3).

*R1.5: General: Large number of studies show that high-freq and low-freq c-q relationships are governed by different factors. Please be clear in your paper based on which type of data you derive/base your assumptions on.*

Thank you for that remark. We clearly mention several times in the manuscript that we consider low frequency log(C)-log(Q) relationships (l.11, 16, 26, 100, 109, 135, 144, 354, 359, 673, 677, 705). However, to further emphasize this point, we will add explicitly that the motivation for this study was a large-scale observational study based on low-frequency conventional monitoring (Moatar et al., 2017) so l.91 would read: "These studies identified distinct linear low-flow and high-flow $NO_3^-$ log(C)-log(Q) regression slopes for a majority of the cases, using low frequency monitoring data."

*R1.6: The title could be improved. I am not a big fan of bent c-q, maybe come up with a better term? For example curved c-q relationship as opposed to linear?*

We would like to refer to the response to the comment Ed1 by the editor.

*R1.7: Line 13 not clear what you mean by more positive slopes. Be exact.*

We have changed the wording here to: "...that steeper positive log(C)-log(Q) slopes under low flow conditions (than under high flows) are linked to biological $NO_3^-$ uptake, …".

*R1.8: Lines 13-15 – what about point source pollution impact on low flow concentration?*

We want to refer the reviewer to editor comment Ed3. for a thorough discussion of the point source impact on the C-Q relationship and the changes we made in the revised manuscript.

**Response to Wil Wollheim**

*WW.1: This study addresses concentration vs. discharge relationships in streams and rivers. The authors hypothesize that instream uptake will result in "bent" logC vs. logQ relationships, because net instream removal will cause lower concentrations than expected given loadings at low flows. They apply a new metric (curvature) to quantify this effect using both field data and modeling results from 13 different river networks that range in size and characteristics. They also do an extensive analysis across parameter space to understand which network characteristics have most influence on curvature and network scale removal. They find that in stream uptake can indeed lead to more bent logC-logQ relationships (because the curvature parameter is more negative), and that channel hydraulics (the*

*width and depth vs. Q relationships) have the strongest influence. Uptake velocity (the biological parameter) seems to have less influence, which was surprising. They suggest that the curvature parameter could be used to quantify network scale removal using only the C vs. Q information, adding a potentially useful tool to understand network scale dynamics.*

*I think this is overall an interesting analysis and potentially a very useful approach for quantifying network scale uptake. There are a few things to consider further, emphasize or discuss, and a couple of things that would increase the understandability.*

We thank the reviewer Wil Wollheim for their useful comments and their interest in our work. We address all the reviewer comments (italic) one by one below with responses in normal font.

***WW.2****: The result depends strongly on the assumption that the relationship between C and Q for loading from the landscape remains linear across seasons (i.e. the parameter b is constant). One of the difficulties getting at broad scale aquatic function is isolating landscape inputs and aquatic processes (inherent in any river network scale analysis). The constant "b" assumption is what allows inference that the bent C vs. Q relationship results from network-scale nutrient retention. Given that this analysis uses C and Q measured across seasons (as opposed to individual storm events), with seasonality correlated with flow conditions, how likely is that? That is, the loading C vs. Q relationship will differ between summer and winter, with the former tending to have lower C (e.g. due to higher riparian uptake). Would that also result in bent curves? I think this is an important consideration, worthy of some discussion.*

We thank the reviewer for this comment. It is correct that the conclusions presented in this paper depend on the assumption of a linear land to stream loading vs Q with a slope that remains constant throughout the seasons. If the loading C vs. Q would differ seasonally, this could indeed result in bent loading curves which would make it hard to attribute observed bending at the catchment outlet to in-stream processes. There are however indications that for land to stream $NO_3^-$ loading, b can be constant (Basu et al., 2011). This also connects to the idea of nitrate being mainly transport limited and not source limited especially in catchments with agriculture. There is a clear indication of this in our well studied test catchment Selke on the basis of high-frequency nitrate concentration analysis: Storm event-CQ slopes rarely changed over the seasons with similar mobilization patterns in summer and winter (Winter et. 2021). We incorporated those considerations into the text of the methods section where b was first introduced as "Here, $b$ is assumed to be constant over the seasons, which is supported by

findings that $NO_3^-$ loading is transport limited rather than source limited, especially in agricultural catchments (Basu et al., 2011; Winter et al., 2021)." Nevertheless, as we cannot exclude alternative loading patterns when detecting bent C-Q relationships in observed data (esp. in low-nitrate environments with potential source limitations). In our work we explore if in-stream removal can result in bent log(C)-log(Q) relationships at the catchment outlet and find that it can under the chosen model setup and the given assumptions. We now added that aspect to the discussion section 4.3. We therefore argue that the assumption of a constant 'b' is feasible for use in the explorative modelling approach presented in this paper but surely deserves attention in future research.

*WW.3: I agree with one of the findings that the hydraulic dimensions are among the dominant factors when considering network scale removal. However, in this analysis (if I understand right), a single hydraulic equation is used for width (and depth), i.e. w = Kw * Q ^ aw, and a single aw is applied over both space and time. However, the hydraulics of rivers are such that the change in width with changing flow at any given site (due to storms) differs from the change in mean flow in the downstream direction (at-a-site vs. downstream hydraulic relationships). Typically the at-a-site change in w is much lower (~0.1) than in the downstream direction (~0.5) (See Knighton 1998. Fluvial forms and processes: a new perspective). It appears the best calibrated fit for one of the watersheds (Table C1) was 0.09, closer to the typical at-a-site relationship. This will greatly affect the pattern of removal within the network (small vs. large rivers) as well as with changing flow. Note that the constant (Kw) is the width (m) when Q = 1m3/s. So if you have a low aw, that means large rivers stay relatively narrower and small rivers stay relatively wide (since width doesn't change much). The calibrated aw is closer to the at-a-site change (where increasing flow is accommodated mostly by changes in velocity) than the downstream change (where increasing flow is accommodated mostly by changes in width). This may explain why uptake velocity is relatively unimportant (which I was surprised by), and also why water velocity comes out as so important. It would be worth confirming whether the modeled widths match observations, and reporting the mean width of small headwater rivers (<5km2) and larger rivers (> ~400km2) to evaluate if they are reasonable.*

We thank the reviewer for these remarks. Width (w) and depth (d) have distinct parameters for their hydraulic equations, i.e. *w = Kw * Q ^ aw* (Eq.2.1) and *d = Kd * Q ^ ad* (Eq.2.2), respectively. Note that both channel hydraulic parameters; w and d, depend on discharge (Q) and therefore they vary depending on flow conditions. However the reviewer is right that each equation is applied over both

space and time. This is made clear in the revised manuscript when these equations are first introduced (l.167).

The effect of a low and high aw leading to constant or varying stream widths respectively the reviewer describes is also illustrated in the conceptual Fig. B3a. This figure is currently briefly referenced in the manuscript, however based on the reviewer's comments we have added a sentence explaining this more in the revised manuscript: "The width-discharge relation in Eq. (2.1) is conceptually illustrated in Fig. B3a for two sets of $a_w$ and $K_w$, where a low $a_w$ corresponds to the width of a channel that does not change much with varying discharge, while a high $a_w$ can result in highly varying channel widths."

[Figure]

**Figure B3: (a) The effect of parameters a_w and K_w on the channel width illustrated for the Q timeseries at the Selke Meisdorf station. (b) In the 'At-a-station' panel the changes in velocity, depth and width with Q for grid cell B (Fig. 3) in the middle of the Selke network are evaluated. The 'Downstream' panel - that considers all the network grid cells - shows the channel characteristics width, depth and velocity for a time t, with a Q of 0.70 m³ s-1 at the outlet. The values for the parameters a_w, K_w, a_d and K_d are the same for both scenarios (Table C1).**

Figure B3b shows a larger variability of Q in the at-a-station panel with higher Q's that are indeed accommodated mostly by the increasing velocity. As the *Curvature* metric captures the shape of an 'At-a-station' log(C) vs log(Q) relationship that is driven by the Q variability, it might explain why the channel characteristics come out as more important for shaping this signal, compared to the uptake velocity. Therefore, we added this Figure to the appendix B.

In the figure below (S4) we show the median widths at the catchment outlet for each of the >11000 model parameter combinations used in the Monte Carlo simulation. These parameter combinations were chosen randomly within some set physical boundaries (Table 1) and will therefore cover realistic as well as unrealistic stream channel widths at the catchment outlet as can be seen in Fig. S4. The width for the Selke Meisdorf was reported by Rode et al., 2016 and is indicated in the boxplot with an asterisk. This point falls well within the simulated width range for this catchment. Following this comment, we added a sentence in the discussion of Sect 3.2 in revised manuscript stating that the modelled widths are to a large degree reasonable. The figure with the width at the outlet is added to the supporting information.

[Figure]

*Figure S4: Distributions of the mean river width at the catchment outlet as a result of the 11107 Monte Carlo simulations that were run for each of the 13 catchments. The star indicates the observed width for the Selke Meisdorf catchment, which falls well within the simulated model range.*

*WW.4: It is interesting and a bit surprising that vf had a relatively small impact. The authors state that if vf = 0, there is no bending (conservative) – and of course I agree. But it seems that a low vf would then result in only slight bending, which will only increase as vf increases. Does this pattern not occur? The choice of vf in the paper is appropriate for denitrification, but it is on the low side total N uptake (assimilation) which could be 5-10x higher than for denitrification (e.g. Mulholland et al. 2008 found denitrification was ~15% of gross nitrate uptake). Net assimilation may also be important in watersheds at certain times, particularly during lower flow summers (storing N over medium time scales, or transforming to PON or DON). Might this ever be a factor in the watershed considered. Could the Monte Carlo analysis address this possibility by using a higher Vf to determine at what point vf dominates the bending?*

We thank the reviewer for their comment. In Fig. B6 below we show C-Q relationships for increasing uptake velocities vf resulting from network model simulations of the Selke Meisdorf catchment. As an example, two different $a_w$ are displayed with the other parameter values as stated in Table C1. We see that although concentrations gradually decrease for increasing vf, *Curvature* remains rather constant for these example simulations apart from values close to zero for very small vf. Because this figure can help to understand the dynamics between vf, aw and *Curvature* we added it in the revised manuscript.

**Figure B6: Log(C)-log(Q) relationships and *Curv$_{max}$* for increasing uptake velocities (vf ) resulting from network model simulations of the Selke Meisdorf catchment. As an example, two different  are displayed and  is varied from almost 0 to 2.5 m day$^{-1}$ with the other parameter values as stated in**

**Table C1.**

[Figure]

We assigned the values for vf based on a database compiled by Marcé et al., 2018. Here vf was collected from 83 published studies for >260 rivers (1-3$^{rd}$ order mainly). The studies used addition experiments that were typically conducted under base flows or low flow conditions and calculated vf based on the nutrient spiraling equations accounting for biotic (assimilatory and dissimilatory) and abiotic uptake (Stream solute workshop, 1990). However the range of 10$^{-4}$ to 0.25 m/day for vf we finally selected based on a subset of this dataset is indeed low. The values for the median vf, 1$^{st}$ and 3$^{rd}$ quartile respectively 1.30, 0.47 and 5.76 m/day, taking into account the entire database. These values might exceed the 'real' in-situ vf as they were mostly obtained from nutrient additions (Hensley et al., 2014; Mulholland and Tank, 2002). As an example Fig. R4 above shows how a wider vf range (between 10$^{-4}$ to 2.5 m/day) does not affect *Curvmax* much. The higher vf values clearly lead to lower concentrations that are sometimes below the limit of quantification in real-world data (which we don't observe at the Selke and the other test catchments). This leads us to conclude that in the Monte Carlo analysis we can focus on the rather lower range of uptake where most changes in bending happens.

In the revised manuscript we added a footnote to Table 1 stating that: "For v_f, the selected range is an order of magnitude smaller than the one proposed by Marce et al., 2018 as we focus the analysis on the lower v_f where most of the bending happens (Sect. 3.3)."

*WW.5: Given that these C vs. Q patterns are based on samples collected over the year there is also the confounding effect of temperature on biological activity. Denitrification is often represented with Q10 = 2, so winter (cold temperature) reactivity could be much lower. I know this was not part of the analysis, but given the use of C collected over seasons, it seems important to factor in somehow, at least in the discussion. The temperature effect, correlated with Q, would cause a more rapid shift to saturation with increasing flow (since most of flow change is likely seasonally driven, given the sampling regime). Should discuss whether this factor is potentially important, why or why not?*

This is an interesting point, thank you. We agree that this factor can be potentially important and surely will also interact with the strong seasonality in discharge and flow velocity. We already mention in l.421-423 in the discussion of the Selke example that "we did not account for the temporally changing effects of environmental factors like temperature and light availability that might (seasonally) influence uptake efficiencies in the river network". We added a reference here in the revised manuscript and come back to this limitation in Section 3.4.

*WW.6: I also had a question about how "bentness" (=curvature) is represented in Figure B1, discussed, and demonstrated. It would help me a lot (and I assume other readers) if some of the empirical patterns of log C vs. log Q were shown. Examples for different values of the curvature parameter (end members, the median, and 0) would be helpful. Especially since one of the conclusions is about the utility of these low frequency empirical data sets (L641) and given that much of the recent literature has used high frequency data to get at C vs. Q relationships.*

We thank the reviewer for this comment and replaced Fig. B1 by the Fig. B1 (reviewed) below, where the iterative fitting of the Selke data is shown in the upper panel and the corresponding local curvature in the lower panel. The value of the *Curvmax* metric results from the region of the largest instantaneous change.

[Figure]

**Figure B1 (reviewed): Conceptual figure explaining Curvmax. The upper panel shows the log(C)-log(Q) relationship for Selke Meisdorf with the smoothed spline fits to this data for different degrees of freedom (df). The corresponding colors in the lower plot show then the local curvature values for these fitted smoothed spline. Also the log(Q) is indicated for which the largest local curvature was found for each of the smoothed splines. Curvmax is then calculated as the largest local curvature value for a certain degree of freedom. Note that when Curvmax < 0 the curve is concave while for Curvmax > 0 the curve is convex.**

*WW.7: Also, I would consider some of the wording regarding "less curvature". I initially assumed that meant straighter. But in fact, "less curvature" meant a more negative curvature parameter, which is actually more bent. It took me a while to get straight.*

We thank the reviewer for pointing this out and agree that the current wording with 'small' and 'large' *Curvature* might be confusing. We therefore decided to use the terms 'low' and 'high' $Curv_{max}$ to describe respective more and less bent simulated log(C)-log(Q) relationships. This wording is now applied consistently throughout the text.

*WW.8: In conceptual figure B1, I think that the bentness as I understand it should show a straight line at high flow parallel to the curvature equal 0 line, but bending down as flows decline. If the dynamic is saturation, it should approach the slope set by the loading function. Would it make sense to modify Figure B1 to reflect that (if indeed correct)? I also think some empirical patterns, showing what the curvature parameters is, would also help increase the intuitiveness of the results. A demonstration of how curvature is fit would be good in the appendix (to make section 2.1 easier to understand).*

Thank you for this comment. A demonstration of the fitting of Curvature is shown in the response of comment WW.6 and replaced as Fig. B1 in the revised manuscript. Nevertheless we agree that showing the slope of the loading function can help the reader to interpret the bending and therefore altered Fig. 2a and b to show the conservative scenario with no uptake (vf=0).

*WW.9: I appreciated the test of the model predictions against observations in the Selke watershed. The correspondence looks excellent! But I did not quite understand how the seasonality of concentration emerges give the low removal proportions (I assume this is network scale removal by the entire network), and the fact the loading C vs. Q relationship is flat (b = 0.014). It seems that loading is fairly constant and removal in Figure 2a is very small (<5% at all times). So what causes the large drop during summer? I would add another line that represents the export assuming conservative mixing (Vf = 0). Also, in Figure 3, add the observed C vs. Q relationship.*

We thank the reviewer for these comments. In Fig. 2 of the manuscript we show the daily median removal percentage (the ratio between the daily total removed load and the daily total incoming load in the river network), so although the absolute uptake can be significant (we added a conservative mixing scenario to Fig. 2 a and b), the percentage load removed can appear small in the entire river network due to the presence of inefficient grid cells. Part of the seasonality of the concentration is driven by the streamflow as the input load $L = a*Q^b+1$, so in the case of b= 0.014, higher Q will still result in higher incoming loads everywhere in the river network although the concentration remains relatively stable around 3 mg/L. In summary, if the removal overall in the catchment is fairly small, this does not mean that there are no locations and times in the network where removal percentages are high (see headwaters in Fig. 3).

For the comment regarding Fig. 3 we would like to refer the reviewer to our response on WW. 34.

*WW.10: What is driving the runoff (water transfer from land to water) variability over time in each watershed?*

Runoff is driven by the variability in meteorological forcings (e.g., P, T, ...); which afterwards is modulated by land-surface properties (e.g., terrain, soil, vegetation, and geological attributes). In the context of this study, the water land to stream transfer over time is dictated by the discharge time series at the catchment outlet. The observed discharge daily discharge variability at the outlet is distributed to the individual stream sections according to their upstream area with the assumption that the discharge [mm/d] on each day is spatially homogeneous. We mentioned this explicitly in l.163-164 of the methods section in the revised manuscript and in Sect 2.3.1 when introducing the 13 selected catchments (see comment Ed.2).

**WW.11**: *While the conclusions provide clear and useful summaries, I found the final conclusion seemed underwhelming. I think more of the implications of these findings could be emphasized, and why they would be useful. Tie back to the big picture of C vs. Q, role of network removal, and management.*

Thank you for these useful suggestions. We revised the final conclusion emphasizing the implications of our findings in context of C-Q relationships and from the view-points of network removal and management aspects. Specifically we added to l. 685: "This also stresses the need to monitor the entire discharge range and capture low flows as well as high flows in a catchment." and l.702 is completed with: "Consequently, anthropogenic impacts in terms of channelization of river networks might lead to lower removal efficiencies."

**WW.12**: *Line 116: should read "log" C-Q*

We thank the reviewer for this remark. We adjusted the notation to read log(C)-log(Q) throughout the lines 117-124.

**WW.13:** *L 135. Where does the value "402" come from?*

The maximum number of coupled C and Q samples within one station is 402. This is now specified in the revised manuscript.

**WW.14**: *L137. Meaning that at least 10% of the observations come from every season? Still, less sampled seasons could be underrepresented. What seasons were most samples collected?*

We have added a new figure to the supplementary information of the revised manuscript (Fig. S2; see also below) displaying the mean number of observations and the standard deviation for each season. In the fall, spring and summer there were on average 35 samples collected per station while in the winter

the average number of collected samples was 30. We argue there was no underrepresentation of a given season, which we now mention when presenting the French data.

[Figure]

**Figure S2: Mean number of coupled French C-Q samples (Dupas et al., 2019) in each season with their corresponding standard deviation.**

*WW.15: L182. What does this parameter definition mean?*

The ratio of $a_d$ to $a_w$ corresponds to a parameter $r$ [-]$\in R^+$which prescribes the cross section geometry relation such that a triangular channel cross section is represented by $r$ = 1, a parabolic channel cross section by $r$ = 2 and channel cross sections with progressively flatter bottoms and steeper banks by increasing values of $r$ (Dingman, 2007). The width-discharge relation in Eq. (2.1) is conceptually illustrated in Fig. B3 for two sets of $a_w$ and $K_w$ (l.181-183). To make it clearer, we have changed the order of these sentences so that they come directly after the definition of the parameters in l. 176.

*WW.16: L196. Why does the equation have "b+1" rather than just b?*

Because the model is mass balance based, we calculate with load (L) rather than concentration (C). As $L = CQ$ and $C = cQ^b \leftrightarrow L = cQ^{b+1}$

*WW.17: L238. Explain what PAWN stands for when first introduced.*

PAWN is derived from the authors names (Pianosi and Wagener) - who introduced this method - and as such it does not have any meaning. Thus, we do not report what PAWN stands for, since it is not relevant to the analyses.

**WW.18:** *Table 1. Kw is not unitless, it has units of the dimension. (it is equivalent to the width at 1 m3/s or whatever units of Q you use). Same with Kd.*

We thank the reviewer for pointing this out. The units of $K_w$ and $K_d$ are adjusted to $[L^{1-3*a_w}.T^{-a_w}]$ and $[L^{1-3*a_d}.T^{-a_d}]$ respectively in Table 1 (Dingman, 2007).

**WW.19:** *Table 2. Please add the watershed scale runoff (mm/d) to this table. It will allow comparison of how the different watersheds function. Q at the outlet is then just that times the watershed area. Is median Q the median of all river reaches, or the median at the mouth over time?*

We thank the reviewer for this comment. The watershed scale runoff is now added to Table 2. The median Q is taken as the median discharge at the basin mouth over time. This is specified in l. 267.

**WW.20:** *Table 3. Why such small ranges for some of these parameter but not others?*

We distributed the non-missing simulation data over 20 percentiles and selected the percentiles corresponding to low, medium and high values (according to literature). Thus each class can have a different range; however for one variable the number of 'simulation data points' in each class is the same. This has been clarified in the header of table 3 in the revised manuscript.

**WW.21:** *Figure B4. Define the variables*

We thank the reviewer for this comment. We added the definition of each variable to the caption of Fig. B2 (revised figure number) as follows: "Here, the flow length through a grid cell $i$ is $l_i$ [L], $w_i$ [L] and $d_i$ [L] are the respective width and average depth of the reach and $P_i$ [L] is the corresponding stream channel wetted perimeter. The uptake velocity is denoted as $v_f$. The local discharge $Q_i$ [L³ T⁻¹] consists of upstream incoming discharge $Q_{i-1}$ [L³ T⁻¹] and land to stream runoff $Q_{ls}$ [L³ T⁻¹]. Similarly, the local load L [M T⁻¹] consists of upstream incoming load $L_{in.up}$ [M T⁻¹] and the land to stream load $L_{in.ls}$ [M T⁻¹], where $L_{in.} = L_{in.up} + L_{in.ls}$. Finally, the local load removed is denoted as $L_{r,i}$ [M T⁻¹]"

**WW.22:** *Table C1. The parameters for the Selke catchment suggests that inputs of NO3 are relatively chemostatic (fairly low "b"). This would lead to C vs. Q flattening out at high flows. It may be helpful to include a "conservative tracer" scenario to each of the catchments, which will be based on the C vs. Q of*

*loading from the landscape. The divergence (always lower), will indicate bentness. Consider representing Figure B1 in this way.*

We thank the reviewer for this helpful comment. Comparing the divergence between the conservative tracer scenario and the resulting log(C)-log(Q) curve would indeed be another way to indicate 'bentness' and the effect of instream uptake. In this paper however 'bentness' is quantified with the *Curvature* metric as no conservative tracer scenario is needed to interpret it. Nevertheless, we agree that indicating the conservative tracer scenario is useful in this explorative approach at least as an example in the Selke Meisdorf case. We refer the reviewer to the response to WW.9 for more details.

**WW.23**: *L295. Explain what KSmax means in words and whether high values are better or worse.*

This information was indeed missing here. We added a sentence in the revised manuscript so the section would read: "In this study, we applied Eq. (7) using $n_i = 10$ conditioning intervals for each input parameter and used the maximum KS value, $KS_{max}$ (ranging from 0 to 1), as a summary statistic, which is appropriate for screening non-influential input parameters. For a given parameter, the highest value of $KS_{max}$ of 1 would indicate a direct dependence of the model output (in this case *Curvmax*) on that parameter, while a value of KS_max of 0 would mean that the parameter is completely non-influential."

**WW.24:** *L312. I am not sure that the catchment wide Da adds much to the overall analysis, and could be dropped.*

We thank the reviewer. The catchment wide Da was included to check if the simulated values distribute around 1 and help the reader to understand if our scenarios rather create overall more reaction or more transport driven cases. This was mainly motivated by the surprisingly low impact of the vf and the prominent role of velocities on the uptake and bending. We thus wanted to explore if all our catchments are just transport driven which is not the case. We prefer to keep the Da number here but now better explain our intention with Da in the revised text (l.468-471).

**WW.25:** *L356-358 and Figure 3. The comparison of % removed and absolute amount removed within each grid cell is interesting and useful, but not the complete story. There are many more medium and large river grid cells than headwater grid cells along any nutrient loads flow path. So cumulative removal by larger rivers likely approaches or maybe even surpassed that of cumulative removal by the headwaters, particularly at high flows (see Wollheim et al. 2006 and 2018). Consider adding that metric as well.*

Interesting point. We added an inset to Fig. 3 that shows the annually removed load within a certain Strahler stream order, for each of the grid cells in that stream order and described these results in the text. There are more first order grid cells that are more efficient in NO3- removal. So they will make up the largest portion of NO3- removal in this example. Although there are indeed more medium and large grid cells along any nutrient flow paths, these grid cells are less efficient at removal as total incoming loads to these grid cells increase in a downstream direction. However this is not a linear relationship as there are more third than second order grid cells in the Selke Meisdorf river that have similar removal efficiencies. Thus third order removal capacity surpasses that of second order removal capacity.

[Figure]

**Figure 3. Spatial distribution of simulated *Curv_max* in the Selke river network (Meisdorf) for a selected parameter set (see Table C1). Three representative grid cells covering low (A), intermediate (B) and high (C) total drainage areas show the incoming land to stream load as (Eq. (3)), the incoming load from upstream as (Eq. (3)), the absolute removed load as (Eq. (5)) and the outgoing load as (Eq. (4)) in the log(L)-log(Q) space. The load removed as a percentage of the incoming load is presented on the secondary axis. Note that the corresponding *Curv_max* for these grid cells are calculated from the log(C)-log(Q) relationships rather than log(L)-log(Q). The insets show the distribution of *Curv_max* and for each of grid cells within a certain stream order. In Fig. 2a the observed and simulated concentrations are compared at point A.**

We also described our findings in the text of the revised manuscript and added your comment on the cumulative load removed per stream order to the discussion in l.395: "The total absolute load removed ( , sum per year per grid cell) is largest for first order grid cells (average 24.1 kg N year$^{-1}$) that represent 55 % of the river network, followed by second and third order grid cells (averaging both around 20 kg N year$^{-1}$) that represent 20 and 25 % of the network (inset Fig. 3). Finally, the total yearly incoming load (Lr , sum per year per grid cell) increases with stream order from 1329 kg N year$^{-1}$ on average in a first order grid cell to 5128 and 42124 kg N year$^{-1}$ in second order and third order river cells. " and l. 416:"Note that the annual percentage load removed accounts for load taken up throughout the entire river network, which may be higher in the headwaters (15 tons) than in downstream locations (7 and 5 tons for second and third order stream sections; inset Fig. 3) "

*WW.26:L384. Wouldn't median over represent low flow periods, rather than total fluxes (since most flows are low, storm flows relatively infrequent).*

That is a good point. With the median we focus not on the total removed load but on how frequent is a certain removal efficiency. This is what we already state in the preceding lines l.423-427. The alternative (removal based on total, cumulative fluxes) would heavily weight single large discharge events.

*WW.27: L416. It is not clear in the table of watershed characteristics why C1 and C10 have so much higher Lr.perc than the others. What causes the large variability among watersheds? Cumulative percent removal should always increase with watershed size. Are you reporting the median within a watershed? I think cumulative removal would be a better metric.*

We thank the reviewer for this comment. It is indeed true that we report the median removal in each river network (see response to comment WW.26). We already discuss some possible reasons for the higher efficiency in C1 and C10 in l.413-l.419 in the manuscript: "The percentage load removed, $L_{r.perc}$, is notably lower catchments with high Q – like 3, 4 and 8 (Table 4) which follows the narrative in Sect. 3.1 that uptake efficiency decreases with increasing Q because of increasing loads to the system (Wollheim et al., 2018; Mulholland et al., 2008) that also result in less efficient uptake within the reactive surface area (Peterson et al., 2001; Hensley et al., 2014). The high $L_{r.perc}$ in small catchments 1 and 10 could then be attributed to their low Q, however why the small catchment 5 does not have similar uptake performance is less clear." Nevertheless we agree it is helpful to add the 50$^{th}$ percentile to

Table 4 in the revised manuscript. Also revised this section to include the discussion of the effect of the runoff [mm/day], added in Table 2 (WW. 19).

*WW.28: L458. I have a hard time understanding why catchments results are distinct, when all the parameters are the same. L461 says local loading and uptake differed, but what basis, since all the parameters are the same! Some of the other explanations in this paragraph are similarly unclear. It seems the model predictions can be summarized to see if the statements are true.*

We would like to thank the reviewer for these comments. We used a fixed set of 11107 parameter combinations (l.230) in each of the study catchments. During a model simulation, one of these parameter combinations was applied and the parameters are kept constant in space and in time for simplicity. However, all catchments do have an individual network structure and individual discharge conditions that largely explain the spatial differences between the catchments. For example, the channel hydraulic variables (w and d; Eq. 2) can vary significantly depending on the discharge values in each study catchment (Q). We clarified this in Sect 2.3 and l. 500.

*WW.29: L470. Is Q higher is some catchments because they are stormier (runoff vs. Q focus). Q integrates watershed size and storminess.*

See WW.19, we added runoff [mm/d] to Table 2. The runoff median runoff at the catchment outlet ranges between a minimum and maximum value of 0.07 and 1.81 mm/day respectively for catchments 1 and 4. Both, absolute Q and Q variability are mainly the response to the climatic drivers precipitation and evapotranspiration. In Germany, the climatic drivers follow an East to West gradient and depend on the altitude. The storminess is captured by the CV of Q that is reported in Table 2 and not correlated to absolute [m3/s] and specific [mm/d] discharge. We now refer to Table 2 in the text at this point.

*WW.30: L473. Is the runoff the same in the small catchments as the large?*

We refer the reviewer to the response on comment WW.29.

*WW.31: L476. Important point! What about flow regime (frequency of different runoff events over time). Are they similar among catchments?*

This is nicely captured by CV (Q) which integrates the frequency of runoff events and the differences in recession constant (so the catchments "flashiness" in response to rainfall) (Botter et al., 2013). We added some statements on the Q and CV (Q) differences in the header of Table 2.

*WW.32: L511. Replace "Curvature" with "Curvature Parameter" because less curvature is more bent.*

We replaced "*Curvature*" with "*Curv$_{max}$*" to distinguish between the mathematical meaning of curvature and the metric "*Curvature*" we introduced. Also see comment Ed.1.

*WW.33: L641. I think to make this conclusion, you need to include more empirical relationships.*

We thank the reviewer for pointing this out. In the revised manuscript we now stress that these conclusions hold true for the data set and parameter range that we used for our analysis. As we mentioned in the paper outlook, enlarging this approach to more catchments and gathering more empirical evidence to explore this further would have to be done in future work. We stress in l.678 of the revised manuscript that our results suggest this.

*WW.34: Figure 3. Add the observed C vs. Q (fitted relationship, with their R2) as a model test to this figure. Important to know how close predictions come to observation*

We thank the reviewer for this comment. In Fig. 2 simulated and observed $NO_3^-$ concentrations are shown at the Selke Meisdorf station with the goodness-of-fit metrics NSE and pbias. Because adding those fits in Fig. 3 as well would make the figure harder to read and repetitive we would not follow your suggestion here. Nevertheless, in the revised manuscript we refer to Fig. 2 at this point.

*WW.35: Figure 5. Nice summary of all the correlations, with color coding.*

We thank the reviewer for this comment.

*WW.36: Figure 7. I found this figure to be impossible to interpret. I think more explanation in caption needed. What are the histograms? What are the decision values? Why do variables show up multiple times? Not sure how useful the Cart analysis is based on the discussion here.*

Thank you. We altered this Figure to the one as seen below and clarified the caption.

[Figure]

[Figure]

**Figure 7: CART decision trees for the response variables L_(r.perc) (accuracy = 0.66), Da (accuracy = 0.51) and v_f (accuracy = 0.40). The histograms illustrate the probability of a test sample to be from a certain response variable class (low, medium or high; Table 3) when following the binary splits of the decision tree. The variables at the binary splits differ per response variable and consist of the median stream velocity, v [m day-1] and Curvmax for response variables L_(r.perc) and v_f, while width coefficient K_w and depth exponent a_d are used additionally for Da. The prediction metrics for each of these classes and response variables are stated in Table C3.**

Generally, we show the CART analysis because it is a visual guide through the multivariate space. Simple correlations do not capture parameter interactions and we therefore argue that CART is a valid tool here. The variables that appear in the internal nodes of the tree can be interpreted as being influential with respect to the dependent variables considered (here Lr, Da and vf). Variables can show up multiple times in the tree, revealing interactions between variables for different values of that variable. CART has been applied before in the context of sensitivity analysis, e.g. in Almeida et al. (2017) to identify the controls of landslides and in Singh et al. (2014) to identify the controls of runoff. In the manuscript, we clarified the objectives of the CART analysis and link them to the previous analyses (PAWN, correlation).

REFERENCES

(Only the references that are not yet mentioned in the manuscript reference list are stated here)

*Almeida, S., Ann Holcombe, E., Pianosi, F. and Wagener, T.: Dealing with deep uncertainties in landslide modelling for disaster risk reduction under climate change, Nat. Hazards Earth Syst. Sci., 17(2), 225–241, doi:10.5194/nhess-17-225-2017, 2017.*

*Singh, R., Wagener, T., Crane, R., Mann, M. E. and Ning, L.: A vulnerability driven approach to identify adverse climate and land use change combinations for critical hydrologic indicator thresholds: Application to a watershed in Pennsylvania, USA, Water Resour. Res., 50(4), 3409– 3427, doi:10.1002/2013WR014988, 2014.*

*Botter, G., Basso, S., Rodriguez-Iturbe, I., & Rinaldo, A. (2013). Resilience of river flow regimes. Proceedings of the National Academy of Sciences of the United States of America, 110(32), 12925-12930. doi:10.1073/pnas.1311920110*

---

## Author Response (AR2)

Response to Reviewer Will Wolheim –

We thank the reviewer for their comments and constructive feedback. We respond directly below each comment (Italic) in normal font.

**WW2. 1**: *I appreciate the addition of an example of how the curvature parameter (now called Curvmax) was calculated (Figure B1). Based on this example, it looks like a linear relationship (in log-log space) would be adequate. Why add the complexity of more degrees of freedom? But that would also mean no curvature. It seems like a metric like AIC would help identify what the best fits are for each site. Given the fits in this figure, how does Local Curvature vary so much in opposite directions and sign? It was unclear which of the fits was used to identify Curvmax in this particular example. If the estimate is so variable, it seems like the noise in the data will contribute greatly to a Curvmax. An objective approach (like AIC) for selecting curvature should be used. This should be very clear in the ms given this is a new approach that is being argued will be helpful.*

We thank the reviewer for these remarks. Several studies (eg. Moatar et al., 2017; Diamond and Cohen, 2018 and Marinos et al., 2020) show that characterizing NO3- log(C)-log(Q) relationships as linear can imply an information loss (l. 88). For example, Moatar et al., 2017 found that 44% of the investigated catchments have higher low flow slopes than high flow slopes for the NO3- log(C)-log(Q) relationship. In Marinos et al., 2020 NO3- log(C)-log(Q) relationships are more accurately represented with a piece wise regression model than a linear regression for 32 of 33 study sites. Based on these and other studies we argue that more degrees of freedom are necessary to characterize NO3- C-Q behaviour in the log-log space.

The Curvmax metric, introduced in this study, complements rather than replaces the established linear regression model, as it allows to quantify the shape of log(C)-log(Q) relationship without the assumption of a fixed form (l.117-121). The Curvmax metric can theoretically range from 0 to + or − infinity, with CurvMax= 0 indicating no bending in the log(C)-log(Q) relationship. Because we study the degree of (non-)linearity rather than choosing between a linear or non-linear model, AIC would not be helpful here.

We explain in l. 124-134 of the manuscript how CurvMax is calculated: "A smoothed spline, $f$, is iteratively fitted with increasing degrees of freedom (df) to capture the general log(C)-log(Q) shape accurately but avoid overfitting (Fig. B1). Initially, df = 3 and the log(Q) region of the largest instantaneous change is identified as $Q_m\pm0.05$ with $Q_m = arg\max_{\log Q}|f''|$. Then, df is increased until, at df=i, the log(Q) corresponding to the largest instantaneous change is not within the initial $Q_m$ region anymore. Consequently, *Curv_max* is calculated for a smoothed spline fit, $f$, with df = i-1 as

$$\begin{cases} \max_{\log Q} f'' \ if \ \left|\max_{\log Q} f''\right| \geq \left|\min_{\log Q} f''\right| \\ \min_{\log Q} f'' \ if \ \left|\max_{\log Q} f''\right| < \left|\min_{\log Q} f''\right| \end{cases}$$

." Because of the nature of the smoothed spline fits, local curvature can vary, especially with higher degrees of freedom. We added a sentence in the caption of Fig. B1 explaining that df=5 was chosen as the final fit.

We examined the robustness of the CurvMax metric by selecting subsamples of observed noisy C-Q data from 444 French stations (Dupas et al., 2019) without replacement but with overlap (l.143-149). The results of this assessment in Fig. B4 shows that Curvmax tends to be robust (Sect. 3.1).

**WW2. 2:** *I think the results also indicate, as discussed by the authors, that linearity may occur (no bentness), across a wide range of uptake velcoity values (Figure B6). Figure B6 suggests most logC vs. LogQ relationships will likely be linear or close to linear when using observational data sets, which likely have enough noise to make it hard to distinguish the slight curvature that may occur. Thus, bending may not be observed, even if network scale uptake is high. The key for determining whether network scale uptake is high is knowing what the loading (vf=0) scenario is. Use of conservative tracer (chloride or specific conductance) may help with this interpretation (if you have those in your observational data sets).*

We thank the reviewer for this comment. Figure B6 mainly shows that the Curvmax metric can remain nearly constant under varying vf values if vf is not close to 0. We showed in Sect 3.1 and Fig. B4, where we calculate CurvMax for NO3- log(C)-log(Q) relationships of 444 French stations that Curvmax ranges between -5.25 and 3.88 (l.353). Moatar et al., 2017 found that for NO3- 44 % of the studied catchments can be classified as bended and 33 % as linear. If in our case a slight curvature occurs we would capture a CurvMax close to 0 thereby not classifying the C-Q relationship as bended or linear. We found that 77 % of the French stations have CurvMax ≤ 0 which is similar to the findings of Moatar et al., 2017. We acknowledge that conservative tracers can help to interpret CurvMax which can be an interesting approach in future studies. We added this aspect to the manuscript in the outlook in l. 718.

**WW2. 3:** *I think more succinct discussion is warranted on what circumstances (across the vf, aw, watershed size and other parameter space) curvature could be evident in observational data sets. I realize the paper is about bentness, but in the end that is just an abstraction. Ultimately bentness will be helpful to understand when river network scale uptake is important, and clearly bentness alone is insufficient for those purposes. I think the regression tree is meant to address this, but I find it very confusing to understand. Perhaps another approach or conceptual figure would help.*

We like to thank the reviewer. To solve this concern we would need more direct measurements of uptake together with observations of C and Q to make this linkage not only in the model but also in the data-reality world. However this would be outside of the scope of this paper where we offer a first insight in what can cause C-Q bending, using conceptual modelling. We indeed use the regression trees as well as Fig. 6 and Fig. B7 to link curvmax values to other parameters. To help with the interpretation of the decision tree we added a sentence in the caption stating that the trees are read from top to bottom, following the binary splits.

**WW2. 4**: *I am still puzzled by the results presented in Figure 2, even after reading the response to reviewers about my earlier question. The percent removed is very low (<5%) throughout the time series. The "no uptake" scenario results in C ~3mg/L, and observations fluctuate between 0.5 and 3mg/L. Based on this, I would expect to see removal fluctuating between 0 and 85%. And again (as with Figure B6), given the noise in the data in Figure 2b, would a nonlinear fit be selected over a linear fit using something like AIC to select the most parsimonious model in the lower panel?*

We thank the reviewer for this comment. The concentration time series in Fig. 2a is observed at the outlet of a catchment. We agree that if we would consider the percentage load removed solely at the outlet grid cell, removal would fluctuate between 0 and 85%. However, the percent load removed is

simulated at the network scale. Which means it considers the incoming and outgoing loads at each network gridcell and is displayed in Fig. 2a as a median value. As we do not choose between linear or non-linear models but rather investigate the "degree of non-linearity" in this study, computing AIC would not help here (also see WW2. 1).

**WW2. 5**: *Regarding the use of uptake velocity in the analysis (comment Ed. 6) I strongly agree that it is the correct term to use. It is equivalent to piston velocity for gases, or settling velocity for sediments.*

We thank the reviewer for reaffirming this.

**WW2. 6:** *I think some of the other reviewers comments are valid regarding the density of results being presented. There is also a lot of information in some of the figures, and it is hard to identify the main the result that should be gleaned from it (Figure 3). More statements in the text guiding the reader as to the gist of the results would be helpful, especially in the topic sentence of paragraphs, to better tell the story, and make the paper more influential. The discussion in 3.3 in particular gets really difficult to follow.*

We thank the reviewer for helping to improve the readability of the manuscript. We revised the discussion in Sect 3.3.

**WW2. 7**: *What is the difference between the Figure B\* series and the appendix figures? It would be simpler to have all in appendix for navigation purposes*

We thank the reviewer. We follow HESS guidelines for manuscript composition here: "Additional figures, tables, as well as technical and theoretical developments which are not critical to support the conclusion of the paper, but which provide extra detail and/or support useful for experts in the field and whose inclusion in the main text would disrupt the flow of descriptions or demonstrations may be presented as appendices. These should be labelled with capital letters: Appendix A, Appendix B etc. Equations, figures and tables should be numbered as (A1), Fig. B5 or Table C6, respectively. Please keep in mind that appendices are part of the manuscript whereas supplements (see below) are published along with the manuscript." We would therefore keep the Appendix figures (B-series) in the Appendix and the supporting figures in the supporting information (S-series).

**WW2. 8:** *L497. Most Da do not seem to be around 1.*

The Monte Carlo output of log(Da) is displayed for each of the catchments below. Here we see that the simulations are distributed around Da equal to 1 (log(Da)=0). We added this plot to the supporting information in the revised manuscript as Figure S3.

[Figure]

**WW2.9**: *L546. I don't agree with the statement "lower Curvmax" is related to high Lr.percent. It implies this condition is needed. If uptake velocity is very high, there is no bending, yet network scale removal can be very high. Same with L585, vf is clearly also important. Please modify wording.*

We agree with the reviewer that this wording can be misleading. In the first case we changed the sentence to "…lower curvmax can be related to higher Lr.perc" in the second case we have changed the verb to 'correlate'.

**WW2.10**: *Conclusions. I recommend making clearer what the take home messages are, rather than bringing in results again. E.g. L689, I think the main point is that bending is only evident under certain geomorphological conditions. I also disagree with statement under this bullet that vf does not influence bending. Curvmax=0 both when there is no uptake, and when uptake is really high (vf high), meaning you can only see bending at intermediate vf. In bullet 4, I think the takehome should be that Curvmax helps with estimating Lr.perc, but only under certain conditions.*

We thank the reviewer for these remarks. We state that bending is shaped primarily by geomorphological parameters while vf has a secondary influence. In Fig. 6 we demonstrate that if we select simulations corresponding to ranges for low, medium and high vf the associated distributions of CurvMax do not differ much and vice versa. Which means you can see bending for low, medium or high vf. We agree with the suggestion of the reviewer on the take home message of the 4th bullet point and included this sentence accordingly in the revised manuscript.

**WW2.11**: *L707. I did not find interpreting the Cart tree simple.*

We changed this sentence in the revised manuscript to remove the word "simple".